# Persistent equatorial Pacific iron limitation under ENSO forcing

Thomas J. Browning[1✉], Mak A. Saito[2], Shungudzemwoyo P. Garaba[3], Xuechao Wang[1], Eric P. Achterberg[1], C. Mark Moore[4], Anja Engel[1], Matthew R. McIlvin[2], Dawn Moran[2], Daniela Voss[3], Oliver Zielinski[3,5,7] & Alessandro Tagliabue[6]

Projected responses of ocean net primary productivity to climate change are highly uncertain[1]. Models suggest that the climate sensitivity of phytoplankton nutrient limitation in the low-latitude Pacific Ocean plays a crucial role[1–3], but this is poorly constrained by observations[4]. Here we show that changes in physical forcing drove coherent fluctuations in the strength of equatorial Pacific iron limitation through multiple El Niño/Southern Oscillation (ENSO) cycles, but that this was overestimated twofold by a state-of-the-art climate model. Our assessment was enabled by first using a combination of field nutrient-addition experiments, proteomics and above-water hyperspectral radiometry to show that phytoplankton physiological responses to iron limitation led to approximately threefold changes in chlorophyll-normalized phytoplankton fluorescence. We then exploited the >18-year satellite fluorescence record to quantify climate-induced nutrient limitation variability. Such synoptic constraints provide a powerful approach for benchmarking the realism of model projections of net primary productivity to climate changes.

Model projections of global marine net primary productivity (NPP) for the year 2100 range from increases to declines by as much as 20% (ref. 1). These divergences largely centre on differences at low latitudes and result from inter-model variations in the underlying mechanisms regulating NPP[1–3]. Such uncertainties are important as they will lead to cascading impacts on ecosystems, ocean biogeochemistry and carbon cycling in models. Climate-regulated NPP changes of several per cent already occur naturally at interannual scales, driven by the impact of the El Niño/Southern Oscillation (ENSO) in the equatorial Pacific[5–7]. These changes in NPP are believed to be driven by variations in the upwelling rate of deep-water nutrients that controls their supply to phytoplankton[5,6,8], with uncertainties in future projections strongly linked to changes in growth-limiting nutrients[3].

Assessing limiting nutrients, their geographic boundaries and their variation over ENSO cycles is key to understanding contemporary ocean NPP and informing climate model projections[4]. In that way, historical variability can act as an 'emergent constraint' on Earth system model projections[7]. Available methods for determining nutrient limitation have so far centred on ship-based observations, including simple environmental predictions based on nutrient concentrations, biochemical signals linked to nutrient stress, and growth limitation tests using biomass changes following nutrient amendment[9–12]. These different approaches deliver unique information from population to community levels that are not directly comparable[9,13]. Furthermore, even with coordinated programmes and the highest throughput approaches[10,12], all are ultimately restricted to documenting snapshots in time and space that are of limited utility in probing large-scale temporal changes. Attempts have been made to scale nutrient-related phytoplankton

fluorescence properties to global remote sensing[14–16], which would offer unparalleled advantages in spatial and temporal data availability, but a range of uncertainties have restricted usage[15,17–21]. Here we tackle these challenges for the tropical Pacific by building an observational dataset of nutrient limitation spanning physiological to community-level scales. Moreover, by mechanistically connecting these ecophysiological signals with coincident measurements of radiometric quantities, we directly link nutrient limitation to satellite observations, to assess large-scale variability in nutrient limitation and constrain ocean models.

## Contrasting nutrient-limited physiology

Field observations were conducted along an approximately 5,000-km-long transect across the tropical Pacific Ocean (Fig. 1 and Extended Data Fig. 1). Nutrient-addition bioassay experiments and nutrient-stress biomarker proteins both revealed a consistent trend from iron (Fe) to nitrogen (N) limitation, which matched the prevailing nitrate gradient (Fig. 1a–d). Specifically, where nitrate concentrations were elevated (Fig. 1a,b), Fe amendment stimulated chlorophyll $a$ biomass increases (sites 2 and 3; one-way analysis of variance (ANOVA) $\alpha = 0.05$, Tukey's honestly significant difference test). Conversely, where nitrate was low at sites 4 and 5, supply of N stimulated small chlorophyll $a$ biomass increases (Fig. 1c and Extended Data Fig. 2), and the in situ abundances of several proteins reflecting N stress (urea ATP-binding cassette transporters, the signal transduction protein P-II, the global N regulator NtcA, and aminotransferase)[22] for *Prochlorococcus*—an important contributor to phytoplankton

[1]Marine Biogeochemistry Division, GEOMAR Helmholtz Centre for Ocean Research Kiel, Kiel, Germany. [2]Woods Hole Oceanographic Institution, Woods Hole, MA, USA. [3]Institute for Chemistry and Biology of the Marine Environment, University of Oldenburg, Oldenburg, Germany. [4]School of Ocean and Earth Science, National Oceanography Centre Southampton, University of Southampton, Southampton, UK. [5]German Research Center for Artificial Intelligence (DFKI), Oldenburg, Germany. [6]Department of Earth, Ocean, Ecological Sciences, University of Liverpool, Liverpool, UK. [7]Present address: Leibniz Institute for Baltic Sea Research Warnemünde (IOW), Warnemünde, Germany. ✉e-mail: tbrowning@geomar.de

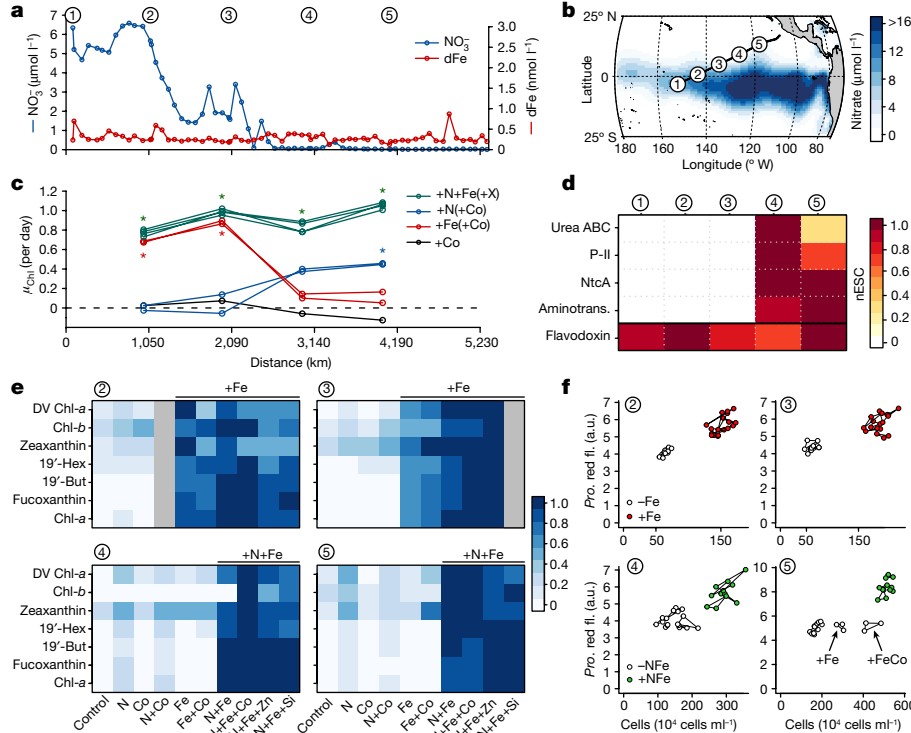

**Fig. 1 | Tropical Pacific nutrient limitation transition. a**, Concentrations of nitrate and dissolved Fe (dFe), with vertical axes scaled according to assumed-average phytoplankton requirements (such that the lower line is the more deficient nutrient; N/Fe of 2,132:1 mol/mol). Circled numbers represent major sampling sites. **b**, Broader regional nitrate gradient. **c**, Summarized net chlorophyll *a* growth in nutrient-addition experiments. Dots are means of triplicates; overlapping lines or symbols of the same colour indicate equivalent treatments except that another nutrient was added in addition to N and/or Fe (either Co, Zn or Si; as indicated by '+X' for N + Fe lines). Asterisks indicate significant chlorophyll *a* enhancements over non-amended control (ANOVA followed by Tukey's post hoc test, $\alpha < 0.05$; Extended Data Fig. 4). **d**, Abundance of *Prochlorococcus* nutrient-stress biomarker proteins; nESC, normalized

biomass (Extended Data Figs. 3 and 4)—all increased (Fig. 1d and Supplementary Tables 1–3). In the bioassay experiments, an array of diagnostic pigments responded similarly to chlorophyll *a*, suggesting that most of the phytoplankton community experienced the same nutrient-limitation regime (Fig. 1e). A level of N–Fe co-stress was also identified at the predominantly N-limited sites 4 and 5, where the Fe stress biomarker flavodoxin was detected[23], N + Fe supply further boosted phytoplankton pigment biomass accumulation in comparison to N alone (Figs. 1c,e and Extended Data Fig. 2), and net accumulation of *Prochlorococcus* cells and increases in chlorophyll *a* per cell were both largest following N + Fe treatment at sites 4 and 5 (Fig. 1f and Extended Data Fig. 5).

A major shift in diel cycles of active fluorescence properties was observed across the nutrient limitation transition ('active' fluorescence denoting stimulation by high-intensity blue light flashes; Fig. 2; refs. 10,24,25). Active variable fluorescence ($F_v$) normalized to maximum fluorescence ($F_m$) showed both lower dawn and dusk values at Fe-limited sites and pronounced daytime and night-time reductions, with the latter matched by synchronous reductions in functional absorption cross-sections of photosystem II (PSII), $\sigma_{PSII}$ (Fig. 2a,b). By contrast, under N limitation, dawn and dusk $F_v/F_m$ was elevated and night-time reductions were smaller (for $F_v/F_m$) or eliminated (for $\sigma_{PSII}$). Such changes in diel active fluorescence properties in this region are well documented[10,24–26] and have been suggested to be due to declines in Fe-rich photosystem I (PSI) and cytochrome $b_6f$ components relative

exclusive spectral counts (for which counts have been normalized to the maximum value at each of the five sites); Aminotrans., aminotransferase; urea ABC, urea ATP-binding cassette transporter. **e**, Responses of individual diagnostic pigments (concentrations normalized to the highest concentration in any treatment). Grey shading indicates not determined. DV Chl-*a,* divinyl chlorophyll *a*; Chl-*b,* chlorophyll *b*; 19'-hex, 19'-hexanoyloxyfucoxanthin; 19'-but, 19'-butanoyloxyfucoxanthin. **f**, Responses of *Prochlorococcus* (*Pro*.) with treatments limiting to the bulk community shown in **c** highlighted by dot colour. Red fl., red fluorescence per cell; a.u., arbitrary units. Triplicate, biologically independent replicates for each treatment are shown as individual points, with the same treatment joined with lines.

to PSII, which contains less Fe (Fig. 2c; refs. 27–29). Such changes in photosynthetic components have not previously been demonstrated in the field, but would probably lead to a more reduced night-time plastoquinone pool in cyanobacteria, in turn triggering night-time energetic decoupling of pigments from PSII (Supplementary Discussion 1; refs. 10,24). Our metaproteomic analyses for field populations of *Prochlorococcus* demonstrated that proteins associated with PSII showed minor changes across the Fe-to-N limitation transition, whereas those associated with PSI and cytochrome $b_6f$ were mostly undetectable at the Fe-limited sites 1–3 but abundant at the predominantly N-limited sites 4 and 5 (Fig. 2c,d). Our data therefore provide in situ confirmation of a predicted, major nutrient-driven stoichiometric adjustment in photosynthetic components[28] and its impact on phytoplankton fluorescence properties[10] (Fig. 2c,d, Methods and Supplementary Tables 1–3).

We found that chlorophyll *a*-normalized active fluorescence ($F$/Chl$_{active}$) was an even clearer diagnostic of the nutrient limitation transition than $\sigma_{PSII}$ and $F_v/F_m$ (Fig. 3a,b). Night-time $F$/Chl$_{active}$ values, calculated by normalizing in vivo night-time active fluorescence to solvent-extracted chlorophyll *a* concentrations, were on average 3.2-fold higher under the elevated-nitrate, Fe-limiting conditions in comparison to the low-nitrate zone (Fig. 3b). The 'additional' fluorescence per chlorophyll *a* produced under elevated-N, Fe-limiting conditions can derive both from a higher ratio of fluorescent PSII to minimally fluorescent PSI (Fig. 2c,d) and from Fe-stress-related pigment protein

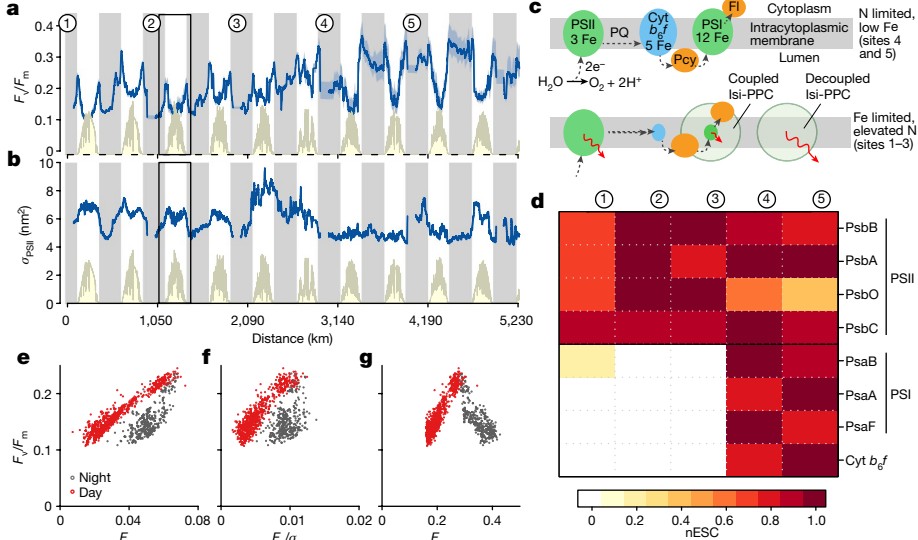

**Fig. 2 | Phytoplankton physiological responses to the Pacific nutrient limitation transition. a,b**, $F_v/F_m$ (**a**) and $\sigma_{PSII}$ (**b**). Light blue shading around the line in **a** indicates the sensitivity of $F_v/F_m$ to the applied blank (mean ± standard deviation; $n = 16$ observations of separate filtrates). Yellow lines and shading indicate sunlight (relative). Daytime $F_v/F_m$ decreases result from the well-known process of non-photochemical quenching[14,15]. Circled numbers represent major sampling sites. **c**, Schematic of the photosynthetic membrane with Fe requirements, highlighting changes in component pool sizes and pigment proteins under the two nutrient limitation regimes (Isi-PPC, Fe-stress-induced pigment protein complex; PQ, plastoquinone)[30–32]. Cyt, cytochrome; Fl, flavodoxin; Pcy, plastocyanin. Red arrows indicate relative fluorescence yields of the different components. **d**, Measured abundances of proteins associated with PSII and PSI. nESC, normalized exclusive spectral counts (for which counts have been normalized to the maximum value at each of the five sites). **e–g**, Controls on night-time $F_v/F_m$ reductions. Data are from a diel cycle highlighted in **a** and **b**.

complexes, which in addition to associating with PSI[30], may accumulate to the point at which a proportion have weak or no energetic connectivity to either reaction centre[29,31–33]. Rapid (44–45 h), threefold reductions in $F$/Chl$_{active}$ following Fe supply at the higher-nitrate sites (sites 2 and 3) provided strong evidence that this signal was Fe regulated (Fig. 3a). Notably, these $F$/Chl$_{active}$ reductions following Fe supply occurred irrespective of the restricted community restructuring (Figs. 1e and 3a and Extended Data Fig. 6). Alongside indistinguishable light absorption properties of bulk phytoplankton communities between the two limitation regimes (Extended Data Fig. 7), both observations suggested that this shift was not due to possible changes in phytoplankton communities with intrinsically different fluorescence

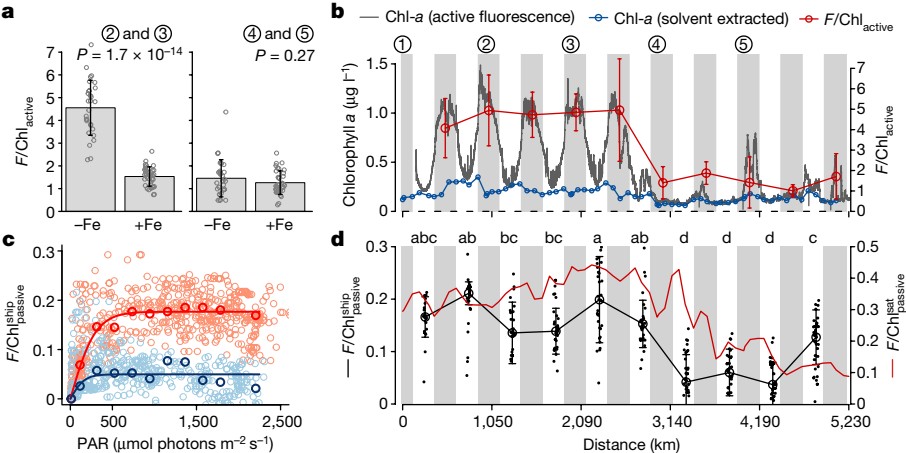

**Fig. 3 | Nutrient regulation of relative fluorescence yields. a,b**, Elevated $F$/Chl$_{active}$ at the high-nitrate, Fe-limited sites, which was reduced following Fe supply. In **a**, bars are means of all experimental treatments with or without added Fe for the Fe-limited sites 2 and 3 (left), and the predominantly N-limited sites 4 and 5 (right). Dots show individual data points, and error bars indicate the standard deviation, with two-sided, unpaired $t$-test $P$ values for significantly different means indicated ($n = 30$ biologically independent samples for −Fe and $n = 36$ for +Fe). In **b**, $F$/Chl$_{active}$ is for night-time fluorescence data alone (points indicate the mean and error bars show the standard deviation; $n = 5–7$ biologically independent samples). **c**, $F$/Chl$_{passive}^{ship}$–photosynthetically active radiation (PAR) response curves from the shipboard radiometry. Red lines are for the Fe-limited zone (sites 1–3) and blue lines are for the predominantly

N-limited zone (sites 4 and 5); bold symbols and lines indicate PAR bin averages with model fits ($F$/Chl$_{passive}^{ship}$ = $a \times \tanh[b \times PAR/a]$, where $a$ and $b$ parameterize the light-saturated plateau and light-limited slope, respectively; ref. 42). **d**, Changes in $F$/Chl$_{passive}$ across the Pacific Ocean transect from the shipboard radiometry and a boreal winter climatological average from the MODIS-Aqua sensor (satellite (sat)). Black dots show individual radiometric observations between 12:00 and 15:00 local time, and open circles and error bars show the mean and standard deviation ($n = 36$ independent radiometric observations). Days with the same letter labels have indistinguishable mean $F$/Chl$_{passive}^{ship}$ values (one-way ANOVA, $\alpha < 0.05$, followed by Tukey's post hoc test). For all panels, units of $F$/Chl$_{passive}$ are Wm$^{-2}$ sr$^{-1}$ µm$^{-1}$ [mg Chl m$^{-3}$]$^{-1}$.

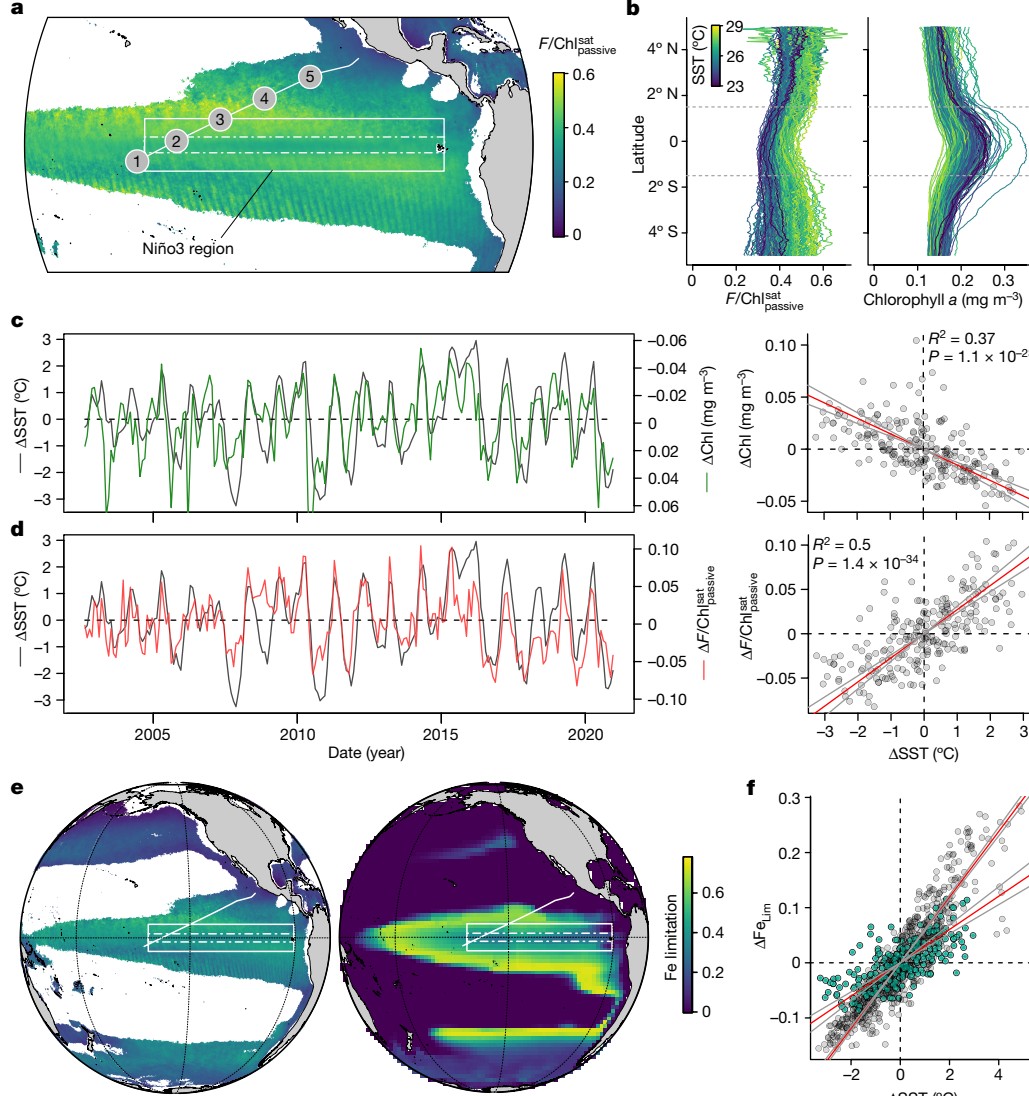

**Fig. 4 | Satellite-derived Fe limitation. a**, Regional pattern of $F/Chl_{passive}^{sat}$, reflecting the level of Fe limitation, for a MODIS-Aqua boreal winter climatological average. The white rectangle highlights the Niño3 region, with dot-dashed lines defining the core upwelling zone. Regions of the ocean with no data have chlorophyll *a* either below the satellite fluorescence detection limit or higher than our validated range. **b**, Longitudinally averaged changes in monthly $F/Chl_{passive}^{sat}$ and chlorophyll *a* in relation to mean Niño3 region SSTs for the MODIS record. Dashed lines define the latitude range for the core of the upwelling zone. **c,d**, Left: time series of chlorophyll *a* and $F/Chl_{passive}^{sat}$ anomalies (value minus record average) for the central Niño3 region. Right: scatter plot correlations showing type II linear regressions (ranged major axis; red lines) and the 95% confidence regions for the regression lines (grey lines). The $R^2$ and *P* values (two-tailed) for the regression significance test are shown (no adjustment for multiple comparisons). **e**, Pacific-wide distribution of $F/Chl_{passive}^{sat}$ (boreal winter climatological average) scaled to model limitation range (left) and model Fe limitation (right). **f**, Fe-limitation sensitivity to temperature changes in the central Niño3 region for the model (grey dots, $R^2 = 0.82$; $P = 9.4 \times 10^{-281}$; slope = 0.06) and derived from $F/Chl_{passive}^{sat}$ (green dots, $R^2 = 0.50$; $P = 4.39 \times 10^{-35}$; slope = 0.029); regression lines and statistics as for **c,d**.

properties[34] that could have potentially accompanied the nutrient limitation transition (Extended Data Figs. 3 and 4).

## Remote sensing nutrient limitation

Sunlight-stimulated chlorophyll fluorescence signals are observed at global, high-frequency scales by the Aqua Moderate Resolution Imaging Spectrometer (MODIS-Aqua). Although they offer unparalleled observational potential, these passively stimulated fluorescence signals are challenging to directly compare to the actively stimulated fluorescence previously discussed, because of known, high biophysical complexity of chlorophyll fluorescence under variable measurement conditions[14,15,18,35,36]. To circumvent this complexity, we additionally measured light naturally emanating from the surface ocean at hyperspectral resolution along the transect, using radiometers fitted to the bow of the research vessel. Fluorescence line height (the upwelling radiance peak height at the fluorescent wavelength, about 680 nm; Methods) and concentrations of chlorophyll *a* were calculated from these radiance signals, and from this, passively stimulated, shipboard $F/Chl$ was derived ($F/Chl_{passive}^{ship}$). We found that $F/Chl_{passive}^{ship}$ was irradiance dependent, increasing with excitation energy at low light levels and then saturating at >500 µmol photons m$^{-2}$ s$^{-1}$ (Fig. 3c). Such a response confirms that for the high-light conditions characterizing satellite observations of sunlight-stimulated chlorophyll fluorescence (about 500–2,500 µmol photons m$^{-2}$ s$^{-1}$) the impact of increasing excitation irradiances, which will increase energy available for fluorescence, is largely balanced out by dynamic non-photochemical quenching processes, which protect phytoplankton

from light damage but reduce fluorescence yields in the process[14,19]. These two major components of $F/\text{Chl}_{\text{passive}}$ thus seem to compensate above the >500 μmol photons m$^{-2}$ s$^{-1}$ irradiance threshold[14,15,19,37]. Subsequently, midday, light-saturated $F/\text{Chl}_{\text{passive}}^{\text{ship}}$ was 3.5-fold higher in the Fe-limited part of the transect in comparison to the low-nitrate zone, quantitatively matching the changes in $F/\text{Chl}_{\text{active}}$ (Fig. 3a,b) and therefore independently monitoring the nutrient limitation transition (Fig. 3d).

Although various forms of satellite fluorescence yields have previously been calculated, their interpretation has been a major challenge owing to limited field constraints of $F/\text{Chl}_{\text{passive}}$ in the context of a myriad of potential drivers[14–21]. As for $F/\text{Chl}_{\text{active}}$ and $F/\text{Chl}_{\text{passive}}^{\text{ship}}$, values of a relative satellite-derived sunlight-stimulated chlorophyll fluorescence yield, $F/\text{Chl}_{\text{passive}}^{\text{sat}}$ (see Methods for derivation), showed a similar approximately threefold decrease across the observed Fe-to-N limitation transition (Fig. 3d) and a consistent geographic pattern of higher values within the Fe-limited equatorial upwelling region and lower values outside (Fig. 4a and Extended Data Fig. 7). In addition to reduced fluorescence yields under low-N conditions, nutrient-replete conditions also decrease the pool of energetically uncoupled pigment protein complexes[14,29]. Consequently, whereas elevated $F/\text{Chl}_{\text{passive}}^{\text{sat}}$ was a general characteristic of the equatorial Pacific (Fig. 4a), the broader-scale pattern revealed by the satellite data was that of distinctly lower values in the core of the upwelling zone nearer the Equator, where supply rates of both N and Fe to surface waters are elevated[38] and reduced phytoplankton Fe stress has been observed[25] (Fig. 4a,b).

## Fe limitation persists over ENSO

A time series of $F/\text{Chl}_{\text{passive}}^{\text{sat}}$ spanning the two decades of MODIS-Aqua observations can reveal how ENSO regulates nutrient limitation in this region at an unprecedented scale. As expected, sea surface temperature anomalies (ΔSST) for the Niño3 region (Fig. 4a)—an indicator of both ENSO phase and the transfer rate of deeper, colder, nutrient-rich waters to the surface—was inversely correlated with chlorophyll $a$ anomalies (ΔChl; $R^2 = 0.22$; $P = 1.86 \times 10^{-13}$)[5,6]. For this region, ΔSST had an even stronger positive correlation with $\Delta F/\text{Chl}_{\text{passive}}^{\text{sat}}$ ($R^2 = 0.41, P = 1.04 \times 10^{-26}$), with warmer, low-chlorophyll $a$ excursions accompanied by increases in $\Delta F/\text{Chl}_{\text{passive}}^{\text{sat}}$ that indicate stronger Fe limitation. Focusing on a narrower latitudinal band (1.5° S–1.5° N; Fig. 4a) showed that this regional trend was largely driven by equatorial changes, where upwelling strength and its variability are strongest (ΔSST versus $\Delta F/\text{Chl}_{\text{passive}}^{\text{sat}}$; $R^2 = 0.50, P = 1.4 \times 10^{-34}$; Fig. 4c,d). The correlation of $\Delta F/\text{Chl}_{\text{passive}}^{\text{sat}}$ with chlorophyll $a$ anomalies was weaker than that with ΔSST ($R^2 = 0.38, P = 9.7 \times 10^{-25}$), implying some decoupling between Fe-limitation levels and chlorophyll $a$ biomass, presumably due to the latter being under additional grazing and/or photoacclimation control. Overall, for 2002–2021, we found that greater upwelling supplied more Fe (ref. 38), reducing Fe limitation[25] and increasing chlorophyll $a$ biomass. Notably, no evidence was found for $\Delta F/\text{Chl}_{\text{passive}}^{\text{sat}}$ declines at the highest ΔSST, which would be expected to accompany an overall switch into N-limited conditions under stronger stratification. Our analysis of satellite observations therefore further showed that Fe limitation in the Niño3 region remained robust to the ENSO extremes encountered over the >18-year MODIS observational period. In the coming decades, $\Delta F/\text{Chl}_{\text{passive}}^{\text{sat}}$ will enable observation of the nutrient-limitation response to possible stronger ENSO events comparable to the 1997/1998 El Niño[8] as well as climate change[1–7].

In addition to providing a synoptic view of ocean nutrient limitation to upscale field observations in the tropical Pacific, the parallel variations of SST and our $F/\text{Chl}_{\text{passive}}^{\text{sat}}$ metric provide an emergent constraint for nutrient limitation in ocean and climate models. When the satellite Fe-limitation diagnostic was compared with output from a model driven

by realistic ENSO dynamics (Methods), we found that spatial patterns in Fe limitation were well reproduced (Fig. 4e). However, the model demonstrated a twofold oversensitivity in the magnitude of Fe-limitation fluctuations to ENSO-driven SST changes (Fig. 4f). This suggests that the natural strength of phytoplankton Fe limitation in the equatorial Pacific is more stable in response to physical changes than models predict. This mismatch may highlight the importance of both Fe recycling[39,40] and restructuring of a diverse phytoplankton community[26,41] in modulating NPP responses to altered upwelled Fe supply rates. For example, a period of progressively lower Fe supply rates probably selects for phytoplankton with intrinsically lower Fe requirements and/or optimized Fe uptake, thus maintaining productivity to a greater extent than predicted within models lacking this flexibility. Better representation of such processes within ocean models in the future, benchmarked by synoptic observations of nutrient limitation such as those presented here, will lead to more realistic projections of ocean NPP to climate change.

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

## Methods

### Research cruise and sample collection

Observations and experiments were conducted on the RV *Sonne* (cruise SO267/2) from 28th January to 14th February 2019. Near-surface (about 2 m) seawater was collected through a towed, trace-metal-clean sampling system equipped with acid-washed tubing and a peristaltic pump. All sampling was conducted using trace-metal-clean techniques within a laboratory environment that was maintained over-pressurized with HEPA-filtered air.

### Nutrient and trace element concentration analyses

Samples for determination of dissolved inorganic macronutrients (nitrate, phosphate and silicic acid) and dissolved Fe were 0.2 μm filtered (AcroPack1000 0.8/0.2 μm filter capsule, Pall). Macronutrient samples were collected into acid-washed 15-ml polypropylene vials and stored frozen at −20 °C until analysis in a laboratory on land. Samples were thawed over 24 h in a refrigerator before determination of concentrations using an autoanalyser (QuAAtro, Seal). Samples for dissolved Fe analyses were collected into acid-washed 125-ml low-density polyethylene bottles (Nalgene) and acidified under a laminar flow hood using 150 μl concentrated hydrochloric acid (10 M; Fisher Optima grade). After >6 months, samples were analysed following pre-concentration using inductively coupled plasma mass spectrometry (Element XR), adhering to the procedure of ref. 43 except that a Preplab instrument (PS Analytical) was used for pre-concentration and standard addition was used for determination of concentrations. Analysis of GEOTRACES intercalibration standard GSP91 in the same analytical run yielded concentrations matching previous determinations (Fe mean ± standard deviation = 0.166 ± 0.048, $n = 6$; https://www.geotraces.org/standards-and-reference-materials/)[44,45].

### Chlorophyll *a* and phytoplankton community structure

Samples for chlorophyll *a* analysis (100 ml) were filtered onto glass-fibre filters (Macherey-Nagel), extracted in 90% HPLC-grade acetone and measured on a calibrated fluorometer (Trilogy, Turner Designs) following the method of ref. 46. Samples for diagnostic phytoplankton pigments (2–3.5 l; experimental samples from pooled treatment replicates) were filtered onto glass-fibre filters (Macherey-Nagel) and stored frozen at −80 °C. Extraction, analysis and peak picking (Chromeleon, Thermo Fisher Scientific) followed the procedure described in ref. 26. Diagnostic phytoplankton pigments were converted to estimated contributions of different phytoplankton types using CHEMTAX[47], using starting pigment ratios from ref. 48. Samples for flow cytometry analysis (2 ml) were fixed with paraformaldehyde (1% final concentration) and stored at −80 °C before analysis using a FACSort flow cytometer (Beckton-Dickinson) and gated and enumerated using CellQuest Pro software (Beckton-Dickinson) following procedures described in ref. 49 (see Supplementary Fig. 1 for an example of the gating strategy).

### Underway active fluorometry and absorption measurements

A fast-repetition-rate fluorometer (Fast Ocean sensor interfaced with FastPro8, Chelsea Technologies) was connected to the ship's underway, near-surface seawater flow-through supply for determination of diel variability in $F_v/F_m = (F_m - F_o)/F_m$ (where $F_o$ indicates minimum fluorescence). Before calculation of $F_v/F_m$, the fluorescence values $F_o$ and $F_m$ were first corrected for blank fluorescence (fluorescence of 0.2-μm-filtered seawater)[50]. Lines shown in Fig. 2a,b are 17-min rolling means of $F_v/F_m$ and $\sigma_{PSII}$. Dots in Fig. 2e–g are 1-min binned means. The same fast-repetition-rate fluorometer was switched to bench-top mode for discrete sample analysis of the experimental samples. A separate fluorometer (Sea-Bird Eco FLNTU) housed with a self-cleaning monitoring box (SMB; -4H-Jena engineering GmbH, Jena) was additionally used to record in vivo chlorophyll *a* fluorescence continuously from the ship's underway flow-through seawater supply (shown in Fig. 3b).

The absorption properties of discrete seawater samples were determined using a point-source integrating-cavity absorption meter. Samples were collected from both the towed trace-metal-clean sampling system and underway flow-through system. The measurement procedure was identical to that described in ref. 51 except that the calibration was made using a solid standard instead of the nigrosin solution[52]. The total absorption of sampled water was measured for unfiltered and filtered (0.2 μm) samples. The values for spectrally resolved (400–710 nm) absorption from filtered samples was subtracted from those for unfiltered samples to yield particulate absorption. Line height absorption was calculated and converted to approximate chlorophyll *a* concentration following ref. 53, which was then used to normalize particulate absorption spectra (Extended Data Fig. 7).

### Metaproteomics

**Sample collection.** Samples for microbial metaproteomics were collected through direct filtration of 100–120 l of seawater diverted from the trace-metal-clean sampling system. Sample collection was always at local night-time and directly after collection of bioassay experimental seawater. Seawater was pumped through two in-line 142-mm filters (3 μm Versapor and 0.2 μm Supor 200 housed in Sartorius filtration holders). Following filtration, both filters were immediately folded, placed in separate 10-ml cryovials, soaked in RNAlater (Sigma-Aldrich) and stored at −80 °C.

**Protein extraction.** Analyses followed methods described in ref. 54. Proteins were extracted using a modified magnetic bead method from ref. 55. Filter sections were placed in 15 ml of protein extraction buffer (50 mM HEPES pH 8.5, 1% SDS in HPLC-grade water). Samples were heated at 95 °C for 10 min and shaken at room temperature for 30 min. Filters were removed and protein extracts were filtered through 5.0 μm Millex low-protein-binding filters. Samples were then centrifuged; supernatant was removed from the pellet and 6–7.5 ml of each sample was transferred to a Vivaspin 5K MWCO ultrafiltration unit. Protein extract was concentrated to approximately 350 μl, washed with 1 ml of lysis buffer and transferred to a 2 ml ethanol-washed microtube. Vivaspins were rinsed with small volumes of protein extraction buffer to remove all concentrated protein, and all samples were brought up to 430 μl with extraction buffer. A 30 μl volume was set aside for total protein quantification and DNA analysis.

**Protein quantification.** Standard curves were generated using albumin standard (Thermo Scientific). Total protein was quantified after extraction and after purification with 2 μl of sample in duplicate using the bicinchoninic acid method (Thermo Scientific Micro BCA Protein Assay Kit). Absorbance was measured on a Nanodrop ND-1000 spectrophotometer (Thermo Scientific).

**Protein reduction and alkylation.** A quantity of 50 units (2 μl) of benzonase nuclease (Novagen) was added to each sample and incubated at 37 °C for 30 min. Samples were reduced by adding 20 μl of 200 mM dithiothreitol (Fisher Scientific) in 50 mM HEPES pH 8.5 at 45 °C for 30 min. Samples were alkylated by adding 40 μl of 400 mM iodoacetamide (Acros) in HEPES pH 8.5 for 30 min at 24 °C, occasionally heating to 37 °C to prevent precipitation. The reaction was quenched by adding 40 μl of 200 mM dithiothreitol in 50 mM HEPES pH 8.5.

**Protein purification and digestion.** SpeedBead magnetic carboxylate-modified particles (GE Healthcare) were prepared according to ref. 55. A 20 μl volume (20 μg μl⁻¹) of magnetic beads was added to 400 μl of extracted protein sample. Samples were heated at 37 °C periodically to avoid precipitation. Samples were acidified to pH 2–3 by adding 50 μl of 10% formic acid. Twice the volume (1,100 μl) of acetonitrile was immediately added. Samples were incubated at 37 °C for 15 min and then at room temperature for 30 min. Samples

were placed on a magnetic rack, incubated for 2 min, and supernatant was removed and discarded. Samples were washed two times removing and discarding supernatants with 1,400 µl of 70% ethanol for 30 s on the magnetic rack. A 1,400 µl volume of acetonitrile was added to each sample for 30 s on the magnetic rack. Supernatant was removed and discarded. Samples were air dried for approximately 4 min until acetonitrile had just evaporated. Samples were removed from the magnetic rack, and beads were reconstituted in 90 µl of 50 mM HEPES pH 8.0. Purified protein was quantified as described above. Trypsin (Promega) dissolved in HEPES pH 8.0 at a concentration of 0.5 µg µl$^{-1}$ was added to samples at a 1:25 trypsin-to-protein ratio and incubated at 37 °C overnight.

**Peptide recovery and preparation.** Acetonitrile was added to digested peptides at a concentration of ≥95% and incubated for 20 min at room temperature. Samples were then placed on the magnetic rack for 2 min and supernatant was removed and discarded. A 1,400 µl volume of acetonitrile was added to samples on the magnetic rack for 15 s. Supernatant was removed and discarded. Samples were air dried for approximately 4 min, just until acetonitrile was evaporated. Beads were reconstituted in 90 µl of 2% dimethylsulfoxide and incubated off the rack at room temperature for ≥15 min. Samples were centrifuged slowly and briefly at a relative centrifugal force of 900 to remove liquid from the tube walls. Samples were incubated on the magnetic rack for 15 min and supernatant containing peptides was transferred to a new ethanol-washed 1.5-ml microtube. This step was repeated to ensure removal of all magnetic beads. Then, 1% trifluoroacetic acid was added to samples for a final concentration of 0.1%. Samples were zip tipped with Pierce C18 tips (Fisher) according to the manufacturer's protocol with a final resuspension in 25 µl of 70% acetonitrile, 0.1% formic acid. Samples were evaporated to approximately 10 µl in a DNA110 Speedvac (ThermoSavant). Samples with lower protein concentrations were further evaporated to minimize acetonitrile percentage in final resuspension–zip tip product to be less than 30% of total final buffer B volume. Samples were finally resuspended to a peptide concentration of 1 µg µl$^{-1}$ in buffer B (2% acetonitrile, 0.1% formic acid).

**Mass spectrometry.** Metaproteomic samples were analysed by liquid chromatography–mass spectrometry (Michrom Advance HPLC coupled to a Thermo Scientific Fusion Orbitrap mass spectrometer with a Thermo Flex source). A 1 µg quantity of each sample (measured before trypsin digestion) was concentrated onto a trap column (C18 Reprosil-Gold, Dr. Maisch GmbH) and rinsed with 100 µl of 0.1% formic acid, 2% acetonitrile, 97.9% water before gradient elution through a reverse-phase C18 column (C18 Reprosil-Gold, Dr. Maisch GmbH) at a flow rate of 500 nl min$^{-1}$. The chromatography consisted of a nonlinear 100-min gradient from 2% to 95% buffer B, with buffer A being 0.1% formic acid in water and buffer B being 0.1% formic acid in acetonitrile (all solvents were Fisher Optima grade). The mass spectrometer was set to carry out mass spectrometry scans on the Orbitrap (240,000 resolution at 200 $m/z$) with a scan range of 380 $m/z$ to 1,280 $m/z$. Tandem mass spectrometry was carried out on the ion trap using data-dependent settings (top speed, dynamic exclusion 10 s, excluding unassigned and singly charged ions, precursor mass tolerance of ±3 ppm, with a maximum injection time of 150 ms).

**Metaproteomic data analysis.** The raw mass spectra files were searched using SEQUEST HT within Thermo Proteome Discoverer 2.2 software using a parent ion tolerance of 10 ppm and a fragment tolerance of 0.6 Da and allowing up to 1 missed cleavage. Oxidation and acetyl dynamic modifications and carbamidomethyl static modifications were included. Percolator peptide spectral matching was used within Proteome Discoverer with a maximum Delta Cn of 0.05 and a decoy search validation based on posterior error probabilities. Processed files were then loaded into Scaffold 5.0 (Proteome Software Inc.)

using prefiltered mode with a protein threshold of 1.0% false discovery rate, a peptide threshold of 0.1% false discovery rate and a minimum of 1 peptide for analysis.

The *Prochlorococcus* nutrient stress and photosystem proteins shown in Figs. 1d and 2d were extracted from the resultant annotated metaproteomics dataset; details are provided in Supplementary Table 1. In cases for which there was more than one detection of the same protein function attributed to *Prochlorococcus* (same annotation) in the total exclusive spectral counts (probably reflecting mapping of measured tryptic peptides to distinct metagenome contigs with sequence variants), the sequence with the highest counts was used. Only proteins with ≥3 exclusive spectral counts for at least one of the sites were used. A subsequent peptide-level analysis was conducted using METATRYP V2.0 (https://metatryp.whoi.edu; ref. 56), and demonstrated that the peptides found in each protein were largely unique to the *Prochlorococcus* taxonomic classification (Supplementary Table 1). The data reported here are in unnormalized total exclusive spectral counts, which as described in ref. 57, can be useful in avoiding biases associated with changes in biological diversity across gradients (note that a subsequent normalization of the data for visualization in Figs. 1d and 2d was applied to scale variability in protein abundances across all sites to between 0 and 1). Normalizations according to total *Prochlorococcus* exclusive spectral counts did not alter the interpretation of these large metaproteome signals. In addition to total exclusive spectral counts, total non-exclusive spectral counts for each of the proteins were inspected to assess for sequence variations across the transect, which could map onto a different annotation despite having the same function (Supplementary Table 2). This analysis showed that total spectral count trends for the highest count proteins showed either much lower values or largely the same trends as the exclusive spectral count trends. To further investigate whether the low abundance or absence of photosystem I and cytochrome $b_6f$ proteins found in the high-nitrate, Fe-limited region (sites 1–3) was due to strain-level differences in sequences that could have restricted their detection relative to low-nitrate strains, we carried out a further peptide analysis using METATRYP V2.0 (https://metatryp.whoi.edu; ref. 56) to assess for *Prochlorococcus* strain matches for peptide sequences (Supplementary Table 3). This analysis showed that all peptides matched to *Prochlorococcus* strains isolated in the equatorial Pacific, in addition to strains isolated in other regions.

### Nutrient-addition bioassay experiments

Bioassay experiments conducted at sites 2–5 directly followed previously published protocols[26]. Acid-washed polycarbonate 1-l bottles (Nalgene) were filled with trace-metal-clean seawater from the sampling system described previously. Initial bottles were used to characterize initial conditions, control bottles were incubated without treatment, and treatment bottles were spiked with full factorial combinations of the nutrients N, Fe and Co. Supplementary N+Fe+Zn and N+Fe+Si treatments were also conducted to assess for potential Zn or Si serial limitation or co-limitation alongside N and Fe. All controls and treatments were conducted in triplicate. The N spike was a combined treatment of nitrate + ammonium (final amended concentration of 2 µM nitrate + 1 µM ammonium). Subsamples from incubations used for determination of macronutrient concentrations indicated that treatments with supplied nitrate were not depleted to <1 µM nitrate in any experiment over the 48-h incubation duration. Fe, Co and Zn were each added to a final added concentration of 2 nM. Silicic acid was added to a final amended concentration of 2 µM. Macronutrient spikes were previously passed through a column of prepared cation exchange resin to remove contaminating trace elements (Chelex-100, BioRad). Trace element spikes were prepared from 99+% purity solid standards dissolved in 0.01 M hydrochloric acid (Fisher Optima grade diluted in Milli-Q deionized water). Treated and control bottles were Parafilm-sealed, placed in a clear sample bag, and incubated in on-deck incubators that were continuously flushed

with near-surface seawater and screened with blue screening (Blue Lagoon, Lee filters) for about 48 h. Following incubation, experiments were ended and each individual replicate was subsampled for photophysiology (fast-repetition-rate fluorometer), chlorophyll *a*, flow cytometry and macronutrients. The remaining samples from each triplicate replicate of a treatment were combined and filtered for HPLC pigment analysis. Calculation of chlorophyll *a*-based net growth rates ($\mu_{Chl}$) in Fig. 1c used the equation ln($Chl_{Treatment}$/$Chl_{Control}$)/$t$, in which $t$ is the duration of the experiment (2 days). The control rather than initial chlorophyll concentration was used for this calculation owing to the consistent chlorophyll reductions between initial and controls that was assumed to be a result of photoacclimation to a higher-light environment in the deck incubators over in situ conditions (Extended Data Fig. 2).

## Shipborne radiometric quantities

Two sets of hyperspectral radiometer systems were used to collect radiometric quantities on the research cruise. Each set consisted of a TriOS RAMSES-ACC hyperspectral cosine irradiance radiometer that measured incoming solar irradiance $E_D(\lambda)$ and two TriOS RAMSES-ARC hyperspectral radiance meters. The radiance radiometers measured bulk ocean surface-leaving radiance $L_{sfc}(\theta_{sfc}, \Phi, \lambda)$ and sky-leaving radiance $L_{sky}(\theta_{sky}, \Phi, \lambda)$. A custom-made frame held the radiance radiometers at fixed viewing angles at the bow of RV *Sonne*. The zenith viewing angle of the sky-facing radiometers was $\theta_{sky} = 45°$, and the nadir viewing angle of the sea-surface-facing radiometer was $\theta_{sfc} = 45°$. Each set was separated by an azimuthal angle of $\Phi = 45°$ from the ship's heading. Irradiance meters were attached to the railing at the top of the mast at least 1 m apart pointing directly upwards. Hyperspectral radiometric quantities were recorded at 5-min intervals from the ultraviolet (320 nm) to near-infrared (950 nm) regions. A correction for sea-surface-reflected glint was applied to derive water-leaving radiance ($L_W$) and remote-sensing reflectance ($R_{RS}$) as proposed in previous studies[58,59]. Fluorescence line height (FLH) values were calculated from $L_W$ using the approximate central wavelengths of the relevant MODIS-Aqua wavebands (667, 678 and 748 nm):

$$FLH = L_{678} - L_{667} - (L_{748} - L_{667}) \times (678 - 667)/(748 - 667).$$

Chlorophyll *a* (Chl) concentrations were derived from $R_{RS}$ using the OC3 algorithm[60]. Although variability in chlorophyll *a* concentrations was relatively minor throughout the cruise track, comparing shipboard radiometry-derived OC3 and filtered and extracted chlorophyll *a* concentrations collected at similar time points yielded a good relationship between both methods ($R^2 = 0.57$, $P = 2.8 \times 10^{-7}$, $n = 33$; Supplementary Fig. 2). FLH values were normalized to chlorophyll *a* values to generate $F/Chl^{ship}_{passive}$. Dots in Fig. 3c represent 25-min rolling averages, and in Fig. 3d, data were extracted for 12:00–15:00 local time for each day to approximately match MODIS-Aqua overpass timing (solid line indicates the mean and dots represent individual observations).

## Satellite remote sensing

MODIS-Aqua level 3 normalized FLH (nFLH), chlorophyll *a* concentration (OCX algorithm) and SST data products were downloaded at about 9 km (1/12 of a degree) spatial resolution as standard mapped images from the NASA (National Aeronautics and Space Administration) Ocean Colour website (https://oceancolor.gsfc.nasa.gov). The nFLH product is calculated using normalized water-leaving radiances, with the normalization accounting for the viewing geometry of the ocean surface with respect to incident solar radiation[14,21,61]. Passive chlorophyll *a* fluorescence signals were observed to be at fully stimulated plateaux at the incident irradiances at the time of the satellite overpass (>500 μmol photons m$^{-2}$ s$^{-1}$; Fig. 3c). Therefore, we followed ref. 21 and carried out a first-order correction of nFLH values to remove the normalization, by multiplying nFLH values by the cosine of the

solar zenith angle calculated at each pixel latitude for the mid-time point of the respective composite L3 image[21]. This conversion led to little change in trends of chlorophyll-normalized fluorescence values in the low-latitude equatorial Pacific cruise region focused on here (however, it completely changed distributions and seasonal trends at higher latitudes; Extended Data Fig. 8). It is important to note that the generality of our FLH versus irradiance observations made in the low-latitude Pacific remains to be tested in other regions, such that the latter satellite observations should be treated with caution (for example, any regions away from the tropical Pacific in regions within Fig. 4e and Extended Data Fig. 8). We also note that a conversion of nFLH back to FLH is essentially equivalent to multiplying the nFLH values by the instantaneous PAR (iPAR) at the time of image capture, as carried out in ref. 14 to correct for the predicted (now verified here for the tropical Pacific) impact of non-photochemical quenching on nFLH[14].

Satellite observations of FLH were divided by satellite-derived chlorophyll *a* concentrations to retrieve $F/Chl^{sat}_{passive}$. We also excluded pixels with chlorophyll *a* concentrations <0.1 or >0.4 mg m$^{-3}$, which bounds the estimated lower FLH detection limit[18,62,63] as well as more uncertain chlorophyll *a* concentration retrievals[64] and upper levels where the relationship between chlorophyll *a* and phytoplankton light absorption become increasingly nonlinear[65,66]. Within this chlorophyll *a* range, the relationship between chlorophyll *a* concentrations and phytoplankton light absorption is indistinguishable from linear[66]; therefore, an attempt to convert chlorophyll *a* to absorption is not necessary in this case (and could possibly introduce error due to the impact of high chlorophyll *a* values on chlorophyll *a* versus absorption equations developed using datasets from the global ocean[66]). For the time series analyses of equatorial Pacific $F/Chl^{sat}_{passive}$ (that is, Fig. 4c,d,f), we used the Niño3 box where $F/Chl^{sat}_{passive}$ was validated, and where chlorophyll *a* consistently falls within the applied chlorophyll *a* thresholds (in >90% images <5% pixels fall outside this range; including these pixels does not change trends). In Fig. 3d, $F/Chl^{sat}_{passive}$ pixels were averaged over 100-km-radii circles around each ship track sampling point, to reduce noise in signals derived from individual pixels[63]. To generate a $F/Chl^{sat}_{passive}$ range comparable to the Fe limitation term in biogeochemical ocean models in Fig. 4e,f, the range in $F/Chl^{sat}_{passive}$ was scaled to range between 0 (no Fe limitation) and 0.78 (maximum Fe limitation predicted by the used model; see next section). The upper value of $F/Chl^{sat}_{passive}$ used to scale to the model limitation range was taken as the $F/Chl^{sat}_{passive}$ threshold where the fraction of pixels above this value from a global-scale monthly composite reached an asymptote ($F/Chl^{sat}_{passive} = 0.7$ W m$^{-2}$ sr$^{-1}$ μm$^{-1}$ [mg Chl m$^{-3}$]$^{-1}$; varying this threshold by 10% had <10% impact on the twofold difference in slopes between the model and observations; Fig. 4f).

## Global biogeochemical ocean model

The Fe limitation term ($L_{Fe}$) from a state-of-the-art global biogeochemical model NEMO-PISCES[67] was compared with spatial patterns and seasonality in $F/Chl^{sat}_{passive}$. PISCES has a relatively complex consideration of the ocean Fe cycle, including the dynamics of phytoplankton Fe uptake, storage and limitation, as well as the parallel processes of zooplankton recycling and scavenging[3,68,69]. The $L_F$ term is the fractional limitation of phytoplankton maximum growth rate (range 0–1), calculated from dynamic phytoplankton Fe quotas relative to emergent minimum and optimum quotas[70]. The model limitation terms were extracted from the surface layer at monthly resolution for diatoms and nanophytoplankton (the phytoplankton groups in PISCES). Weighting these terms by the carbon biomass of each phytoplankton type generated an average $L_{Fe}$. In Fig. 4e,f and Extended Data Fig. 8 this average $L_{Fe}$ was subtracted from 1 to generate an Fe limitation term ($Fe_{Lim} = 1 - L_{Fe}$), so that higher values indicate greater Fe limitation (and vice versa). The model results were drawn from an online ocean model simulation forced by atmospheric reanalysis product JRA55 from 1958 to 2019

under the OMIP2 protocol. The model is forced at a 6-h time frequency over three cycles between 1648 and 2019[71].

## Statistics

Statistics and calculations were conducted using R. Type II linear regressions (ranged major axis) in Fig. 4c,d,f were conducted using the lmodel2 package.

## Reporting summary

Further information on research design is available in the Nature Portfolio Reporting Summary linked to this article.

## Data availability

Biogeochemical data have been deposited in Zenodo (https://doi.org/10.5281/zenodo.8059552)[72]. Metaproteomics data are available through the Proteomics Identifications database (accession number PXD030610) and ProteomeXchange (accession number PXD030610). Hyperspectral radiometry data are available through Pangaea (https://doi.org/10.1594/PANGAEA.924038)[73]. The MODIS satellite data are available from the NASA Ocean Colour website (https://oceancolor.gsfc.nasa.gov) with the data product names 'Fluorescence Line Height (normalized)', 'Chlorophyll concentration' and 'Sea Surface Temperature'.

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

**Acknowledgements** We thank the captain, officers and crew of the RV *Sonne* SO276/2 cruise. K. Nachtigall, T. Klüver, T. Steffens, A. Mutzberg, D. Jasinski, C. Beckmann, H. Stuhr, P. Buchanan, S. Kinne, B. Tietjen, J. Wollschläger and R. Henkel are thanked for technical and logistical assistance. The research was financially supported by the German Ministry of Education and Research (BMBF) grants 'EqPac co-limitation' (03G0267TA) to T.J.B. and E.P.A. and 'OceanLight' (03G0267TB) to D.V. and O.Z. A.T. acknowledges funding from European Research Council grant agreement no. 724289. S.P.G. was financially supported by Deutsche Forschungsgemeinschaft (grant 417276871).

**Author contributions** T.J.B. designed the overall project. T.J.B. and X.W. conducted the seawater sampling, shipboard active fluorescence measurements and bioassay experiments. T.J.B. conducted the trace element analysis. E.P.A. oversaw the nutrient and trace element analysis. A.E. oversaw the flow cytometry analysis. S.P.G., D.V. and O.Z. conducted the shipboard radiometry measurements. D.V. conducted the phytoplankton light absorption measurements. D.M., M.R.M. and M.A.S. conducted the metaproteomics analysis. A.T. conducted the ocean biogeochemical model simulations. T.J.B. carried out the data analysis and wrote the first draft of the paper. T.J.B., A.T., C.M.M., M.A.S. and S.P.G. worked on subsequent drafts. All authors provided comments on the manuscript.

**Funding Open** access funding provided by GEOMAR Helmholtz-Zentrum für Ozeanforschung Kiel.

**Competing interests** The authors declare no competing interests.

**Additional information**
**Correspondence and requests for materials** should be addressed to Thomas J. Browning.

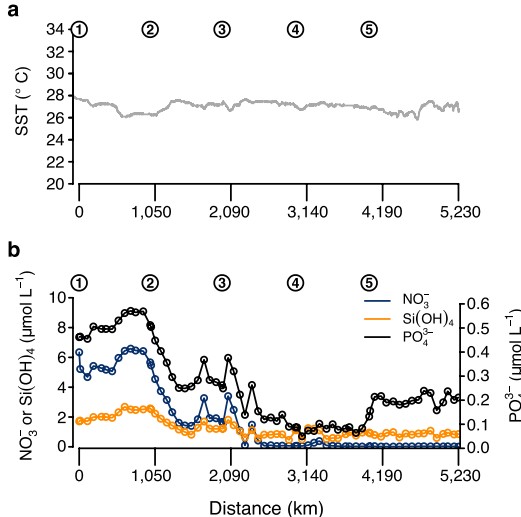

**Extended Data Fig. 1 | Temperature and macronutrient trends along the cruise transect. a**, Sea surface temperature (SST). **b**, Surface macronutrient concentrations. The phosphate scale in **b** is the nitrate/silicate scale multiplied by 1/16, the theoretical P:N requirement of phytoplankton. Note that in all cases the phosphate line lies above the nitrate line, therefore indicating conditions of excess phosphate relative to nitrate throughout this region.

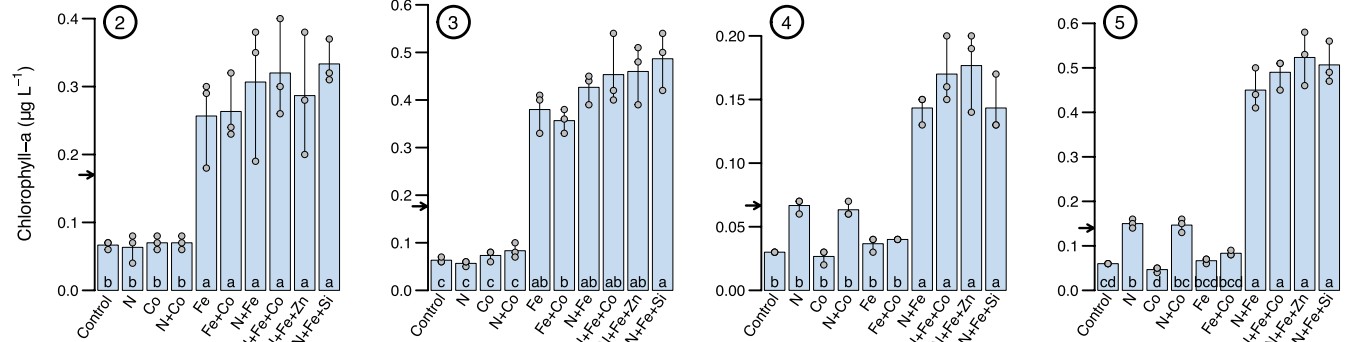

**Extended Data Fig. 2 | Chlorophyll-a responses in nutrient addition bioassay experiments.** Bar heights represent mean responses across three biological replicates. Dots show individual replicates and error bars show the range. Letters indicate the results of a statistical test, where bars labelled with the same letter have statistically indistinguishable means (one way ANOVA, α = 0.05, followed by Tukey posthoc test). Arrows indicate initial concentrations. Circled numbers are experiment sampling sites.

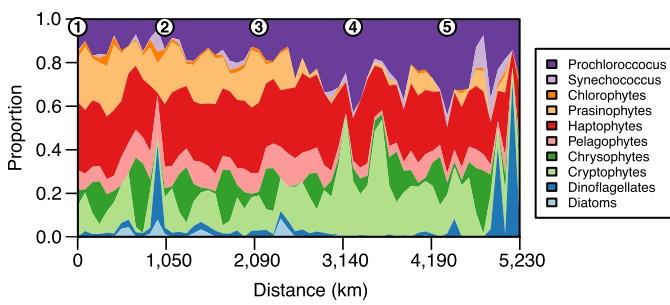

**Extended Data Fig. 3 | Phytoplankton community composition along the cruise transect.** Values reflect the proportion contribution of the phytoplankton types to total chlorophyll-a, as determined via diagnostic pigment analyses followed by CHEMTAX processing (see Methods). Circled numbers are metaproteomics/bioassay experiment sampling sites. Note that recent genomic and proteomic analyses in this region suggest that the haptophyte contribution indicated by the pigment analysis and CHEMTAX could be overestimated, with larger contributions from dinoflagellates instead[74].

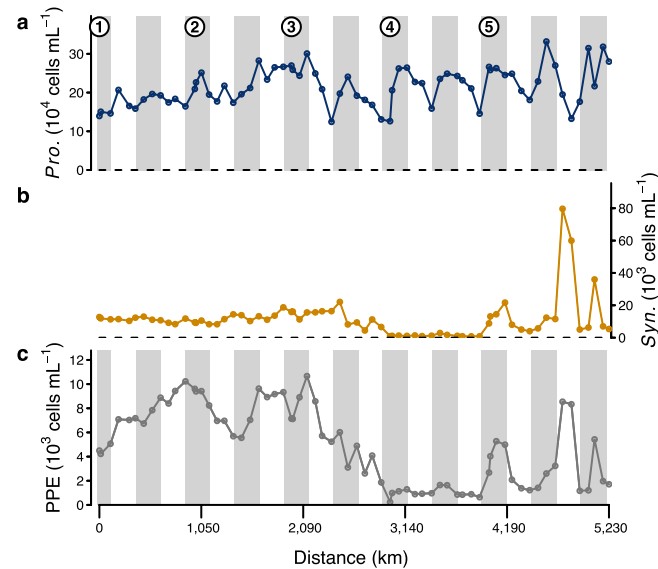

**Extended Data Fig. 4 | Picophytoplankton cell concentrations along the cruise transect. a**, *Pro.* is *Prochlorococcus*; **b**, *Syn.* is *Synechococcus*; **c**, PPE is photosynthetic picoeukaryotes. Note the order of magnitude higher axis scaling for *Prochlorococcus* in **a**. Grey shading represents local night-time. Circled numbers are metaproteomics/bioassay experiment sampling sites.

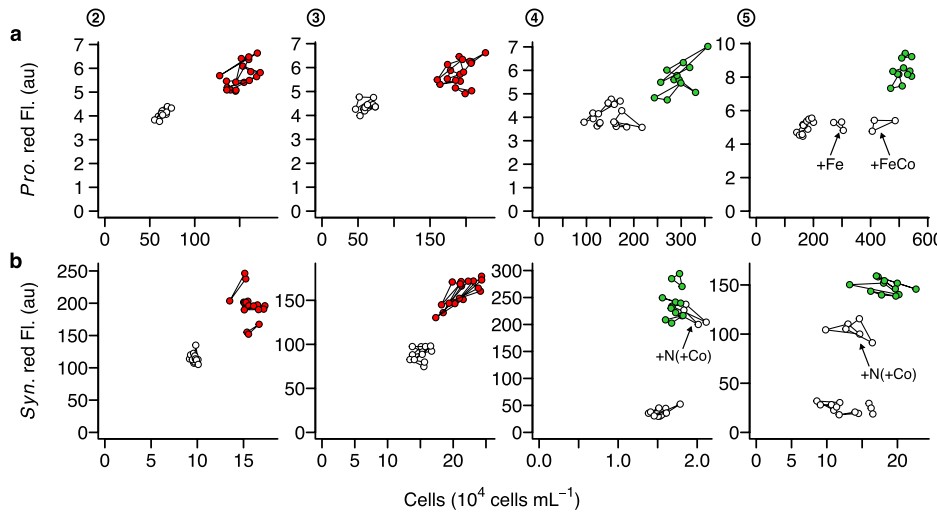

**Extended Data Fig. 5 | Cyanobacteria response to nutrient enrichment.**
**a**, *Prochlorococcus* (Pro. as in Fig. 1f). **b**, *Synechococcus (Syn.)*. At Fe limited Sites 2 and 3, any treatment with added Fe is highlighted in red; at N-Fe co-/serially limited Sites 4 and 5, any treatment with added N+Fe is highlighted in green. Triplicate biologically independent replicates for each treatment are shown as individual points, with the same treatment joined with lines.

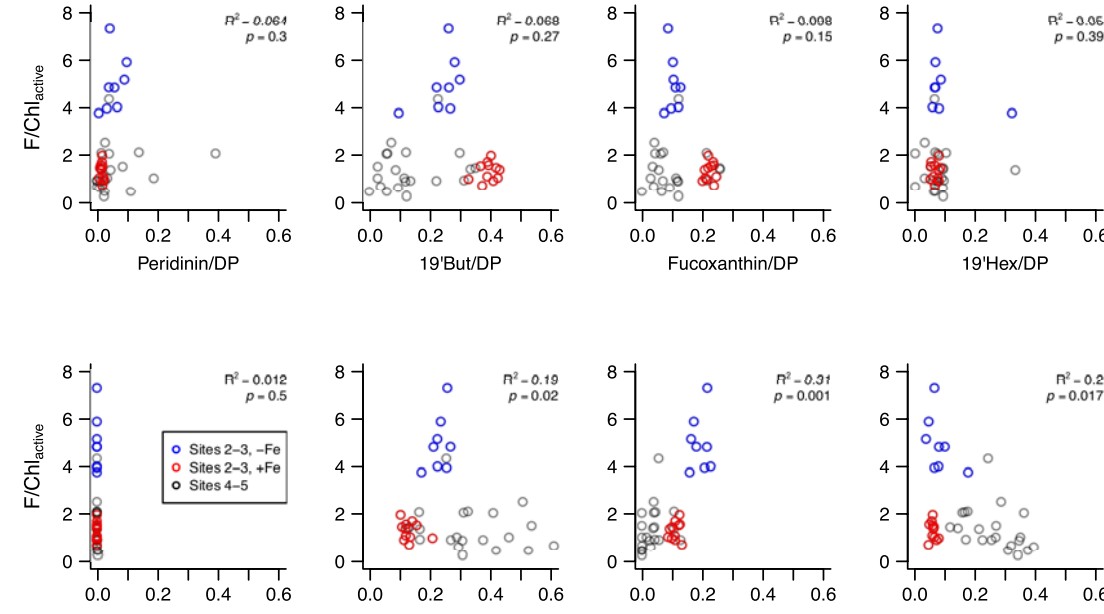

**Extended Data Fig. 6 | Restricted changes in $F/Chl_{active}$ in relation to diagnostic pigments.** For Fe limited sites 2–3, $F/Chl_{active}$ was reduced following Fe supply to the approximate levels observed at N limited Sites 4–5. In contrast, relationships with the ratio of diagnostic pigment to total diagnostic pigment are less clear, suggesting pigmentation/community structure is not the major driver of $F/Chl_{active}$. 19'But = 19'-butanoyloxyfucoxanthin; 19'Hex = 19'-hexanoyloxyfucoxanthin; Allox = alloxanthin; Chl-b = chlorophyll-b; DV-Chl-a = divinyl chlorophyll-a. The $R^2$ and p-values (2-tailed) for the linear model regression significance test are shown (no adjustment for multiple comparisons).

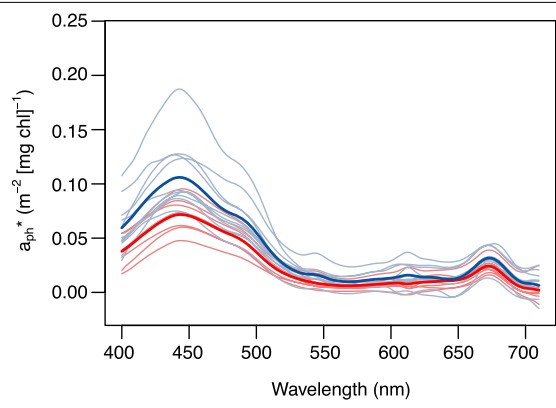

**Extended Data Fig. 7 | Spectrally resolved phytoplankton specific light absorption coefficients.** Red and blue lines indicate near surface samples from the Fe limited (defined as south of Site 3) and N-Fe co-limited zone (defined as north of Site 4) respectively. Bold lines are averages for each zone.

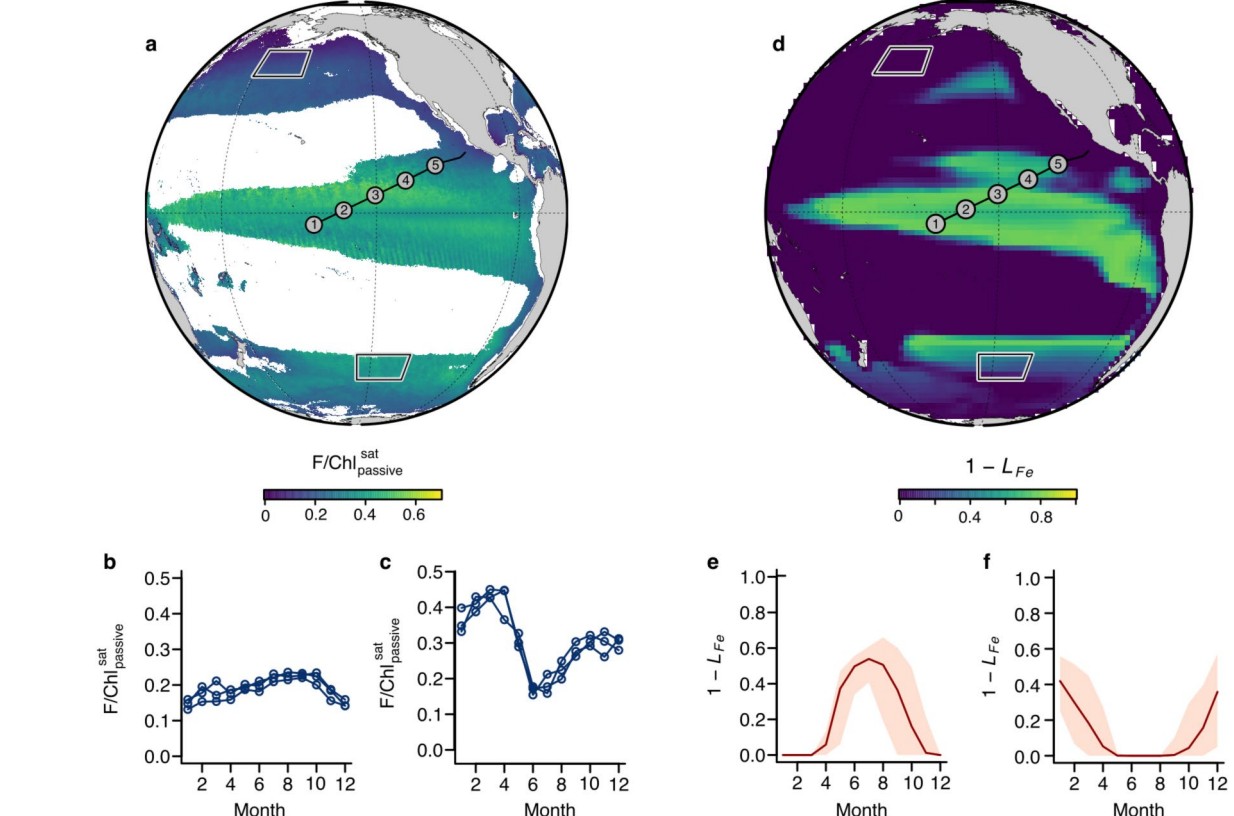

**Extended Data Fig. 8 | Elevated $F/Chl_{passive}^{sat}$ for regions and seasons of expected Fe limitation. a**–**c**, $F/Chl_{passive}^{sat}$. **d**–**f**, Model Fe limitation factor (0=no Fe limitation; 1=strong Fe limitation). Monthly timeseries are for North Pacific (**b**, **e**) and Southern Ocean (**c**, **f**) regions indicated in the maps in **a** and **d**. For $F/Chl_{passive}^{sat}$ these are for three example years (2015–2017); for the model these are the mean (central red line) and range (shading). Although these regions outside of the tropical Pacific study area should be interpreted with caution, $F/Chl_{passive}^{sat}$ indicates transitions into stronger Fe stress in summer, as light limitation is reduced and winter-entrained Fe stocks are drawdown, which is consistent with field observations[75–77].

# Reporting Summary

## Statistics

For all statistical analyses, confirm that the following items are present in the figure legend, table legend, main text, or Methods section.

| n/a | Confirmed | |
|---|---|---|
| ☐ | ☒ | The exact sample size (*n*) for each experimental group/condition, given as a discrete number and unit of measurement |
| ☐ | ☒ | A statement on whether measurements were taken from distinct samples or whether the same sample was measured repeatedly |
| ☐ | ☒ | The statistical test(s) used AND whether they are one- or two-sided<br>*Only common tests should be described solely by name; describe more complex techniques in the Methods section.* |
| ☒ | ☐ | A description of all covariates tested |
| ☒ | ☐ | A description of any assumptions or corrections, such as tests of normality and adjustment for multiple comparisons |
| ☐ | ☒ | A full description of the statistical parameters including central tendency (e.g. means) or other basic estimates (e.g. regression coefficient) AND variation (e.g. standard deviation) or associated estimates of uncertainty (e.g. confidence intervals) |
| ☐ | ☒ | For null hypothesis testing, the test statistic (e.g. *F*, *t*, *r*) with confidence intervals, effect sizes, degrees of freedom and *P* value noted<br>*Give P values as exact values whenever suitable.* |
| ☒ | ☐ | For Bayesian analysis, information on the choice of priors and Markov chain Monte Carlo settings |
| ☒ | ☐ | For hierarchical and complex designs, identification of the appropriate level for tests and full reporting of outcomes |
| ☒ | ☐ | Estimates of effect sizes (e.g. Cohen's *d*, Pearson's *r*), indicating how they were calculated |

*Our web collection on statistics for biologists contains articles on many of the points above.*

## Software and code

Policy information about availability of computer code

| Data collection | Flow cytometry: CellQuest Pro v5 software (Becton Dickenson).<br>High performance liquid chromatography: Chromeleon v7.0 (Thermo Fisher Scientific).<br>Fast repetition rate florometry: FastPro8 (Chelsea Technologies).<br>ICP-MS: ELEMENT 2/XR software (Thermo Scientific)<br>Metaproteomics: Thermo Proteome Discoverer v2.2 software, Scaffold v5.0 (Proteome Software Inc.), METATRYP v2.0<br>Statistics, calculations, figures: R v4.1.0 |
|---|---|
| Data analysis | Statistics and other calculations were conducted using R v4.1.0. |

For manuscripts utilizing custom algorithms or software that are central to the research but not yet described in published literature, software must be made available to editors and reviewers. We strongly encourage code deposition in a community repository (e.g. GitHub). See the Nature Portfolio guidelines for submitting code & software for further information.

## Data

Policy information about availability of data

All manuscripts must include a data availability statement. This statement should provide the following information, where applicable:
- Accession codes, unique identifiers, or web links for publicly available datasets
- A description of any restrictions on data availability
- For clinical datasets or third party data, please ensure that the statement adheres to our policy

Biogeochemical data are deposited in Zenodo (https://doi.org/10.5281/zenodo.8059552). Metaproteomics data are available via the PRoteomics IDEntifications (PRIDE) database (accession number: PXD030610) and ProteomeXchange (accession number: PXD030610). Hyperspectral radiometry data are available via Pangaea (https://doi.org/10.1594/PANGAEA.924038). The MODIS satellite data are available from the NASA Ocean Colour website (https://oceancolor.gsfc.nasa.gov) with the data product names 'Fluorescence Line Height (normalized), 'Chlorophyll concentration', and 'Sea Surface Temperature'.

## Research involving human participants, their data, or biological material

Policy information about studies with human participants or human data. See also policy information about sex, gender (identity/presentation), and sexual orientation and race, ethnicity and racism.

| | |
|---|---|
| Reporting on sex and gender | NA |
| Reporting on race, ethnicity, or other socially relevant groupings | NA |
| Population characteristics | NA |
| Recruitment | NA |
| Ethics oversight | NA |

Note that full information on the approval of the study protocol must also be provided in the manuscript.

# Field-specific reporting

Please select the one below that is the best fit for your research. If you are not sure, read the appropriate sections before making your selection.

☐ Life sciences   ☐ Behavioural & social sciences   ☒ Ecological, evolutionary & environmental sciences

For a reference copy of the document with all sections, see nature.com/documents/nr-reporting-summary-flat.pdf

# Ecological, evolutionary & environmental sciences study design

All studies must disclose on these points even when the disclosure is negative.

| | |
|---|---|
| Study description | Field sampling and experiments were conducted onboard the RV Sonne in January/February 2019 (SO267/2). Experimental: Seawater was collected under trace-metal-clean conditions using a towed water sampling device (~2 m depth) and filled in 1 L acid-washed polycarbonate bottles (Nalgene). Triplicate amendments of nutrients (see Methods section for full details) were performed and were incubated for 2 days. Additionally, three bottles were incubated with no amendment (controls) and three were sampled for initial conditions. Nutrient, trace element and phytoplankton community structure samples were collected alongside experimental seawater. Following incubation, bottles were sub-sampled for chlorophyll-a concentrations, flow cytometry cell counts, fast repetition rate fluorometry and diagnostic pigments (pooled samples from triplicate replicates). Underway sampling: Seawater was collected under trace-metal-clean conditions using a towed water sampling device (~2 m depth) for macronutrients, dissolved iron, chlorophyll-a concentrations, flow cytometry cell counts, fast repetition rate fluorometry, diagnostic pigments, and metaproteomics (5 sites only). Radiometric quantities were recorded continuously via hyperspectral radiometers at the bow of the ship. |
| Research sample | Natural mixed assemblages of microbial communities in surface seawaters encountered on the research cruise. |
| Sampling strategy | Nutrient amendment experiments were conducted with triplicate biological replicates, thus allowing for statistical testing whilst remaining logistically feasible in carrying out the field study. The underway sampling and nutrient amendment experiments were conducted at the highest spatial and temporal resolution possible during the oceanographic research cruise. |
| Data collection | Samples were collected by T. Browning, X. Wang, S. Garaba, and D. Voss on the research cruise. Samples were analyzed by T. Browning and several technical staff at GEOMAR Helmholtz Centre for Ocean Research Kiel (Germany). Proteomics samples were analyzed by M. McIlvin and D. Moran at Woods Hole Oceanographic Institute (USA). |
| Timing and spatial scale | Samples were collected between 28th January - 14th February 2019. Underway sampling was continuous or at as high temporal |

| | |
|---|---|
| Timing and spatial scale | frequency as possible. Sampling for bioassay experiments was conducted at regular intervals, setting up a new experiment after the previous had ended. Experimental samples were collected at night time in order that phytoplankton were dark acclimated. |
| Data exclusions | No data excluded |
| Reproducibility | Experimental: Identical experiments were conducted 4 times at different locations with treatments having triplicate replication. This was the maximum reproducibility possible during the fieldwork. |
| Randomization | Incubation bottles for the nutrient amendment experiments were filled at random. |
| Blinding | Investigators were not blinded to nutrient treatments. |

Did the study involve field work? ☒ Yes ☐ No

## Field work, collection and transport

| | |
|---|---|
| Field conditions | The seawater temperatures throughout the research cruise are shown in Extended Data Figure 1a. Sea conditions were calm. |
| Location | Samples were collected within the following domain: 151 degrees west to 107 degrees west, 3 degrees south to 17 degrees north. Samples were all collected from the near-sea surface (~2 m depth). |
| Access & import/export | No sampling was conducted within the Exclusive Economic Zones of any country. |
| Disturbance | Minimal disturbance was generated by the open ocean fieldwork activities (i.e., the presence of the research ship and towing of the seawater sampling device). All chemicals and seawater exposed to chemicals were transported back to Germany for disposal. |

# Reporting for specific materials, systems and methods

We require information from authors about some types of materials, experimental systems and methods used in many studies. Here, indicate whether each material, system or method listed is relevant to your study. If you are not sure if a list item applies to your research, read the appropriate section before selecting a response.

### Materials & experimental systems

| n/a | Involved in the study |
|---|---|
| ☒ | Antibodies |
| ☒ | Eukaryotic cell lines |
| ☒ | Palaeontology and archaeology |
| ☒ | Animals and other organisms |
| ☒ | Clinical data |
| ☒ | Dual use research of concern |
| ☒ | Plants |

### Methods

| n/a | Involved in the study |
|---|---|
| ☒ | ChIP-seq |
| ☐ | Flow cytometry |
| ☒ | MRI-based neuroimaging |

## Flow Cytometry

### Plots

Confirm that:

☒ The axis labels state the marker and fluorochrome used (e.g. CD4-FITC).

☒ The axis scales are clearly visible. Include numbers along axes only for bottom left plot of group (a 'group' is an analysis of identical markers).

☒ All plots are contour plots with outliers or pseudocolor plots.

☒ A numerical value for number of cells or percentage (with statistics) is provided.

### Methodology

| | |
|---|---|
| Sample preparation | Samples (2 mL) were fixed with neutralized paraformaldehyde at a 1% final concentration (paraformaldehyde: methanol-free 16% 10 mL glass ampules, Alfa Aesar/Thermo Fisher), vortex-mixed, and left in the dark for 10 minutes before being transferred to a –80 °C freezer. Samples were thawed at room temperature before analysis. |
| Instrument | FACSCalibur flow cytometer (Becton Dickenson, Oxford, United Kingdom). |

| | |
|---|---|
| Software | CellQuest software (Becton Dickenson). |
| Cell population abundance | Identification and counts of phytoplankton populations only (no cell sorting). Cell population counts determined using the gating strategy described below. |
| Gating strategy | Plots of orange fluorescence versus red fluorescence were used to identify and enumerate Synechococcus from other photosynthetic picoeukaryotes, and plots of side scatter versus red fluorescence (with any Synechococcus gated out) were used to enumerate photosynthetic picoeukaryotes. Gates were checked and adjusted manually for every sample to account for variations in fluorescence per cell. |

☒ Tick this box to confirm that a figure exemplifying the gating strategy is provided in the Supplementary Information.

