## [Peer Review File · Nature]

Manuscript Title: Persistent Equatorial Pacific iron limitation under ENSO forcing

Reviewer Comments & Author Rebuttals

Reviewer Reports on the Initial Version:

Referees' comments:

Referee #1 (Remarks to the Author):

The current manuscript adds important new observations from a ship transect in the equatorial Pacific to a substantial body of literature regarding nutrient stress in this region and diagnostic chlorophyll fluorescence patterns. The manuscript is well written and the observational data are impressive. To my knowledge, the diagnostic pigment results are novel, as is the presentation of the time series of chlorophyll and fluorescence quantum yields for the Nino 3 region of the equatorial Pacific. While I congratulate the authors on a very nice study, I did not find that sufficient new discoveries were presented to warrant publication in Nature and would encourage the authors to resubmit the manuscript to a more specialized journal.

The following is my basis for the above recommendation. The observed responses to N and Fe amendments shown in figure 1c are consistent with many similar measurements made in the past in the equatorial Pacific region. While the addition of cobalt, zinc, and silicic acid treatments has not been a common practice, these additions did not elicit significant responses by the phytoplankton during the current study. The nutrient stress biomarker results for Prochlorococcus shown in figure 1d are consistent with findings recently published in Science by Ustick et al. (2021). The diel fluorescence patterns (Fv/Fm, sigma-PSII) shown in figure 2a,b are essentially identical to those published for a much larger field data set in the same region in the cited Behrenfeld et al (2006) Nature paper and both the responses to iron additions (in terms of changes Fo, Fm, Fv/Fm, sigma-PSII) and the mechanistic interpretation of these patterns are essentially the same as earlier reported by Behrenfeld and colleagues in a series of papers (most of which are cited in the current manuscript). The changes in photosynthetic proteins shown in figure 2d are, to my knowledge, largely unique for field measurements, but are still very consistent with previous laboratory studies. The enhancement in fluorescence yields under iron limitation was also the basis for Behrenfeld et al's (2006) reassessment of satellite based primary production estimates for the equatorial Pacific and for the detection of iron stress from MODIS fluorescence quantum yield data in the cited Behrenfeld et al. (2009), which to first order is repeated here in figure 4. This earlier 2009 paper also reported a comparison between the satellite-detected iron stress regions and those predicted from an ecosystem model, much like the analysis currently presented in figure 4e. Thus again, while the current authors have presented a very nice analysis, I did not find that it represented a sufficient breakthrough to justify publication in Nature, but rather provides confirmation and further interpretation of previous findings.

Detailed comments (in order of appearance):

1. (lines 162-163) I would suggest that some additional analysis be conducted to evaluate if it is F_m that is particularly increased by iron stress or F_o . My guess is that it is the latter in terms of what is driving changes in F_v .

2. (Figure 1) (a) a bit more detail on what is meant by 'assumed-average phytoplankton requirements' would be helpful here. (b) panel e – the response of chlorophyll b to N addition seems to be similar between station 2 and 5, why might this be? (c) Why is the response at station 5 different between N and NCo? If anything, I would expect the latter to exceed the former but the observations indicate the opposite.

3. (Figure 2) (a) panel c – the term 'isi' is often specific in the literature to iron stress induced compounds in cyanobacteria, but the current study is conducted on mixed prokaryotic and eukaryotic species, so it might be better to use a different term than isi (?). (b) With respect to panel c, it is shown that the abundance of PSII is not down regulated under iron stress relative to other components of the linear electron transport chain (LETC) (and this is also stated in the main text), but readers might appreciate some interpretation as to why. In earlier studies on light-limited phytoplankton (e.g., Sukinec et al 1987), all elements of LETC are down regulated in parallel.

4. (Figure 3) (a) the emphasis in panel b is on fluorescence yield changes at night, while the figure is building up to the use of satellite fluorescence data to detect iron stress. In panel b it is not so clear that the yield per unit chlorophyll will be much different between the Fe limited and N limited regions in the middle of the day when satellite fluorescence measurements are collected. This expectation is realized in panel d where the difference between regions is rather small (order 0.1 units). (b) panel c – it appears that there are blue dots considerably higher than the blue line curve, but currently they are hidden behind the orange dots. It might be best to plot the blue and orange data on separate panels so the reader can see the scatter better. (c) panel c – the overall pattern that emerges in this figure of saturation of fluorescence at high light was already described in the cited Behrenfeld et al. (2009) paper. In this earlier paper, fluorescence quantum yields were calculated using normalized water leaving radiance data where an NPQ correction was applied that followed an inverse light function at saturating light (i.e., NPQ correction removes the dependence of yields on incident light at saturating levels, as illustrated in the current panel c). (d) panel d- the red line in this panel is supposed to correspond to MODIS-Aqua data, but the red line is continuous and varies both day and night. Why are MODIS data shown at night when the instrument only collects data near noon? (by the way, you should check for consistency in notation, some places in the text it is called Aqua-MODIS and other MODIS-Aqua, the latter form is the more common). (e) why compare shipboard data collected on specific days with MODIS data averaged over an entire season?

5. (Figure 4) (a) Why is so much of the subtropical/tropical Pacific whited out in panels a and e? It is not stated in the figure caption. I believe this is because of the tight threshold range of chlorophyll values used for flagging the data, but see comment below. (b) It would be useful to show a comparison of the time series of delta-Chl and delta phi in addition to the time series shown in panels c and d. By visual inspection, I think the chl-phi comparison will show a pretty remarkable correlation, but I could be wrong. What I'm afraid of here is that the delta-phi time series might be an artifact of the chlorophyll normalization rather than the proposed explanation that increased

chlorophyll will correspond to more Fe in the system and lower quantum yields. This explanation might be the case to first order, but the apparent correspondence between $\Delta\phi$ and Δ chlorophyll looks too close to be accounted for by this simple explanation.

6. (lines 739-742) It would be good to show a figure of this correlation between OC3 chlorophyll and measured chlorophyll, as well as report the slope of the regression, not just the R^2 and p value.

7. (lines 757-769) I do not understand why for the current study it was chosen to revert nFLH to FLH rather than follow published approaches that correct nFLH for NPQ. See note above regarding how the inverse light function of NPQ corrections accounts for the flattening of the fluorescence yield response at saturating light. Following earlier approaches with NPQ corrections might yield a similar qualitative pattern, but it might also give quantitatively higher or lower yields. Another element that is not discussed but has been in earlier papers is that the absolute fluorescence yield is dependent on photoacclimation state, which can give spatial patterns in satellite fields that are not necessarily indicative of different levels of iron stress.

8. (lines 773-780) (a) why divide satellite FLH data by chlorophyll concentration rather than absorption? Pigment absorption is the more directly measured property from space, while chlorophyll is more derived. (b) I am concerned about the small range of chlorophyll concentrations over which satellite fluorescence quantum yields were assessed. The lower bound of <0.1 mg Chl m^{-3} is largely based on the original assessment of Abbott and Letelier, prior to actual MODIS measurements. The upper bound of >0.4 mg Chl m^{-3} is not necessary if effects of pigment packaging are accounted for. In the Behrenfeld et al. (2009) study, they show that by correcting for pigment packaging and NPQ, a first order linear relationship exists between chlorophyll and fluorescence yields across chlorophyll concentrations ranging from ~ 0.03 to > 2 mg Chl m^{-3} (see their figure 2). Moreover, if the detection limit truly is 0.1 mg Chl m^{-3} , then calculated fluorescence yields below this threshold would be random (it would look like speckling in a global map), but in the fore-stated 2009 publication it is clear that this is not the case and instead there are very coherent spatial patterns in observed yields across the central ocean gyres.

9. (lines 830-855) (a) I do not understand the justification for stating that the rate constant for fluorescence loss would not be expected to change at night under iron limiting conditions. If the PQ pool is highly reduced under these conditions (and there is strong published evidence that this is the case), then a back transfer of electrons to Q_a will result in an increase in F_o (i.e., an enhanced k_f during the initial FRR flashes). In the absence of disconnected antennae complexes, this would yield a decrease in F_v (as observed), while F_m would not be impacted by iron stress (which seems to be the case in iron/macronutrient co-limited cells). (b) A direct link between night time increases in fluorescence and disconnected antennae was earlier postulated by Behrenfeld and colleagues, with the specific idea that reduction of the PQ pool at night would signal for a state transition, but these transitions would not be complete (i.e., connection to PSI) in many cases if PSI levels are very low (as indicated in lab studies and during the current study). (c) if the F_v change is largely or in part due to back transfers to Q_a with a highly reduced PQ pool and the state transitions are triggered by the same reduced pool, then both F_v and σ -PSII changes will be correlated but not necessarily causatively. Division of F_v by σ -PSII will thus result in a product that exhibits a dampened correlation with F_v/F_m . It is interesting that covariations in F_v/F_m and σ -PSII were also often

reported by the Behrenfeld group, but they noted that this was not always the case – as shown in the currently cited Behrenfeld and Kolber (1999) paper in Science.

Referee #2 (Remarks to the Author):

This paper reports ground-truthing measurements of increases in chlorophyll fluorescence of marine phytoplankton communities induced by iron limitation in a transect from low iron, high nitrate (HNLC) waters of the equatorial Pacific to low nitrate, low iron waters farther north. The authors were able to confirm iron limitation in the equatorial Pacific through a combination of iron and nitrate concentration measurements, increases chlorophyll concentrations with iron addition in bottle incubation experiments, molecular biomarkers for iron and nitrogen limitation, and measurement of chlorophyll (Chl) fluorescence and ambient Chl. Most importantly, they showed that the HNLC waters near the equator had high ratios of Chl fluorescence to Chl, something that had been previously well documented in laboratory and field studies, including those in the equatorial Pacific. They further showed that fluorescence to Chl ratios could be measured remotely in response to daytime solar radiation, initially using a ship mounted fluorometer and discrete Chl measurements in samples, but also from archived satellite-based measurements of both chlorophyll and Chl fluorescence collected over an 18-year period in the equatorial Pacific. The data showed that iron limitation in the equatorial Pacific decreased with upwelling strength, and attendant decreases in seawater temperature as a result of the upwelling of colder, iron- and nitrogen-rich deeper waters. Fluorescence-based values for iron limitation then increased with increasing temperature away from upwelling centers as the upwelled water warmed and iron and nitrogen concentrations were drawn down by the growth of the emerging phytoplankton community. The resulting relationship between increasing sea-surface temperature and fluorescence:Chl ratios determined from the satellite data were significant and invariant over many El Niño–Southern Oscillation (ENSO) cycles, but differed by twofold from values predicted from a biogeochemical model for chlorophyll and marine productivity currently used to predict future effects of climate change. These are important results because they suggest that existing and future satellite data can be used to map iron limitation world-wide in the ocean and that models for marine chlorophyll and productivity currently used to predict future effects of Climate Change likely need modification.

Specific comments

Lines 118-122 – Yes, these diel changes in phytoplankton pigment fluorescent patterns are well documented for the transition from the iron limited equatorial Pacific to the N-limited North Pacific central gyre.

Lines 131-160 – The lack of detection of PSI and cyt b6/f Prochlorococcus proteins at the iron limited stations 1-3 is problematic as the cyanobacteria would need these proteins for linear electron flow between PSII and flavodoxin, needed for the production of reductant (NADPH) used for the fixation of carbon and the reduction of nitrate to ammonium. Note that PSII proteins and flavodoxin were detected at these stations, suggesting that PSI and the b6/f complexes must have also been present. A possible explanation for this discrepancy is that PSI and b6/f complexes were present in these

Prochlorococcus communities, but that their proteins were different enough in these low-iron adapted populations from the protein “standards” used for identification, that they were not identified. The use of appropriate “standards” for protein identification is a common problem with proteomics.

Lines 167-169 – How rapid was this decrease? The data in figure 3a are for the samples taken at the end of the incubation experiments. However, time course data in other experiments conducted in the Equatorial Pacific iron-limited region indicate that such decreases in fluorescence/Chl occur within one day’s time (see Behrenfeld and Milligan 2013). This paper and those cited therein should be mentioned here.

Lines 180-181 – There is substantial evidence that the highly fluorescent light harvesting complexes under conditions of iron limitation and high available nitrogen (i.e., HNLC conditions) result not just from LNCs that are “weakly connected to reaction centers” but rather are disconnected to photosynthetic reaction centers (again see Behrenfeld and Milligan 2013).

Lines 189 and 190 – This has also been shown previously (again see Behrenfeld and Milligan 2013 and references cited therein), which should be cited here.

William Sunda

Referee #3 (Remarks to the Author):

The manuscript reports changes in the chlorophyll-fluorescence characteristics of phytoplankton populations along an oceanic transect. This transect transitions between distinct zones of nutrient limitation. The manuscript correlates measured physiology to changes in the satellite fluorescence record to derive modifications to climate models.

I have been asked to comment on the in situ photo-physiology aspects of this manuscript and I hope the authors find these comments helpful in the revision of the manuscript.

As mentioned in the manuscript, this is not the first report of changes associated with in situ photo-physiology as one moves north from the equatorial Pacific. I think the manuscript does a good job acknowledging the previous observations and interpretations. The manuscript also does a very good job established the nature of nutrient limitation in the different sampling locations.

The novelty associated with this work is the addition of proteomics to mechanistically explain what was previously hypothesized.

I have two issues with the manuscript that could be addressed by the authors.

First of all, the manuscript focusses on a derived fluorescence parameter, F_v , which is variable fluorescence. F_v is not directly measured by a fluorometer, but instead it is calculated by subtracting

the minimal level of fluorescence (F_0) from the fluorescence maximum (F_m) measured by a fast repetition rate fluorometer. The manuscript goes through some effort to explain why it uses this parameter, but I am confused as to why this decision was made. Ultimately, disconnecting of light harvesting antenna from the photosystems causes an increase in F_0 and this is what drives changes in the variable fluorescence parameter (F_v). This is well established in the literature. So, the authors should consider changing the approach to use F_0 throughout this work. Supplementary discussion 1 includes a calculation derived from Oxborough et al. (2012). I note they use some different symbols, which is confusing. Please provide more detail about how this equation was derived. I was unable to convert the approach of Oxborough to what is reported here so there appear to be some missing steps or assumptions in this manuscript.

A key argument associated with the in situ physiology is the detachment of light harvesting antenna from the photosystems. The manuscript convincingly shows a change in the ratio of photosystem II to photosystem I core proteins via proteomics. However, were the light harvesting antenna not observed in this data set? Knowing the relative abundance changes in these proteins (or if they were even detected) would go a long way to supporting the hypotheses associated with the fluorescence data. Either way, the data shows changes in the proteome of *Prochlorococcus* but the interpretation of this data is based around a wide variety of organisms that have different light harvesting systems from *Prochlorococcus* including cyanobacteria with phycobilisomes (ref 32,37) or even eukaryotes (ref 31,36). This is not appropriate. Finally, the manuscript should not use the acronym LHC as it refers to a specific family of chlorophyll a binding proteins from eukaryotes.

Referee #4 (Remarks to the Author):

Browning et al present multiple data types from a cruise in the equatorial Pacific aimed at understanding the physiological impacts of nutrient limitation on phytoplankton physiology and optical properties and using constructs from that data to more tightly constrain NPP projections in the context of climate change. The set of experimental measurements presented are uniquely comprehensive and well-integrated, making this study particularly valuable as a synoptic perspective. For example, strikingly clear evidence of nutrient limitation was observed using meta-proteomics approaches and the conclusions from that analysis are also supported by shipboard incubation experiments in which nutrients were added and the chlorophyll response was measured. The comprehensiveness and virtuosity of this work are important because they leave little to doubt about the physiological state of the phytoplankton, so that the association with corresponding optical measurements is sound. This is beautiful work.

A central self-reported unique contribution from this work is utilizing radiometers on the vessel to derive a relative measurement of sunlight-stimulated chlorophyll fluorescence, and showing that a relative measure of SSCF can be obtained from satellite data, using the assumption of irradiance independence that is supported by data shown in Fig. 3C. The authors then go on to use these constructs to produce global maps of satellite-derived Fe limitation. A number of studies, which are cited, previously have applied remotely sensed optical signatures to predict nutrient limitation and primary productivity in this region of the ocean. Recommendations follow. I like this paper, but I

think the message might need to be distilled to its essence.

1. Perhaps a bit simplistic, but isn't it to be expected that iron limitation will be somewhat binary, in that the phytoplankton community either doesn't have enough, or they do. Could the global model then still be right (i.e. not be overestimating NPP sensitivity to Fe input by 2 fold), in the sense that NPP is changing with changing Fe input but not the Fe limitation status at the cellular level? Towards the end of the paper this view is represented but towards the beginning the emphasis seems to be on the model underestimates.

2. While I was impressed with the quality of this manuscript I was left uncertain about observations that confirm previous reports and observations that are new with respect to satellite detection. I recommend the authors clarify. Specifically, the authors should briefly contrast their approach and findings with previous studies that predicted Fe limitation from satellite data. My impression is this manuscript uses satellite data differently, and is different in emphasizing ENSO variation and its relatively low impact on Fe limitation, and suggesting that models overstate ENSO impacts on Fe limitation. But, is this insight a direct consequence of the new relative measure of SSCF?

3. Line 255, and elsewhere. Central to the findings is the observation that nutrient limitation is resilient to fluctuations in nutrient input caused by ENSO. At various points in the manuscript there is a suggestion that phytoplankton community restructuring could explain this resilience, but the reasoning behind this is unclear. The data in the manuscript don't resolve phytoplankton diversity in great detail, except for the metaproteomics, which are very specific. Are the authors suggesting that taxa adapted to a continuous state of iron limitation inhabit iron-limited ocean regions? My interpretation of the manuscript is that the change in expression of proteins is a reflection of the physiological state of the cells, but elsewhere it sounds like the authors are invoking community turnover. To avoid confusion perhaps be more clear about how community turnover (i.e. changes in communities) could decouple nutrient input from NPP.

4. Line 796, L Lowercase F and (Line 792), L lowercase Fe. Is that a typo? If not clarify difference. I suggest defining Delta Fe lowercase lim, y axis label Fig. 4f, in legend since its easy for a reader to become confused.

5. If more space is needed to strengthen the narrative then I suggest reducing the section about diel cycles. Its interesting but not essential to the central arguments.

Author Rebuttals to Initial Comments:

Response to Reviewer comments

We thank the three Reviewers for their detailed comments. We have responded to these below in blue.

To refer between review responses, we have numbered reviewer comments (R1_1, for Reviewer 1, comment 1, and so on).

Referee #1 (Remarks to the Author):

The current manuscript adds important new observations from a ship transect in the equatorial Pacific to a substantial body of literature regarding nutrient stress in this region and diagnostic chlorophyll fluorescence patterns. The manuscript is well written and the observational data are impressive. To my knowledge, the diagnostic pigment results are novel, as is the presentation of the time series of chlorophyll and fluorescence quantum yields for the Nino 3 region of the equatorial Pacific. While I congratulate the authors on a very nice study, I did not find that sufficient new discoveries were presented to warrant publication in Nature and would encourage the authors to resubmit the manuscript to a more specialized journal.

The following is my basis for the above recommendation. The observed responses to N and Fe amendments shown in figure 1c are consistent with many similar measurements made in the past in the equatorial Pacific region. While the addition of cobalt, zinc, and silicic acid treatments has not been a common practice, these additions did not elicit significant responses by the phytoplankton during the current study. The nutrient stress biomarker results for Prochlorococcus shown in figure 1d are consistent with findings recently published in Science by Ustick et al. (2021). The diel fluorescence patterns (Fv/Fm, sigma-PSII) shown in figure 2a,b are essentially identical to those published for a much larger field data set in the same region in the cited Behrenfeld et al (2006) Nature paper and both the responses to iron additions (in terms of changes Fo, Fm, Fv/Fm, sigma-PSII) and the mechanistic interpretation of these patterns are essentially the same as earlier reported by Behrenfeld and colleagues in a series of papers (most of which are cited in the current manuscript). The changes in photosynthetic proteins shown in figure 2d are, to my knowledge, largely unique for field measurements, but are still very consistent with previous laboratory studies. The enhancement in fluorescence yields under iron limitation was also the basis for Behrenfeld et al's (2006) reassessment of satellite based primary production estimates for the equatorial Pacific and for the detection of iron stress from MODIS fluorescence quantum yield data in the cited Behrenfeld et al. (2009), which to first order is repeated here in figure 4. This earlier 2009 paper also reported a comparison between the satellite-detected iron stress regions and those predicted from an ecosystem model, much like the analysis currently presented in figure 4e. Thus again, while the current authors have presented a very nice analysis, I did not find that it represented a sufficient breakthrough to justify publication in Nature, but rather provides confirmation and further interpretation of previous findings.

R1_1: We thank the Reviewer for their detailed consideration of our manuscript. The major contribution of our study is the quantification of two decades of shifting iron limitation in the Equatorial Pacific that is over estimated by a state-of-the-art global model. This raises new questions regarding the responses of marine ecology and productivity to changing iron stress. None of these insights have been realised by the prior studies mentioned by the reviewer (and cited in our manuscript) because they require the multi-disciplinary, coherent approach of our study - spanning ocean biogeochemistry, genomics (both nutrient limitation

and photophysiology), optics, remote sensing and ocean modelling. That being said, we agree that these core messages were diluted by discussion of issues not strictly pertinent. The manuscript has now undergone major revision to streamline our findings and their importance.

The observed responses to N and Fe amendments shown in figure 1c are consistent with many similar measurements made in the past in the equatorial Pacific region. While the addition of cobalt, zinc, and silicic acid treatments has not been a common practice, these additions did not elicit significant responses by the phytoplankton during the current study.

R1_2: We agree the experiments confirm the nutrient limitation regime, but do not add substantially beyond what we knew already (i.e., systems limited by Fe versus N). However, conducting and presenting the results of these experiments were fundamental to the bigger objective of the project, as they are currently the only means to assess nutrient limitation of the overall phytoplankton community. Therefore, while we believe it was imperative to conduct these, detailed discussion of the results were probably not needed and this part of the revised manuscript has been modified to simply state the key experimental findings.

Confirmation of the absence of limitation by other nutrients was also something we believed was important to show to make the claim of iron versus nitrogen limitation (as the Reviewer points out, such experimental tests have not been conducted in previous experiments in the systematic way we did here).

As an additional note, we point out that the resolution of community structure by both pigment analysis and flow cytometry was also not conducted in most previous experiments where active chlorophyll fluorescence signals were being evaluated (i.e., Behrenfeld et al., 2006), but are crucial in confidently interpreting the active fluorescence measurements in terms of nutrient limitation (Proctor and Roesler, 2010; Suggett et al., 2009).

References

- Proctor, C.W. and Roesler, C.S., 2010. New insights on obtaining phytoplankton concentration and composition from in situ multispectral Chlorophyll fluorescence. *Limnol. Oceanogr. Meth.* **8**, 695-708.
- Suggett, D.J., Moore, C.M., Hickman, A.E. and Geider, R.J., 2009. Interpretation of fast repetition rate (FRR) fluorescence: signatures of phytoplankton community structure versus physiological state. *Marine Ecology Progress Series*, **376**, pp.1-19.

The nutrient stress biomarker results for *Prochlorococcus* shown in figure 1d are consistent with findings recently published in Science by Ustick et al. (2021).

R1_3: We agree, although the Ustick study focussed on metagenomic data, rather than the measurements of metaproteomics in our work. We reiterate that our major finding is not to demonstrate the Fe-N limitation transition, but to explore and quantify climate variations in nutrient limitation. In many respects, the more directly relevant prior work is the nutrient stress biomarker study of Saito et al. (2013). However, whilst very much breakthroughs in their own right, neither Ustick et al. (2021) nor Saito et al. (2013) conducted complimentary nutrient addition bioassay experiments or any type of fluorescence measurement. The latter are both central to the main objective of this work.

References

Saito, M.A., McIlvin, M.R., Moran, D.M., Goepfert, T.J., DiTullio, G.R., Post, A.F. and Lamborg, C.H., 2014. Multiple nutrient stresses at intersecting Pacific Ocean biomes detected by protein biomarkers. *Science* **345**, 1173-1177.

The diel fluorescence patterns (Fv/Fm, sigma-PSII) shown in figure 2a,b are essentially identical to those published for a much larger field data set in the same region in the cited Behrenfeld et al (2006) Nature paper and both the responses to iron additions (in terms of changes Fo, Fm, Fv/Fm, sigma-PSII) and the mechanistic interpretation of these patterns are essentially the same as earlier reported by Behrenfeld and colleagues in a series of papers (most of which are cited in the current manuscript).

R1_4: We fully agree that the diel signals in active fluorescence measurements, Fv/Fm and sigma-PSII, are basically the same as in Behrenfeld et al. (2006). We point out that chlorophyll-normalized active fluorescence measurements were not reported in Behrenfeld et al. (2006 or others), but are a key stepping stone towards chlorophyll-a normalized *passive* fluorescence measurements.

As a minor aside, we also note that our interpretation of diel cycles in Fv/Fm and sigma-PSII differed somewhat to Behrenfeld et al. (2006) as shown by our extended analysis of the data (e.g., Fig. 2e–g, Supplementary Discussion 1, see also responses to other reviewers below). However, in the revised manuscript we have now streamlined this section of the main text to more quickly come to the main point about relative chlorophyll fluorescence yields, with these details provided in the supplementary text.

The changes in photosynthetic proteins shown in figure 2d are, to my knowledge, largely unique for field measurements, but are still very consistent with previous laboratory studies.

R1_5: Similar results have indeed been found for a restricted set of phytoplankton cultures (reviewed in e.g., Behrenfeld and Milligan, 2013). But, as the reviewer suggests, the changes in relative abundances of PSII, PSI, and cytochrome b6f are, to our knowledge, unique for field measurements. We feel this is an important advance, although again, on its own not the major novelty of our work. Specifically, these protein observations feed into a much stronger interpretation of the fluorescence data (and then onto satellite observations of nutrient limitation).

The enhancement in fluorescence yields under iron limitation was also the basis for Behrenfeld et al's (2006) reassessment of satellite based primary production estimates for the equatorial Pacific and for the detection of iron stress from MODIS fluorescence quantum yield data in the cited Behrenfeld et al. (2009), which to first order is repeated here in figure 4.

R1_6: We list below several key factors that we believe are critical in distinguishing our contribution from important prior works by Behrenfeld et al (2006, 2009). We consider these to be fundamental to meaningful use of the satellite fluorescence data:

- We diagnose the nutrient limitation regime via a unique combination of phytoplankton biomass changes in experiments and protein biomarkers (neither of these were conducted in Behrenfeld et al. (2006) and not conducted together in any prior study to our knowledge).
- We demonstrate changes in active fluorescence signals following short-term nutrient addition were independent of phytoplankton community structure shifts.
- We show that sunlight-stimulated, *passive* fluorescence (unlike other studies that have used *active* fluorescence only, which is not what the satellite observes and

could potentially lead to major differences; Cullen et al., 1988; Cullen and Lewis, 1995; Huot and Babin, 2010) is light saturated at irradiances below that of satellite chlorophyll fluorescence retrievals. We have only demonstrated this for the tropical Pacific (measurements elsewhere not conducted), and this is one reason why we focus our remote sensing study on this region.

- We show that changes in this light-saturated, sunlight-stimulated chlorophyll-normalized fluorescence matches up with the iron-to-nitrogen limitation transition.
- We document that the fold change in (i) active fluorescence between iron and nitrogen limited sites, (ii) active fluorescence following experimental iron addition at iron limited sites, (iii) shipboard, sunlight-stimulated *passive* chlorophyll-normalized fluorescence under light saturation between iron and nitrogen limited sites, and finally (iv) the satellite-derived passive fluorescence between iron and nitrogen limited sites, are all consistently around a value of 3. Collectively, this strong quantitative consistency convincingly supports nutrient regulation of this signal (and specifically N versus Fe limitation).
- Following this extensive field validation (never before conducted), we are able to quantify variability in satellite-derived tropical Pacific nutrient limitation over 20 years for the first time and compare this to a model with realistic ENSO dynamics. Key findings from this are: (i) iron limitation varies with upwelling strength and that even under the weakest upwelling on record, the equatorial system remains iron (rather than nitrogen) limited. (ii) changes in the intensity of iron limitation following physical forcing over ENSO cycles are overestimated by the ocean biogeochemical model.

References

- Behrenfeld, M.J., Worthington, K., Sherrell, R.M., Chavez, F.P., Strutton, P., McPhaden, M. and Shea, D.M., 2006. Controls on tropical Pacific Ocean productivity revealed through nutrient stress diagnostics. *Nature* **442**, 1025-1028.
- Behrenfeld, M.J., Westberry, T.K., Boss, E.S., O'Malley, R.T., Siegel, D.A., Wiggert, J.D., Franz, B.A., McLain, C.R., Feldman, G.C., Doney, S.C. and Moore, J.K., 2009. Satellite-detected fluorescence reveals global physiology of ocean phytoplankton. *Biogeosciences* **6**, 779–794.
- Cullen, J.J., Yentsch, C.M., Cucci, T.L. and MacIntyre, H.L., 1988, August. Autofluorescence and other optical properties as tools in biological oceanography. In *Ocean optics IX* (925, 149-156). International Society for Optics and Photonics.
- Cullen, J.J. and Lewis, M.R., 1995. Biological processes and optical measurements near the sea surface: Some issues relevant to remote sensing. *J. Geophys. Res. Oceans* **100**, 13255-13266.
- Huot, Y. and Babin, M., 2010. Overview of fluorescence protocols: theory, basic concepts, and practice. In *Chlorophyll a Fluorescence in Aquatic Sciences: Methods and Applications* (pp. 31-74). Springer, Dordrecht.

This earlier 2009 paper also reported a comparison between the satellite-detected iron stress regions and those predicted from an ecosystem model, much like the analysis currently presented in figure 4e.

R1_7: Whilst the spatial comparison in Figure 4e is similar to the approach used in Behrenfeld et al. (2009), the focus of our analysis (time series analysis, model comparison) is very different.

Thus again, while the current authors have presented a very nice analysis, I did not find that it represented a sufficient breakthrough to justify publication in Nature, but rather provides confirmation and further interpretation of previous findings.

Detailed comments (in order of appearance):

1. (lines 162-163) I would suggest that some additional analysis be conducted to evaluate if it is Fm that is particularly increased by iron stress or Fo. My guess is that it is the latter in terms of what is driving changes in Fv.

R1_8: We find that both Fo/Chl and Fm/Chl increase; this is a clear indicator that the majority of the differences in the fluorescence yields are due to energetically decoupled pigment protein complexes (see e.g., Macey et al. 2014). As the reviewer predicts, in the iron limited zone values of Fo/Chl between non-Fe amended and Fe-amended treatments show a mean fold difference of 4.2, whilst Fm/Chl shows a mean fold difference of 3.0, i.e., around 30% lower, however this still indicates that >70% of the difference (i.e., the majority) is due to simultaneous changes in both Fo and Fm. Moreover, the light-saturated *active* fluorescence (i.e., Fm) is most comparable to that of light-saturated *passive* fluorescence, which is why here it is most meaningful to present the Fm values. More detailed discussion on the interpretation of Fv is included in the responses to the latter specific questions below.

Reference

Macey, A.I., Ryan-Keogh, T., Richier, S., Moore, C.M. and Bibby, T.S., 2014. Photosynthetic protein stoichiometry and photophysiology in the high latitude North Atlantic. *Limnol. Oceanogr.* **59**, 1853-1864.

2. (Figure 1) (a) a bit more detail on what is meant by 'assumed-average phytoplankton requirements' would be helpful here.

R1_9: The phrase 'assumed average phytoplankton nutrient requirements' was used as the value comes from an average of a compilation of values (Moore et al., 2013), which we assume is reflective of an average across multiple phytoplankton types and growth conditions. In the revised manuscript, the value used has been added in (N:Fe of 2132 mol:mol).

(b) panel e – the response of chlorophyll b to N addition seems to be similar between station 2 and 5, why might this be?

R1_10: We cannot identify the similarity: specifically, in Experiment 2, the normalized chlorophyll-b concentrations are:

N addition (with or without added Co): 0.51

Fe addition (with or without added Co): 0.73-0.81

N+Fe addition (with or without Co, Zn or silicic acid): 0.71-1

Conversely, in Experiment 5, the normalized chlorophyll-b concentrations are:

N addition (with or without added Co): 0.27-0.41

Fe addition (with or without added Co): 0.16-0.2

N+Fe addition (with or without Co, Zn or silicic acid): 0.87-1

Therefore, the trends in chlorophyll-b closely follow that of overall chlorophyll-a, responding most to Fe addition in Experiment 2 (with or without N) and a small amount to N addition but most to N+Fe addition in Experiment 5.

(c) Why is the response at station 5 different between N and NCo? If anything, I would expect the latter to exceed the former but the observations indicate the opposite.

R1_11: We cannot identify the difference: Experiment 5 shows almost identical chlorophyll-a response to N or N+Co (see the overlapping blue points in Fig. 3c).

3. (Figure 2) (a) panel c – the term ‘isi’ is often specific in the literature to iron stress induced compounds in cyanobacteria, but the current study is conducted on mixed prokaryotic and eukaryotic species, so it might be better to use a different term than isi (?).

R1_12: As ‘isi’ has consistently been used as ‘iron stress induced’ in the literature we do not see why it should not be used in a general sense here (i.e., overall phytoplankton community).

(b) With respect to panel c, it is shown that the abundance of PSII is not down regulated under iron stress relative to other components of the linear electron transport chain (LETC) (and this is also stated in the main text), but readers might appreciate some interpretation as to why. In earlier studies on light-limited phytoplankton (e.g., Sukinec et al 1987), all elements of LETC are down regulated in parallel.

R1_13: We expect this to result from PSII having a relatively low Fe requirement. This is similar to other laboratory studies; e.g., Strzepek and Harrison (2004) showed that PSII abundance changed relatively little in response to low Fe conditions in comparison to PSI, which showed large decreases.

4. (Figure 3) (a) the emphasis in panel b is on fluorescence yield changes at night, while the figure is building up to the use of satellite fluorescence data to detect iron stress. In panel b it is not so clear that the yield per unit chlorophyll will be much different between the Fe limited and N limited regions in the middle of the day when satellite fluorescence measurements are collected. This expectation is realized in panel d where the difference between regions is rather small (order 0.1 units).

R1_14: The differences in chlorophyll-normalized active fluorescence between the two nutrient limitation regimes in the daytime is expected to be minimal, or at least quite variable, as strong and variable NPQ processes depress daytime active fluorescence in both regions. In contrast, the passive, sunlight stimulated fluorescence is balanced by increased absorbed PAR energy for fluorescence and NPQ processes (see Fig. 3c). Although the scatter in the data is greater for the passive observations in panel ‘d’, the fold difference between the two nutrient limitation regimes is similar to that of the night-time chlorophyll-normalized active fluorescence in panel ‘b’ (3.5 and 3.2-fold differences respectively).

(b) panel c – it appears that there are blue dots considerably higher than the blue line curve, but currently they are hidden behind the orange dots. It might be best to plot the blue and orange data on separate panels so the reader can see the scatter better.

R1_15: At low PAR there are indeed some outlier points falling higher than the red line (see below for replot; replaced in revised manuscript). These are however only a minor fraction of the total data and are included in the binned average shown by the thick line/points.

(c) panel c – the overall pattern that emerges in this figure of saturation of fluorescence at high light was already described in the cited Behrenfeld et al. (2009) paper. In this earlier paper, fluorescence quantum yields were calculated using normalized water leaving radiance data where an NPQ correction was applied that followed an inverse light function at saturating light (i.e., NPQ correction removes the dependence of yields on incident light at saturating levels, as illustrated in the current panel c).

R1_16: Indeed, Behrenfeld et al. (2009) used an inverse light function to NPQ-correct satellite fluorescence data. Essentially their NPQ correction, which was multiplication of nFLH by iPAR (they then scaled this to the mean global iPAR, but this does not impact trends), is equivalent to multiplying nFLH by the cosine of the solar zenith angle, which is what we do here (following Gower et al., 2014). The latter could be considered a more straightforward approach (i.e., by assuming the impacts of sunlight stimulation and NPQ have opposite effects on FLH that cancel each other out at elevated PAR values, hence no correction needed). As acknowledged by Behrenfeld et al. (2009), no data were provided or available in the literature that actually showed the functional form of the passive, chlorophyll-normalized fluorescence under iron and nitrogen limitation. Indeed, this might have been quite different (Cullen, 2009), and thus had a major impact on the resultant data. Therefore, our finding that both iron and nitrogen limited sites reach passive fluorescence plateaus at irradiances below that of satellite operational irradiance is a crucial part of our use of the satellite fluorescence a reliable nutrient limitation diagnostic.

Related to this, in order to avoid confusion, we have now renamed ‘relative fluorescence quantum yields’ (defined as Φ_{rel}) to simply F/Chl in the revised manuscript.

References

- Behrenfeld, M.J., Westberry, T.K., Boss, E.S., O'Malley, R.T., Siegel, D.A., Wiggert, J.D., Franz, B.A., McLain, C.R., Feldman, G.C., Doney, S.C. and Moore, J.K., 2009. Satellite-detected fluorescence reveals global physiology of ocean phytoplankton. *Biogeosciences* **6**, 779–794.
- Cullen, J.J., 2009. Interactive comment on “Satellite-detected fluorescence reveals global physiology of ocean phytoplankton” by MJ Behrenfeld et al. *Biogeosci. Discuss*, **5**, pp.S2646-S2655.
- Gower, J.F.R., 2014. A simpler picture of satellite chlorophyll fluorescence. *Remote Sens. Lett.* **5**, 583-589.

(d) panel d- the red line in this panel is supposed to correspond to MODIS-Aqua data, but the red line is continuous and varies both day and night. Why are MODIS data shown at night when the instrument only collects data near noon? (by the way, you should check for

consistency in notation, some places in the text it is called Aqua-MODIS and other MODIS-Aqua, the latter form is the more common).

R1_17: The line showing the MODIS data corresponds to the continuous location along the transect (note that the x-axis of the plot is distance) and not the specific day-night of our fieldwork (overplotted with shading for reference).

In the revised manuscript we correct for MODIS-Aqua and Aqua-MODIS (all to MODIS-Aqua, as recommended).

(e) why compare shipboard data collected on specific days with MODIS data averaged over an entire season?

R1_18: The satellite fluorescence data unfortunately suffers from cloud cover and gaps in the overpass coverage for any given day (particularly pronounced at low latitudes), which largely prevents same-day/location field-satellite match-ups. Therefore, a composite image was used to get optimal coverage.

5. (Figure 4) (a) Why is so much of the subtropical/tropical Pacific whited out in panels a and e? It is not stated in the figure caption. I believe this is because of the tight threshold range of chlorophyll values used for flagging the data, but see comment below.

R1_19: The majority of the whited-out panels corresponds to data with chlorophyll-a concentrations $<0.1 \text{ mg m}^{-3}$. This is the calculated detection limit for MODIS-Aqua FLH (Huot et al. 2013; See also response to R1_24 where this is discussed in more detail). At the lowest chlorophyll-a concentrations, errors in the satellite chlorophyll retrievals can also be up to 50%. In combination, this could introduce major errors/bias in the resultant fields. Evidently, more fieldwork needs to be conducted in these regions to investigate this before they can be robustly interpreted.

The upper boundary of 0.4 mg m^{-3} was set as beyond these values pigment packaging can start to have an important effect (see R1_23 for more detail). As pigment packaging can show important regional variability, a single chlorophyll-to-absorption conversion equation could introduce error/bias. In addition, and perhaps most importantly, these very low and higher chlorophyll-a systems respectively were not investigated in this study with regards to irradiance-passive fluorescence relationships (Fig. 3c) and we therefore cannot be fully confident that the high light passive fluorescence saturation found in our tropical Pacific study region applies in these systems.

References

Huot, Y., Franz, B.A. and Fradette, M., 2013. Estimating variability in the quantum yield of Sun-induced chlorophyll fluorescence: A global analysis of oceanic waters. *Remote Sens. Environ.* **132**, 238-253.

(b) It would be useful to show a comparison of the time series of delta-Chl and delta phi in addition to the time series shown in panels c and d. By visual inspection, I think the chl-phi comparison will show a pretty remarkable correlation, but I could be wrong. What I'm afraid of here is that the delta-phi time series might be an artifact of the chlorophyll normalization rather than the proposed explanation that increased chlorophyll will correspond to more Fe in the system and lower quantum yields. This explanation might be the case to first order, but the apparent correspondence between delta phi and delta chlorophyll looks too close to

be accounted for by this simple explanation.

R1_20: The correlation statistic between delta-chl and delta-phi is $R^2=0.40$, $p=6.4e-26$. This is less strong than the correlation between delta-phi and SST ($R^2=0.50$, $p=1.4e-34$). Therefore, whilst we expect both to correlate (i.e., less upwelling, enhanced Fe limitation, and lower chlorophyll are all expected to co-vary), the delta-phi appears much more sensitive to upwelling strength. This is likely because physiological nutrient limitation is directly influenced by nutrient supply rates, whereas chlorophyll-a biomass will also be under top-down grazing control. We have now included this in the revised manuscript. Note that delta-phi is now simply stated as F/Chl in the revised manuscript.

6. (lines 739-742) It would be good to show a figure of this correlation between OC3 chlorophyll and measured chlorophyll, as well as report the slope of the regression, not just the R^2 and p value.

R1_21: This has now been included in the revised methods section.

7. (lines 757-769) I do not understand why for the current study it was chosen to revert nFLH to FLH rather than follow published approaches that correct nFLH for NPQ. See note above regarding how the inverse light function of NPQ corrections accounts for the flattening of the fluorescence yield response at saturating light. Following earlier approaches with NPQ corrections might yield a similar qualitative pattern, but it might also give quantitatively higher or lower yields. Another element that is not discussed but has been in earlier papers is that the absolute fluorescence yield is dependent on photoacclimation state, which can give spatial patterns in satellite fields that are not necessarily indicative of different levels of iron stress.

R1_22: We refer the reviewer to our earlier response, R1_16. We also note that throughout our manuscript we referred to our yields as relative, as the absolute values will depend upon their treatment (including for example normalization to absorbed irradiance or simply chlorophyll-a). In the revised manuscript delta-phi is now simply stated as F/Chl to avoid any confusion.

We agree that photoacclimation state might be playing an important role in regulating passive fluorescence yields in higher latitude regions with strong seasonality in available light. This is currently an unknown, with (to our knowledge) no direct seasonal field or laboratory data to support or refute it (i.e., passive fluorescence versus irradiance responses such as in Figure 3c need to be investigated under such conditions with corresponding ancillary data). However, there is minimal variability in upper water column growth irradiance for the equatorial region focussed on here, meaning that changes in photoacclimation state will in this case be minor (see e.g., Behrenfeld et al., 2005 Fig. 2f and Table 1).

Reference

Behrenfeld, M.J., Boss, E., Siegel, D.A. and Shea, D.M., 2005. Carbon-based ocean productivity and phytoplankton physiology from space. *Global biogeochemical cycles*, 19(1).

8. (lines 773-780) (a) why divide satellite FLH data by chlorophyll concentration rather than absorption? Pigment absorption is the more directly measured property from space, while chlorophyll is more derived.

R1_23: We agree that a robust satellite phytoplankton light absorption parameter would be a preferable option for normalizing FLH. However, current algorithms for directly measuring phytoplankton absorption from satellite remain much less well validated in comparison to chlorophyll-a algorithms. We note that the previous attempt to normalize satellite fluorescence to phytoplankton absorption used a general published relationship between absorption and chlorophyll; that is, they introduced an additional further derivation/assumption regarding a globally-constant chlorophyll–phytoplankton absorption relationship (Behrenfeld et al., 2009). This potentially introduces bias/error as (i) the relationship might not be globally uniform, (ii) the relationship at low chlorophyll-a (which dominates our study region as well as the global ocean) is indistinguishable from linear (see, for example, Figure 4 in Bricaud et al. (1995) and figure reproduced with Equatorial Pacific data below), but a relationship generated across a full chlorophyll-a range (including many very high chlorophyll values in the dataset) leads to an extension of the curvature to the lower chlorophyll-a range (see e.g. Bricaud et al., 1995 Fig. 4 and Behrenfeld et al., 2009 Fig. 2b). For regions with relatively low chlorophyll-a concentrations, representing our study region, chlorophyll-a concentrations themselves therefore currently remain the better parameter for satellite-based normalization.

Plot of phytoplankton absorption at 440 nm versus total chlorophyll-a (chlorophyll-a + divinyl chlorophyll-a) for the Olipac cruise discussed in Bricaud et al. (2004), which investigated the equatorial and subequatorial Pacific region (i.e., similar to this study). The red line is the global relationship (Chl ~0 to >20 mg m⁻³) of Bricaud et al. (1995) $a_{ph}(440)=0.0403Chl^{0.668}$. Note how the curvature in the global fit is not apparent in the Equatorial Pacific data, which are indistinguishable from linear.

References

- Behrenfeld, M.J., Westberry, T.K., Boss, E.S., O'Malley, R.T., Siegel, D.A., Wiggert, J.D., Franz, B.A., McLain, C.R., Feldman, G.C., Doney, S.C. and Moore, J.K., 2009. Satellite-detected fluorescence reveals global physiology of ocean phytoplankton. *Biogeosciences* **6**, 779–794.
- Bricaud, A., Babin, M., Morel, A. and Claustre, H., 1995. Variability in the chlorophyll-specific absorption coefficients of natural phytoplankton: Analysis and parameterization. *Journal of Geophysical Research: Oceans*, *100*(C7), pp.13321-13332.

Bricaud, A., Claustre, H., Ras, J. and Oubelkheir, K., 2004. Natural variability of phytoplanktonic absorption in oceanic waters: Influence of the size structure of algal populations. *Journal of Geophysical Research: Oceans*, 109(C11).

(b) I am concerned about the small range of chlorophyll concentrations over which satellite fluorescence quantum yields were assessed. The lower bound of <0.1 mg Chl m^{-3} is largely based on the original assessment of Abbott and Letelier, prior to actual MODIS measurements. The upper bound of >0.4 mg Chl m^{-3} is not necessary if effects of pigment packaging are accounted for. In the Behrenfeld et al. (2009) study, they show that by correcting for pigment packaging and NPQ, a first order linear relationship exists between chlorophyll and fluorescence yields across chlorophyll concentrations ranging from ~ 0.03 to > 2 mg Chl m^{-3} (see their figure 2). Moreover, if the detection limit truly is 0.1 mg Chl m^{-3} , then calculated fluorescence yields below this threshold would be random (it would look like speckling in a global map), but in the fore-stated 2009 publication it is clear that this is not the case and instead there are very coherent spatial patterns in observed yields across the central ocean gyres.

R1_24: Our lower bound was not based on the pre-launch detection limit of Letelier and Abbott (1996), who estimated this at 0.5 mg Chl m^{-3} , but on Huot et al. (2013). They calculated the detection limit in two ways:

- Firstly, by using the same approach as Letelier and Abbott (1996) but with actual MODIS-Aqua observations. This produced a detection limit between 0.035 and 0.14 mg chl m^{-3} .
- Secondly, by taking the mean and standard deviation of 273,383 groups of 25 adjacent pixels in satellite images. Setting a detection limit requirement of a signal: noise ratio greater than 2 for at least 50% of the points, they found a detection limit of 0.1 mg chl m^{-3} . A similar value for the detection limit was also estimated by Hu et al. (2012).

In addition to fluorescence signals, chlorophyll-a concentration estimates themselves also become less accurate at very low values. For example, in an extensive field-satellite matchup through the North and South Atlantic gyres, Brewin et al. (2016) found that at very low chlorophyll concentrations (<0.05 mg m^{-3}), satellite chlorophyll-a could be 50% of the observed value. Therefore, together, normalization of below-detection-limit fluorescence values to chlorophyll-a concentrations potentially inaccurate by 50% could lead to important inaccuracies in the resultant data fields not related to phytoplankton physiology.

Finally, we note that the field data do not exist to validate either, (i) the irradiance-passive fluorescence response (i.e., Fig. 3c) in the cores of the oligotrophic subtropical gyres where chlorophyll-a reaches such low values, or (ii) the nutrient dependence of potentially (but still an unknown) light-saturated values. We note that the exclusion of these values has no impact on our analysis, which focusses on the tropical Pacific region hosting chlorophyll-a concentrations > 0.1 mg m^{-3} .

References

Brewin, R.J., Dall'Olmo, G., Pardo, S., van Dongen-Vogels, V. and Boss, E.S., 2016. Underway spectrophotometry along the Atlantic Meridional Transect reveals high performance in satellite chlorophyll retrievals. *Remote sensing of environment*, 183, pp.82-97.

- Hu, C., Feng, L., Lee, Z., Davis, C.O., Mannino, A., McClain, C.R. and Franz, B.A., 2012. Dynamic range and sensitivity requirements of satellite ocean color sensors: learning from the past. *Applied Optics*, 51(25), pp.6045-6062.
- Huot, Y., Franz, B.A. and Fradette, M., 2013. Estimating variability in the quantum yield of Sun-induced chlorophyll fluorescence: A global analysis of oceanic waters. *Remote Sens. Environ.* **132**, 238-253.
- Letelier, R., & Abbott, M. R. (1996). An analysis of chlorophyll fluorescence algorithms for the moderate resolution imaging spectrometer (MODIS). *Remote Sensing of Environment*, 58, 215–223.

9. (lines 830-855) (a) I do not understand the justification for stating that the rate constant for fluorescence loss would not be expected to change at night under iron limiting conditions. If the PQ pool is highly reduced under these conditions (and there is strong published evidence that this is the case), then a back transfer of electrons to Qa will result in an increase in F_o (i.e., an enhanced k_f during the initial FRR flashes). In the absence of disconnected antennae complexes, this would yield a decrease in F_v (as observed), while F_m would not be impacted by iron stress (which seems to be the case in iron/macronutrient co-limited cells).

R1_25: In response to both these questions and those from Reviewer 3 (see below), we now include a fuller formal description of the derivation and interpretation of the active fluorescence parameters. Before providing this, we note a number of key points: Firstly, the previous interpretation of changes in F_o , F_m and F_v by Behrenfeld and colleagues as well as others did not account for the expected (and demonstrable) effect that the marked changes in σ_{PSII} over the diel cycles must be having on all of F_o , F_m and F_v . A detailed mathematical treatment is included below, but the key point is that the changes in σ_{PSII} are, by definition, indicating that the amount of excitation energy being delivered to PSII is varying. As such the corresponding absolute fluorescence yields must, to first order, vary proportionally (Oxborough et al. 2012). Secondly, the interpretation of ‘back transfer’ from a reduced PQ pool to Qa is, as far as we are aware (e.g., Behrenfeld and Milligan 2013), solely based on analysis of the very types of active chlorophyll fluorescence we (and Behrenfeld’s group) are presenting. Instead, our interpretation would suggest that this is at most a minor component of the ‘ F_v ’ signal at night (noting notation issues outlined below). If there is independent evidence that such back transfer occurs, we would be very grateful if the reviewer would direct us to this. In the meantime, we believe the interpretation we provide is both more fully supported by theory and more parsimonious. Finally, we note that the main conclusions of our manuscript are not dependent on detailed interpretation of these diel signals. However, in providing a thorough treatment we hope to correct some previous issues with interpretation of such data in the region which have only recently become apparent as the theoretical treatment has been advanced (Oxborough et al. 2012).

To provide a formal treatment, starting from first principles (see, e.g., Kolber et al. 1998; Kramer et al. 2004; Oxborough et al. 2012) we can write:

$$F_o \propto \frac{k_f}{k_p + k_f + k_d} \sigma_{LHII}[RCII]$$

and:

$$F_m \propto \frac{k_f}{k_f + k_d} \sigma_{LHII}[RCII]$$

Where F_o and F_m are the minimum and maximum fluorescence values measured, k_f , k_p and k_d are the intrinsic rate constants for fluorescence, photochemistry and non-radiative decay respectively, $[RCII]$ is the concentration of RCII within the measured volume and σ_{LHII} is the average absorption cross section of the light harvesting system of all the RCII.

Further taking (Kolber et al. 1998; Oxborough et al. 2021):

$$\sigma_{PSII} = \frac{k_p}{k_p + k_f + k_d} \sigma_{LHII}$$

where σ_{PSII} is absorption cross of PSII photochemistry (as measured using a single turnover active chlorophyll fluorescence technique such as Fast Repetition Rate fluorometry) and for simplicity neglecting the coefficient of proportionality, which will simply scale the actual measured value(s) of fluorescence for a given instrument, we have:

$$F_o = \frac{k_f}{k_p} \sigma_{PSII} [RCII]$$

and

$$F_m = \frac{k_f(k_p + k_f + k_d)}{(k_f + k_d)(k_p + k_f + k_d)} \sigma_{PSII} [RCII]$$

Thus:

$$F_v = F_m - F_o = \frac{1}{1 + k_d/k_f} \sigma_{PSII} [RCII]$$

And therefore:

$$\frac{F_v}{\sigma_{PSII}} = \frac{1}{1 + k_d/k_f} [RCII]$$

This demonstrates how a proportionality between F_v and σ_{PSII} is expected under conditions where neither k_d or k_f vary. We further note that these are the intrinsic rate constants for non-radiative and fluorescence decay respectively, i.e., they would not be expected to vary if, for example, a proportion of the RCII are shut.

Note, the terminology above is used under conditions where it is explicitly assumed that all of the reaction centres are open at F_o and closed at F_m .

Assuming now that a given measurement actually corresponded to a condition where a proportion of the reaction centres (C) were closed at the point of initiation of the measurement, i.e., where the measured minimal fluorescence $F^{(')} > F_o$ and thus the variable fluorescence should for clarity be denoted by a different symbol (usually F_q) to indicate the important difference in variable value and meaning (Genty et al., 1989; Kramer et al., 2004; Oxborough et al., 2012). Note further the prime notation $(')$ is usually added to indicate the measurements are made under the influence of background irradiance, which are normally the cause of closure of a proportion of the reaction centres, but this is clearly not the case if measuring at night, so it is neglected here.

Thus, if it were the case that a proportion (C) of the RCII were closed at night due to backreactions between a reduced PQ pool and the primary acceptor (Qa) of RCII we would have the measured variable fluorescence given by (Genty et al., 1989; Kramer et al., 2004; Oxborough et al., 2012):

$$F_q = F_m - F = F_m - (CF_m + (1 - C)F_o) = (1 - C)F_m - (1 - C)F_o = (1 - C)(F_m - F_o)$$

Due to the minimal fluorescence F resulting from the combined fluorescence from closed centres (C) fluorescing at F_m and the open centres (1-C) fluorescing at F_o , i.e., as $F = CF_m + (1 - C)F_o$

Thus, substituting in above we have:

$$F_q = (1 - C) \frac{1}{1 + k_d/k_f} \sigma_{PSII} [RCII]$$

and hence:

$$\frac{F_q}{\sigma_{PSII}} = (1 - C) \frac{1}{1 + k_d/k_f} [RCII] = (1 - C) \frac{F_v}{\sigma_{PSII}}$$

i.e., the variable fluorescence divided by the absorption cross of PSII photochemistry should drop in proportion to the fraction of closed centres (as an aside, note, rearrangement of above can provide the very well-known result $F_q/F_v = (1-C)$; e.g., Genty et al., 1989; Kramer et al., 2004). As no change in variable fluorescence normalised to the absorption cross of PSII photochemistry is actually observed at night (see Fig. 2f), the data actually indicate that there is unlikely to be significant closure of PSII at night due to back reactions between a reduced PQ pool and Qa.

References

- Genty, B., Briantais, J.M. and Baker, N.R., 1989. The relationship between the quantum yield of photosynthetic electron transport and quenching of chlorophyll fluorescence. *Biochim. Biophys. Acta - Gen. Subj.* **990**, 87-92.
- Kolber, Z.S., Prášil, O. and Falkowski, P.G., 1998. Measurements of variable chlorophyll fluorescence using fast repetition rate techniques: defining methodology and experimental protocols. *Biochim. Biophys. Acta - Bioenerg.* **1367**, 88-106.
- Kramer, D.M., Johnson, G., Kiirats, O. and Edwards, G.E., 2004. New fluorescence parameters for the determination of QA redox state and excitation energy fluxes. *Photosynth. Res.* **79**, 209-218.
- Oxborough, K., Moore, C.M., Suggett, D.J., Lawson, T., Chan, H.G. and Geider, R.J., 2012. Direct estimation of functional PSII reaction center concentration and PSII electron flux on a volume basis: a new approach to the analysis of Fast Repetition Rate fluorometry (FRRf) data. *Limnol. Oceanogr. Meth.* **10**, 142-154.

(b) A direct link between night time increases in fluorescence and disconnected antennae was earlier postulated by Behrenfeld and colleagues, with the specific idea that reduction of the PQ pool at night would signal for a state transition, but these transitions would not be complete (i.e., connection to PSI) in many cases if PSI levels are very low (as indicated in lab studies and during the current study).

R1_26: We agree that this remains a reasonable hypothesis, although, as we argued above, some decoupling from PSI and PSII is likely required as Fm/Chl increases at night (see also Macey et al. 2014).

Reference

Macey, A.I., Ryan-Keogh, T., Richier, S., Moore, C.M. and Bibby, T.S., 2014. Photosynthetic protein stoichiometry and photophysiology in the high latitude North Atlantic. *Limnol. Oceanogr.* **59**, 1853-1864.

(c) if the Fv change is largely or in part due to back transfers to Qa with a highly reduced PQ pool and the state transitions are triggered by the same reduced pool, then both Fv and sigma-PSII changes will be correlated but not necessarily causatively. Division of Fv by sigma-PSII will thus result in a product that exhibits a dampened correlation with Fv/Fm. It is interesting that covariations in Fv/Fm and sigma-PSII were also often reported by the Behrenfeld group, but they noted that this was not always the case – as shown in the currently cited Behrenfeld and Kolber (1999) paper in Science.

R1_27: See above (R1_25). The absolute values of Fv and σ_{PSII} are expected to be related from first principles and there is, as far as we are aware, no other evidence for back transfer of electrons from the PQ pool acting to close (i.e., reduce Qa). Although there is evidence for changes in downstream electron transport rates (Behrenfeld and Milligan 2013), this still does not provide any direct evidence for closure of PSII due to a reduced PQ pool. Rather, as indicated above, such an effect has been inferred on basis of previous interpretations of Fv (and Fo) changes which did not fully consider the implications of the changes in σ_{PSII} potentially generated by the state-transitions (and in general energetic decoupling) referenced by the reviewer.

Reference

Behrenfeld, M.J., Worthington, K., Sherrell, R.M., Chavez, F.P., Strutton, P., McPhaden, M. and Shea, D.M., 2006. Controls on tropical Pacific Ocean productivity revealed through nutrient stress diagnostics. *Nature* **442**, 1025-1028.

Referee #2 (Remarks to the Author):

This paper reports ground-truthing measurements of increases in chlorophyll fluorescence of marine phytoplankton communities induced by iron limitation in a transect from low iron, high nitrate (HNLC) waters of the equatorial Pacific to low nitrate, low iron waters farther north. The authors were able to confirm iron limitation in the equatorial Pacific through a combination of iron and nitrate concentration measurements, increases chlorophyll concentrations with iron addition in bottle incubation experiments, molecular biomarkers for iron and nitrogen limitation, and measurement of chlorophyll (Chl) fluorescence and ambient Chl. Most importantly, they showed that the HNLC waters near the equator had high ratios of Chl fluorescence to Chl, something that had been previously well documented in laboratory and field studies, including those in the equatorial Pacific. They further showed that fluorescence to Chl ratios could be measured remotely in response to daytime solar radiation, initially using a ship mounted fluorometer and discrete Chl measurements in samples, but also from archived satellite-based measurements of both chlorophyll and Chl fluorescence collected over an 18-year period in the equatorial Pacific. The data showed that iron limitation in the equatorial Pacific decreased with upwelling strength, and attendant decreases in seawater temperature as a result of the upwelling of colder, iron- and nitrogen-rich deeper waters. Fluorescence-based values for iron limitation then increased with increasing temperature away from upwelling centers as the upwelled water warmed and iron and nitrogen concentrations were drawn down by the growth of the emerging phytoplankton community. The resulting relationship between increasing sea-surface temperature and fluorescence:Chl ratios determined from the satellite data were significant and invariant over many El Niño–Southern Oscillation (ENSO) cycles, but differed by twofold from values predicted from a biogeochemical model for chlorophyll and marine productivity currently used to predict future effects of climate change. These are important results because they suggest that existing and future satellite data can be used to map iron limitation world-wide in the ocean and that models for marine chlorophyll and productivity currently used to predict future effects of Climate Change likely need modification.

We thank the Reviewer for their evaluation and useful comments. Specific responses are provided below.

Specific comments

Lines 118-122 – Yes, these diel changes in phytoplankton pigment fluorescent patterns are well documented for the transition from the iron limited equatorial Pacific to the N-limited North Pacific central gyre.

R2_1: We agree with the reviewer. Although we believe some important new insights were gained from our analysis, in response to feedback from Reviewer 1 and 4, we have now reduced discussion of diel variability in active fluorescence signals in the main text to focus on the main novel aspects of our study.

Lines 131-160 – The lack of detection of PSI and cyt b6/f Prochlorococcus proteins at the iron limited stations 1-3 is problematic as the cyanobacteria would need these proteins for linear electron flow between PSII and flavodoxin, needed for the production of reductant (NADPH) used for the fixation of carbon and the reduction of nitrate to ammonium. Note that PSII proteins and flavodoxin were detected at these stations, suggesting that PSI and the b6/f complexes must have also been present. A possible explanation for this discrepancy is that PSI and b6/f complexes were present in these Prochlorococcus communities, but that their proteins were different enough in these low-iron adapted populations from the protein

“standards” used for identification, that they were not identified. The use of appropriate “standards” for protein identification is a common problem with proteomics.

R2_2: In order to assess for lack of detection of strain-specific PSI and cytochrome b6f (but not PSII) we performed an analysis that is outlined in the Supplementary Information (lines in the original manuscript: 678–685), which is repeated below here:

To further investigate if the low abundance/absence of photosystem I and cytochrome b₆-f proteins found in the high nitrate, Fe limited region (Sites 1–3) were due to strain-level differences in sequences that could have restricted their detection relative to low nitrate strains, we performed a further peptide analysis using METATRYP V2.0 (<https://metatryp.whoj.edu>; Ref. 62) to assess for Prochlorococcus strain matches for peptide sequences (Table S3). This analysis showed that all peptides matched to Prochlorococcus strains isolated in the Equatorial Pacific, in addition to strains isolated in other regions.

In general, and in line with the reviewer, we expect the lack of detection implies that they are at sufficiently low abundance not to be observed by this technique and not that there is none at all at the Fe limited sites.

Lines 167-169 – How rapid was this decrease? The data in figure 3a are for the samples taken at the end of the incubation experiments. However, time course data in other experiments conducted in the Equatorial Pacific iron-limited region indicate that such decreases in fluorescence/Chl occur within one day’s time (see Behrenfeld and Milligan 2013). This paper and those cited therein should be mentioned here.

R2_3: This decrease was in the iron addition bioassay experiments that were conducted over ~44-45 hours duration (now stated in the manuscript), but we agree that changes were likely operating on even faster timescales. We note that previous studies have focussed observations on changes in Fv/Fm in response to iron addition, whereas we focus on fluorescence per unit chlorophyll (as this is ultimately what the satellite observes). We nevertheless have now also referenced the suggested review paper at this point.

Lines 180-181 – There is substantial evidence that the highly fluorescent light harvesting complexes under conditions of iron limitation and high available nitrogen (i.e., HNLC conditions) result not just from LNCs that are “weakly connected to reaction centers” but rather are disconnected to photosynthetic reaction centers (again see Behrenfeld and Milligan 2013).

R2_4: Now rephrased to state ‘weak or no energetic connectivity’.

Lines 189 and 190 – This has also been shown previously (again see Behrenfeld and Milligan 2013 and references cited therein), which should be cited here.

R2_5: At this point we are specifically refereeing to chlorophyll-a normalized active fluorescence, which has to our knowledge not been demonstrated (note the Behrenfeld studies show Fv/Fm and other active fluorescence signals).

William Sunda

Referee #3 (Remarks to the Author):

The manuscript reports changes in the chlorophyll-fluorescence characteristics of phytoplankton populations along an oceanic transect. This transect transitions between distinct zones of nutrient limitation. The manuscript correlates measured physiology to changes in the satellite fluorescence record to derive modifications to climate models.

I have been asked to comment on the in situ photo-physiology aspects of this manuscript and I hope the authors find these comments helpful in the revision of the manuscript.

As mentioned in the manuscript, this is not the first report of changes associated with in situ photo-physiology as one moves north from the equatorial Pacific. I think the manuscript does a good job acknowledging the previous observations and interpretations. The manuscript also does a very good job established the nature of nutrient limitation in the different sampling locations.

The novelty associated with this work is the addition of proteomics to mechanistically explain what was previously hypothesized.

We thank the Reviewer for their evaluation and useful comments. Specific responses are provided below.

I have two issues with the manuscript that could be addressed by the authors.

First of all, the manuscript focusses on a derived fluorescence parameter, F_v , which is variable fluorescence. F_v is not directly measured by a fluorometer, but instead it is calculated by subtracting the minimal level of fluorescence (F_o) from the fluorescence maximum (F_m) measured by a fast repetition rate fluorometer. The manuscript goes through some effort to explain why it uses this parameter, but I am confused as to why this decision was made. Ultimately, disconnecting of light harvesting antenna from the photosystems causes an increase in F_o and this is what drives changes in the variable fluorescence parameter (F_v). This is well established in the literature. So, the authors should consider changing the approach to use F_o throughout this work. Supplementary discussion 1 includes a calculation derived from Oxborough et al. (2012). I note they use some different symbols, which is confusing. Please provide more detail about how this equation was derived. I was unable to convert the approach of Oxborough to what is reported here so there appear to be some missing steps or assumptions in this manuscript.

R3_1: Unfortunately, there are some misconceptions in these assertions and we refer the reviewer to Response R1_25 above for a fuller formal treatment. We apologise for the lack of a formal derivation in the previous manuscript and have now provided this, plus have changed our notation to be consistent with Oxborough et al. (use of k_d rather than k_h for the rate constant for non-radiative decay). We note that we actually use interpretations of both F_o , F_m and F_v in our analysis. We do so on the basis of first principle derivations (Genty, 1989; Kramer et al., 2004; Oxborough et al., 2012). In relation to the specific point raised by the reviewer here, it is clear that both F_o and F_m are increasing as a result of Fe stress, which can, as we argue and the reviewer points out, only be interpreted in terms of energetically decoupled pigment-protein complexes (Macey et al. 2014). We further note that we present the light-saturated active fluorescence (i.e., F_m) as it is most comparable to that of light-saturated passive fluorescence, which is a major focus of this study. We note that chlorophyll-normalized F_m shows marked, ~3-fold changes decrease between both Fe

limited and N limited sites and in response to Fe addition at Fe limited sites, strongly suggesting that it is very much impacted by nutrient limitation (in addition to Fo and Fv).

References

- Genty, B., Briantais, J.M. and Baker, N.R., 1989. The relationship between the quantum yield of photosynthetic electron transport and quenching of chlorophyll fluorescence. *Biochim. Biophys. Acta - Gen. Subj.* **990**, 87-92.
- Kramer, D.M., Johnson, G., Kiirats, O. and Edwards, G.E., 2004. New fluorescence parameters for the determination of QA redox state and excitation energy fluxes. *Photosynth. Res.* **79**, 209-218.
- Oxborough, K., Moore, C.M., Suggett, D.J., Lawson, T., Chan, H.G. and Geider, R.J., 2012. Direct estimation of functional PSII reaction center concentration and PSII electron flux on a volume basis: a new approach to the analysis of Fast Repetition Rate fluorometry (FRRf) data. *Limnol. Oceanogr. Meth.* **10**, 142-154.

A key argument associated with the in-situ physiology is the detachment of light harvesting antenna from the photosystems. The manuscript convincingly shows a change in the ratio of photosystem II to photosystem I core proteins via proteomics. However, were the light harvesting antenna not observed in this data set? Knowing the relative abundance changes in these proteins (or if they were even detected) would go a long way to supporting the hypotheses associated with the fluorescence data. Either way, the data shows changes in the proteome of *Prochlorococcus* but the interpretation of this data is based around a wide variety of organisms that have different light harvesting systems from *Prochlorococcus* including cyanobacteria with phycobilisomes (ref 32,37) or even eukaryotes (ref 31,36). This is not appropriate. Finally, the manuscript should not use the acronym LHC as it refers to a specific family of chlorophyll a binding proteins from eukaryotes.

R3_2: We agree with the reviewer that it would have been very useful to show changes in the detached pigment proteins from the proteomics analysis. We indeed searched but could not find these in the proteomics dataset, which could either be due to the protein being at relatively low abundance or due to an annotation issue. We also agree, that whilst it would be desirable to use metaproteomics to do a similar analysis for all of the most abundant phytoplankton types present, the methodology for field samples is not there yet; hence, we focus on one dominant phytoplankton type (i.e., *Prochlorococcus*). We are cautious in the manuscript to note the difference between community level signals in fluorescence and *Prochlorococcus*-specific protein changes. Finally, 'LHC' has now been replaced by the more general term 'pigment protein complexes'.

Referee #4 (Remarks to the Author):

Browning et al present multiple data types from a cruise in the equatorial Pacific aimed at understanding the physiological impacts of nutrient limitation on phytoplankton physiology and optical properties and using constructs from that data to more tightly constrain NPP projections in the context of climate change. The set of experimental measurements presented are uniquely comprehensive and well-integrated, making this study particularly valuable as a synoptic perspective. For example, strikingly clear evidence of nutrient limitation was observed using meta-proteomics approaches and the conclusions from that analysis are also supported by shipboard incubation experiments in which nutrients were added and the chlorophyll response was measured. The comprehensiveness and virtuosity of this work are important because they leave little to doubt about the physiological state of the phytoplankton, so that the association with corresponding optical measurements is sound. This is beautiful work.

We thank the Reviewer for their positive comments. We respond to each specific comment in turn below.

A central self-reported unique contribution from this work is utilizing radiometers on the vessel to derive a relative measurement of sunlight-stimulated chlorophyll fluorescence, and showing that a relative measure of SSCF can be obtained from satellite data, using the assumption of irradiance independence that is supported by data shown in Fig. 3C. The authors then go on to use these constructs to produce global maps of satellite-derived Fe limitation. A number of studies, which are cited, previously have applied remotely sensed optical signatures to predict nutrient limitation and primary productivity in this region of the ocean. Recommendations follow. I like this paper, but I think the message might need to be distilled to its essence.

R4_1: We thank the Reviewer for their recommendations – in addition to comments from other reviewers, we recognize that our main novel message required distilling and we have worked to do this in the revised manuscript.

1. Perhaps a bit simplistic, but isn't it to be expected that iron limitation will be somewhat binary, in that the phytoplankton community either doesn't have enough, or they do. Could the global model then still be right (i.e. not be overestimating NPP sensitivity to Fe input by 2 fold), in the sense that NPP is changing with changing Fe input but not the Fe limitation status at the cellular level? Towards the end of the paper this view is represented but towards the beginning the emphasis seems to be on the model underestimates.

R4_2: This is a good point made by the Reviewer; however, we think that the model should still be able to reflect relative levels of Fe limitation through the model Fe limitation term (in the manuscript methods referred to as L_F). This term is variable between 0-1 depending on Fe availability (intracellular availability, relative to minimum and maximum phytoplankton quotas). This term is then multiplied by the temperature/light regulated maximum phytoplankton growth rate*, which therefore directly (and dynamically) regulates modelled NPP.

**Provided it remains lower than any other nutrient limitation term for other model nutrients, which is the case in the Equatorial Pacific.*

2. While I was impressed with the quality of this manuscript I was left uncertain about observations that confirm previous reports and observations that are new with respect to

satellite detection. I recommend the authors clarify. Specifically, the authors should briefly contrast their approach and findings with previous studies that predicted Fe limitation from satellite data. My impression is this manuscript uses satellite data differently, and is different in emphasizing ENSO variation and its relatively low impact on Fe limitation, and suggesting that models overstate ENSO impacts on Fe limitation. But, is this insight a direct consequence of the new relative measure of SSCF?

R4_3: We have now included a statement about how our field validation of the satellite fluorescence observations, with respect to irradiance and nutrient limitation state in the context of phytoplankton community structure and light absorption properties, now makes interpretation of these signals tractable for the tropical Pacific region. Our finding is that ENSO variability drives major variations in Fe limitation, but these are not so large as the model predicts (see response R4_4 below for more discussion on this). This finding is a direct consequence of validating the satellite data, as field observations are far too scarce to generate reliable time series at any scale (intra-, interannual, or decadal). We also moderately expand our finding that the satellite data showed no evidence for transitions into Equatorial Pacific nitrogen limitation during the El Niño events in the MODIS record, but the capacity now exists to observe this into the future (including responses to stronger El Niño events and ongoing climate change impacts).

3. Line 255, and elsewhere. Central to the findings is the observation that nutrient limitation is resilient to fluctuations in nutrient input caused by ENSO. At various points in the manuscript there is a suggestion that phytoplankton community restructuring could explain this resilience, but the reasoning behind this is unclear. The data in the manuscript don't resolve phytoplankton diversity in great detail, except for the metaproteomics, which are very specific. Are the authors suggesting that taxa adapted to a continuous state of iron limitation inhabit iron-limited ocean regions? My interpretation of the manuscript is that the change in expression of proteins is a reflection of the physiological state of the cells, but elsewhere it sounds like the authors are invoking community turnover. To avoid confusion perhaps be more clear about how community turnover (i.e. changes in communities) could decouple nutrient input from NPP.

R4_4: We have now clarified this in the manuscript by stating that changes in community structure could buffer against the strength of Fe limitation because of variable (i) Fe requirements, and (ii) Fe uptake abilities. Both of these are likely highly simplistic in the model for the two model phytoplankton types, nanophytoplankton and diatoms, in comparison to the real ocean. We therefore predict that NPP will be maintained to a greater extent in the real ocean in response to lower Fe availability than in the model, due to the possibility for selection from a far more diverse range of phytoplankton with different Fe quotas and uptake mechanisms. A simple, and non-exclusive, example could be a shift to progressively smaller picophytoplankton in the ocean with higher surface area to volume ratios for Fe uptake, whereas the model can only switch from diatoms to nanophytoplankton.

4. Line 796, L Lowercase F and (Line 792), L lowercase Fe. Is that a typo? If not clarify difference. I suggest defining Delta Fe lowercase lim, y axis label Fig. 4f, in legend since its easy for a reader to become confused.

R4_5: We agree this was easy for the reader to become confused. L_{Fe} is the model Fe limitation term, ranging from 0-1, with lower values indicating stronger Fe limitation (i.e., phytoplankton maximum growth rate is multiplied by a smaller value). In Figures 4e,f and S8 we convert this to another Fe limitation term, Fe_{Lim} , where $Fe_{Lim} = 1 - L_{Fe}$, so that higher

values indicate stronger Fe limitation. This has now been clarified in the model methods section.

5. If more space is needed to strengthen the narrative then I suggest reducing the section about diel cycles. Its interesting but not essential to the central arguments.

R4_6: We agree with the reviewer, and have indeed shortened the manuscript including significantly (by ca. 500 words) by reducing the discussion of diel active fluorescence cycles and details of phytoplankton responses in the bioassay experiments. We agree that this has strengthened the narrative of the paper.

Reviewer Reports on the First Revision:

Referees' comments:

Referee #1 (Remarks to the Author):

Once again I would like to congratulate the authors on a very nice study and well written manuscript. I would also like to thank the authors for taking the time to consider my earlier comments carefully and provide thoughtful responses. It is always appreciated by reviewers! In considering these responses and evaluating the revised manuscript, I now believe that the study does merit publication in Nature and represents a significant scientific advancement.

Referee #2 (Remarks to the Author):

This is an impressive paper.

The authors have adequately responded to my and the other reviewers' comments.

Referee #4 (Remarks to the Author):

The revised version of Brown et al. delivers essentially the same message but is considerably clearer and more focused than the first submission. The issues I raised in my original review are answered to my satisfaction in the response and in the manuscript. My contribution to reviewing his manuscript is mainly derived from my knowledge of cells, ecology and physiology, and proteomics measurements, extending to photosynthetic metabolism. Other reviewers of this manuscript have more expert knowledge of analyses of optical data, particularly satellite data, and I defer to those reviewers on those issues. The comments that follow are intended to be a brief narrative of my reactions to the lengthy response and not criticisms that require responses.

The unique conclusions of this proposal appear to be clearly delineated from previously published work and to result from choices of study design that led to a more complex and robust interpretation of optical data. The new findings reported appear to be significant and to impact our ability to estimate global ocean NPP.

The physiological interpretations were clear to me and consistent with typical physiological responses to environmental variation. My comments about community vs cellular adjustments in metabolism are addressed concisely and clearly in the revised manuscript, and the arguments in the revised manuscript consequently fit the rational narrative more compellingly. Although this manuscript is very concise, the relationships between the models and community structure, which may impact future research, are clear.

The proteomics contributions are de-emphasized but I am made more confident in the broad findings by the actual in situ measurements of cell physiological status, which support the interpretations made with much higher frequency measurements.

The answer to my question about the model and the binary nature of nutrient limitation clear and makes fundamental sense.

Minor issues I raised were easily addressed.

Author Rebuttals to First Revision:

Response to Reviewer comments

We thank the three Reviewers for their detailed comments. We have responded to these below in blue.

To refer between review responses, we have numbered reviewer comments (R1_1, for Reviewer 1, comment 1, and so on).

Referee #1 (Remarks to the Author):

The current manuscript adds important new observations from a ship transect in the equatorial Pacific to a substantial body of literature regarding nutrient stress in this region and diagnostic chlorophyll fluorescence patterns. The manuscript is well written and the observational data are impressive. To my knowledge, the diagnostic pigment results are novel, as is the presentation of the time series of chlorophyll and fluorescence quantum yields for the Nino 3 region of the equatorial Pacific. While I congratulate the authors on a very nice study, I did not find that sufficient new discoveries were presented to warrant publication in Nature and would encourage the authors to resubmit the manuscript to a more specialized journal.

The following is my basis for the above recommendation. The observed responses to N and Fe amendments shown in figure 1c are consistent with many similar measurements made in the past in the equatorial Pacific region. While the addition of cobalt, zinc, and silicic acid treatments has not been a common practice, these additions did not elicit significant responses by the phytoplankton during the current study. The nutrient stress biomarker results for Prochlorococcus shown in figure 1d are consistent with findings recently published in Science by Ustick et al. (2021). The diel fluorescence patterns (Fv/Fm, sigma-PSII) shown in figure 2a,b are essentially identical to those published for a much larger field data set in the same region in the cited Behrenfeld et al (2006) Nature paper and both the responses to iron additions (in terms of changes Fo, Fm, Fv/Fm, sigma-PSII) and the mechanistic interpretation of these patterns are essentially the same as earlier reported by Behrenfeld and colleagues in a series of papers (most of which are cited in the current manuscript). The changes in photosynthetic proteins shown in figure 2d are, to my knowledge, largely unique for field measurements, but are still very consistent with previous laboratory studies. The enhancement in fluorescence yields under iron limitation was also the basis for Behrenfeld et al's (2006) reassessment of satellite based primary production estimates for the equatorial Pacific and for the detection of iron stress from MODIS fluorescence quantum yield data in the cited Behrenfeld et al. (2009), which to first order is repeated here in figure 4. This earlier 2009 paper also reported a comparison between the satellite-detected iron stress regions and those predicted from an ecosystem model, much like the analysis currently presented in figure 4e. Thus again, while the current authors have presented a very nice analysis, I did not find that it represented a sufficient

breakthrough to justify publication in Nature, but rather provides confirmation and further interpretation of previous findings.

R1_1: We thank the Reviewer for their detailed consideration of our manuscript. The major contribution of our study is the quantification of two decades of shifting iron limitation in the Equatorial Pacific that is over estimated by a state-of-the-art global model. This raises new questions regarding the responses of marine ecology and productivity to changing iron stress. None of these insights have been realised by the prior studies mentioned by the reviewer (and cited in our manuscript) because they require the multi-disciplinary, coherent approach of our study - spanning ocean biogeochemistry, proteomics (both nutrient limitation and photophysiology), optics, remote sensing and ocean modelling. That being said, we agree that these core messages were diluted by discussion of issues not strictly pertinent. The manuscript has now undergone major revision to streamline our findings and their importance.

The observed responses to N and Fe amendments shown in figure 1c are consistent with many similar measurements made in the past in the equatorial Pacific region. While the addition of cobalt, zinc, and silicic acid treatments has not been a common practice, these additions did not elicit significant responses by the phytoplankton during the current study.

R1_2: We agree the experiments confirm the nutrient limitation regime, but do not add substantially beyond what we knew already (i.e., systems limited by Fe versus N). However, conducting and presenting the results of these experiments were fundamental to the bigger objective of the project, as they are currently the only means to assess nutrient limitation of the overall phytoplankton community. Therefore, while we believe it was imperative to conduct these, detailed discussion of the results were probably not needed and this part of the revised manuscript has been modified to simply state the key experimental findings.

Confirmation of the absence of limitation by other nutrients was also something we believed was important to show to make the claim of iron versus nitrogen limitation (as the Reviewer points out, such experimental tests have not been conducted in previous experiments in the systematic way we did here).

As an additional note, we point out that the resolution of community structure by both pigment analysis and flow cytometry was also not conducted in most previous experiments where active chlorophyll fluorescence signals were being evaluated (i.e., Behrenfeld et al., 2006), but are crucial in confidently interpreting the active fluorescence measurements in terms of nutrient limitation (Proctor and Roesler, 2010; Suggett et al., 2009).

References

Proctor, C.W. and Roesler, C.S., 2010. New insights on obtaining phytoplankton concentration and composition from in situ multispectral Chlorophyll fluorescence. *Limnol. Oceanogr. Meth.* **8**, 695-708.

Suggett, D.J., Moore, C.M., Hickman, A.E. and Geider, R.J., 2009. Interpretation of fast repetition rate (FRR) fluorescence: signatures of phytoplankton community structure versus physiological state. *Marine Ecology Progress Series*, **376**, pp.1-19.

The nutrient stress biomarker results for *Prochlorococcus* shown in figure 1d are consistent with findings recently published in *Science* by Ustick et al. (2021).

R1_3: We agree, although the Ustick study focussed on metagenomic data, rather than the measurements of metaproteomics in our work. We reiterate that our major finding is not to demonstrate the Fe-N limitation transition, but to explore and quantify climate variations in nutrient limitation. In many respects, the more directly relevant prior work is the nutrient stress biomarker study of Saito et al. (2013). However, whilst very much breakthroughs in their own right, neither Ustick et al. (2021) nor Saito et al. (2013) conducted complimentary nutrient addition bioassay experiments or any type of fluorescence measurement. The latter are both central to the main objective of this work.

References

Saito, M.A., McIlvin, M.R., Moran, D.M., Goepfert, T.J., DiTullio, G.R., Post, A.F. and Lamborg, C.H., 2014. Multiple nutrient stresses at intersecting Pacific Ocean biomes detected by protein biomarkers. *Science* **345**, 1173-1177.

The diel fluorescence patterns (Fv/Fm, sigma-PSII) shown in figure 2a,b are essentially identical to those published for a much larger field data set in the same region in the cited Behrenfeld et al (2006) *Nature* paper and both the responses to iron additions (in terms of changes Fo, Fm, Fv/Fm, sigma-PSII) and the mechanistic interpretation of these patterns are essentially the same as earlier reported by Behrenfeld and colleagues in a series of papers (most of which are cited in the current manuscript).

R1_4: We fully agree that the diel signals in active fluorescence measurements, Fv/Fm and sigma-PSII, are basically the same as in Behrenfeld et al. (2006). We point out that chlorophyll-normalized active fluorescence measurements were not reported in Behrenfeld

et al. (2006 or others), but are a key stepping stone towards chlorophyll-a normalized *passive* fluorescence measurements.

As a minor aside, we also note that our interpretation of diel cycles in Fv/Fm and sigma-PSII differed somewhat to Behrenfeld et al. (2006) as shown by our extended analysis of the data (e.g., Fig. 2e–g, Supplementary Discussion 1, see also responses to other reviewers below). However, in the revised manuscript we have now streamlined this section of the main text to more quickly come to the main point about relative chlorophyll fluorescence yields, with these details provided in the supplementary text.

The changes in photosynthetic proteins shown in figure 2d are, to my knowledge, largely unique for field measurements, but are still very consistent with previous laboratory studies.

R1_5: Similar results have indeed been found for a restricted set of phytoplankton cultures (reviewed in e.g., Behrenfeld and Milligan, 2013). But, as the reviewer suggests, the changes in relative abundances of PSII, PSI, and cytochrome b6f are, to our knowledge, unique for field measurements. We feel this is an important advance, although again, on its own not the major novelty of our work. Specifically, these protein observations feed into a much stronger interpretation of the fluorescence data (and then onto satellite observations of nutrient limitation).

The enhancement in fluorescence yields under iron limitation was also the basis for Behrenfeld et al's (2006) reassessment of satellite based primary production estimates for the equatorial Pacific and for the detection of iron stress from MODIS fluorescence quantum yield data in the cited Behrenfeld et al. (2009), which to first order is repeated here in figure 4.

R1_6: We list below several key factors that we believe are critical in distinguishing our contribution from important prior works by Behrenfeld et al (2006, 2009). We consider these to be fundamental to meaningful use of the satellite fluorescence data:

- We diagnose the nutrient limitation regime via a unique combination of phytoplankton biomass changes in experiments and protein biomarkers (neither of these were conducted in Behrenfeld et al. (2006) and not conducted together in any prior study to our knowledge).
- We demonstrate changes in active fluorescence signals following short-term nutrient addition were independent of phytoplankton community structure shifts.
- We show that sunlight-stimulated, *passive* fluorescence (unlike other studies that have used *active* fluorescence only, which is not what the satellite observes and could potentially lead to major differences; Cullen et al., 1988; Cullen and Lewis, 1995; Huot and Babin, 2010) is light saturated at irradiances below that of satellite chlorophyll fluorescence retrievals. We have only demonstrated this for the tropical

Pacific (measurements elsewhere not conducted), and this is one reason why we focus our remote sensing study on this region.

- We show that changes in this light-saturated, sunlight-stimulated chlorophyll-normalized fluorescence matches up with the iron-to-nitrogen limitation transition.
- We document that the fold change in (i) active fluorescence between iron and nitrogen limited sites, (ii) active fluorescence following experimental iron addition at iron limited sites, (iii) shipboard, sunlight-stimulated *passive* chlorophyll-normalized fluorescence under light saturation between iron and nitrogen limited sites, and finally (iv) the satellite-derived passive fluorescence between iron and nitrogen limited sites, are all consistently around a value of 3. Collectively, this strong quantitative consistency convincingly supports nutrient regulation of this signal (and specifically N versus Fe limitation).
- Following this extensive field validation (never before conducted), we are able to quantify variability in satellite-derived tropical Pacific nutrient limitation over 20 years for the first time and compare this to a model with realistic ENSO dynamics. Key findings from this are: (i) iron limitation varies with upwelling strength and that even under the weakest upwelling on record, the equatorial system remains iron (rather than nitrogen) limited. (ii) changes in the intensity of iron limitation following physical forcing over ENSO cycles are overestimated by the ocean biogeochemical model.

References

- Behrenfeld, M.J., Worthington, K., Sherrell, R.M., Chavez, F.P., Strutton, P., McPhaden, M. and Shea, D.M., 2006. Controls on tropical Pacific Ocean productivity revealed through nutrient stress diagnostics. *Nature* **442**, 1025-1028.
- Behrenfeld, M.J., Westberry, T.K., Boss, E.S., O'Malley, R.T., Siegel, D.A., Wiggert, J.D., Franz, B.A., McLain, C.R., Feldman, G.C., Doney, S.C. and Moore, J.K., 2009. Satellite-detected fluorescence reveals global physiology of ocean phytoplankton. *Biogeosciences* **6**, 779–794.
- Cullen, J.J., Yentsch, C.M., Cucci, T.L. and MacIntyre, H.L., 1988, August. Autofluorescence and other optical properties as tools in biological oceanography. In *Ocean optics IX* (925, 149-156). International Society for Optics and Photonics.
- Cullen, J.J. and Lewis, M.R., 1995. Biological processes and optical measurements near the sea surface: Some issues relevant to remote sensing. *J. Geophys. Res. Oceans* **100**, 13255-13266.
- Huot, Y. and Babin, M., 2010. Overview of fluorescence protocols: theory, basic concepts, and practice. In *Chlorophyll a Fluorescence in Aquatic Sciences: Methods and Applications* (pp. 31-74). Springer, Dordrecht.

This earlier 2009 paper also reported a comparison between the satellite-detected iron stress regions and those predicted from an ecosystem model, much like the analysis currently presented in figure 4e.

R1_7: Whilst the spatial comparison in Figure 4e is similar to the approach used in Behrenfeld et al. (2009), the focus of our analysis (time series analysis, model comparison) is very different.

Thus again, while the current authors have presented a very nice analysis, I did not find that it represented a sufficient breakthrough to justify publication in Nature, but rather provides confirmation and further interpretation of previous findings.

Detailed comments (in order of appearance):

1. (lines 162-163) I would suggest that some additional analysis be conducted to evaluate if it is Fm that is particularly increased by iron stress or Fo. My guess is that it is the latter in terms of what is driving changes in Fv.

R1_8: We find that both Fo/Chl and Fm/Chl increase; this is a clear indicator that the majority of the differences in the fluorescence yields are due to energetically decoupled pigment protein complexes (see e.g., Macey et al. 2014). As the reviewer predicts, in the iron limited zone values of Fo/Chl between non-Fe amended and Fe-amended treatments show a mean fold difference of 4.2, whilst Fm/Chl shows a mean fold difference of 3.0, i.e., around 30% lower, however this still indicates that >70% of the difference (i.e., the majority) is due to simultaneous changes in both Fo and Fm. Moreover, the light-saturated *active* fluorescence (i.e., Fm) is most comparable to that of light-saturated *passive* fluorescence, which is why here it is most meaningful to present the Fm values. More detailed discussion on the interpretation of Fv is included in the responses to the latter specific questions below.

Reference

Macey, A.I., Ryan-Keogh, T., Richier, S., Moore, C.M. and Bibby, T.S., 2014. Photosynthetic protein stoichiometry and photophysiology in the high latitude North Atlantic. *Limnol. Oceanogr.* **59**, 1853-1864.

2. (Figure 1) (a) a bit more detail on what is meant by 'assumed-average phytoplankton requirements' would be helpful here.

R1_9: The phrase 'assumed average phytoplankton nutrient requirements' was used as the value comes from an average of a compilation of values (Moore et al., 2013), which we assume is reflective of an average across multiple phytoplankton types and growth conditions. In the revised manuscript, the value used has been added in (N:Fe of 2132 mol:mol).

(b) panel e – the response of chlorophyll b to N addition seems to be similar between station 2 and 5, why might this be?

R1_10: We cannot identify the similarity: specifically, in Experiment 2, the normalized chlorophyll-b concentrations are:

N addition (with or without added Co): 0.51

Fe addition (with or without added Co): 0.73-0.81

N+Fe addition (with or without Co, Zn or silicic acid): 0.71-1

Conversely, in Experiment 5, the normalized chlorophyll-b concentrations are:

N addition (with or without added Co): 0.27-0.41

Fe addition (with or without added Co): 0.16-0.2

N+Fe addition (with or without Co, Zn or silicic acid): 0.87-1

Therefore, the trends in chlorophyll-b closely follow that of overall chlorophyll-a, responding most to Fe addition in Experiment 2 (with or without N) and a small amount to N addition but most to N+Fe addition in Experiment 5.

(c) Why is the response at station 5 different between N and NCo? If anything, I would expect the latter to exceed the former but the observations indicate the opposite.

R1_11: We cannot identify the difference: Experiment 5 shows almost identical chlorophyll-a response to N or N+Co (see the overlapping blue points in Fig. 3c).

3. (Figure 2) (a) panel c – the term 'isi' is often specific in the literature to iron stress induced compounds in cyanobacteria, but the current study is conducted on mixed prokaryotic and eukaryotic species, so it might be better to use a different term than isi (?).

R1_12: As 'isi' has consistently been used as 'iron stress induced' in the literature we do not see why it should not be used in a general sense here (i.e., overall phytoplankton community).

(b) With respect to panel c, it is shown that the abundance of PSII is not down regulated under iron stress relative to other components of the linear electron transport chain (LETC) (and this is also stated in the main text), but readers might appreciate some interpretation as to why. In earlier studies on light-limited phytoplankton (e.g., Sukinec et al 1987), all elements of LETC are down regulated in parallel.

R1_13: We expect this to result from PSII having a relatively low Fe requirement. This is similar to other laboratory studies; e.g., Strzepek and Harrison (2004) showed that PSII abundance changed relatively little in response to low Fe conditions in comparison to PSI, which showed large decreases.

4. (Figure 3) (a) the emphasis in panel b is on fluorescence yield changes at night, while the figure is building up to the use of satellite fluorescence data to detect iron stress. In panel b it is not so clear that the yield per unit chlorophyll will be much different between the Fe limited and N limited regions in the middle of the day when satellite fluorescence measurements are collected. This expectation is realized in panel d where the difference between regions is rather small (order 0.1 units).

R1_14: The differences in chlorophyll-normalized active fluorescence between the two nutrient limitation regimes in the daytime is expected to be minimal, or at least quite variable, as strong and variable NPQ processes depress daytime active fluorescence in both regions. In contrast, the passive, sunlight stimulated fluorescence is balanced by increased absorbed PAR energy for fluorescence and NPQ processes (see Fig. 3c). Although the scatter in the data is greater for the passive observations in panel 'd', the fold difference between the two nutrient limitation regimes is similar to that of the night-time chlorophyll-normalized active fluorescence in panel 'b' (3.5 and 3.2-fold differences respectively).

(b) panel c – it appears that there are blue dots considerably higher than the blue line curve, but currently they are hidden behind the orange dots. It might be best to plot the blue and orange data on separate panels so the reader can see the scatter better.

R1_15: At low PAR there are indeed some outlier points falling higher than the red line (see below for replot; replaced in revised manuscript). These are however only a minor fraction of the total data and are included in the binned average shown by the thick line/points.

(c) panel c – the overall pattern that emerges in this figure of saturation of fluorescence at high light was already described in the cited Behrenfeld et al. (2009) paper. In this earlier paper, fluorescence quantum yields were calculated using normalized water leaving radiance data where an NPQ correction was applied that followed an inverse light function at saturating light (i.e., NPQ correction removes the dependence of yields on incident light at saturating levels, as illustrated in the current panel c).

R1_16: Indeed, Behrenfeld et al. (2009) used an inverse light function to NPQ-correct satellite fluorescence data. Essentially their NPQ correction, which was multiplication of nFLH by iPAR (they then scaled this to the mean global iPAR, but this does not impact trends), is equivalent to multiplying nFLH by the cosine of the solar zenith angle, which is what we do here (following Gower et al., 2014). The latter could be considered a more straightforward approach (i.e., by assuming the impacts of sunlight stimulation and NPQ have opposite effects on FLH that cancel each other out at elevated PAR values, hence no correction needed). As acknowledged by Behrenfeld et al. (2009), no data were provided or available in the literature that actually showed the functional form of the passive, chlorophyll-normalized fluorescence under iron and nitrogen limitation. Indeed, this might have been quite different (Cullen, 2009), and thus had a major impact on the resultant data. Therefore, our finding that both iron and nitrogen limited sites reach passive fluorescence plateaus at irradiances below that of satellite operational irradiance is a crucial part of our use of the satellite fluorescence a reliable nutrient limitation diagnostic.

Related to this, in order to avoid confusion, we have now renamed ‘relative fluorescence quantum yields’ (defined as Φ_{rel}) to simply F/Chl in the revised manuscript.

References

Behrenfeld, M.J., Westberry, T.K., Boss, E.S., O'Malley, R.T., Siegel, D.A., Wiggert, J.D., Franz, B.A., McLain, C.R., Feldman, G.C., Doney, S.C. and Moore, J.K., 2009. Satellite-detected fluorescence reveals global physiology of ocean phytoplankton. *Biogeosciences* **6**, 779–794.

Cullen, J.J., 2009. Interactive comment on “Satellite-detected fluorescence reveals global physiology of ocean phytoplankton” by MJ Behrenfeld et al. *Biogeosci. Discuss*, 5, pp.S2646-S2655.

Gower, J.F.R., 2014. A simpler picture of satellite chlorophyll fluorescence. *Remote Sens. Lett.* 5, 583-589.

(d) panel d- the red line in this panel is supposed to correspond to MODIS-Aqua data, but the red line is continuous and varies both day and night. Why are MODIS data shown at night when the instrument only collects data near noon? (by the way, you should check for consistency in notation, some places in the text it is called Aqua-MODIS and other MODIS-Aqua, the latter form is the more common).

R1_17: The line showing the MODIS data corresponds to the continuous location along the transect (note that the x-axis of the plot is distance) and not the specific day-night of our fieldwork (overplotted with shading for reference).

In the revised manuscript we correct for MODIS-Aqua and Aqua-MODIS (all to MODIS-Aqua, as recommended).

(e) why compare shipboard data collected on specific days with MODIS data averaged over an entire season?

R1_18: The satellite fluorescence data unfortunately suffers from cloud cover and gaps in the overpass coverage for any given day (particularly pronounced at low latitudes), which largely prevents same-day/location field-satellite match-ups. Therefore, a composite image was used to get optimal coverage.

5. (Figure 4) (a) Why is so much of the subtropical/tropical Pacific whited out in panels a and e? It is not stated in the figure caption. I believe this is because of the tight threshold range of chlorophyll values used for flagging the data, but see comment below.

R1_19: The majority of the whited-out panels corresponds to data with chlorophyll-a concentrations $<0.1 \text{ mg m}^{-3}$. This is the calculated detection limit for MODIS-Aqua FLH (Huot et al. 2013; See also response to R1_24 where this is discussed in more detail). At the lowest chlorophyll-a concentrations, errors in the satellite chlorophyll retrievals can also be up to 50%. In combination, this could introduce major errors/bias in the resultant fields. Evidently, more fieldwork needs to be conducted in these regions to investigate this before they can be robustly interpreted.

The upper boundary of 0.4 mg m^{-3} was set as beyond these values pigment packaging can start to have an important effect (see R1_23 for more detail). As pigment packaging can show important regional variability, a single chlorophyll-to-absorption conversion equation could introduce error/bias. In addition, and perhaps most importantly, these very low and higher chlorophyll-a systems respectively were not investigated in this study with regards to irradiance-passive fluorescence relationships (Fig. 3c) and we therefore cannot be fully confident that the high light passive fluorescence saturation found in our tropical Pacific study region applies in these systems.

References

Huot, Y., Franz, B.A. and Fradette, M., 2013. Estimating variability in the quantum yield of Sun-induced chlorophyll fluorescence: A global analysis of oceanic waters. *Remote Sens. Environ.* **132**, 238-253.

(b) It would be useful to show a comparison of the time series of delta-Chl and delta phi in addition to the time series shown in panels c and d. By visual inspection, I think the chl-phi comparison will show a pretty remarkable correlation, but I could be wrong. What I'm afraid of here is that the delta-phi time series might be an artifact of the chlorophyll normalization rather than the proposed explanation that increased chlorophyll will correspond to more Fe in the system and lower quantum yields. This explanation might be the case to first order, but the apparent correspondence between delta phi and delta chlorophyll looks too close to be accounted for by this simple explanation.

R1_20: The correlation statistic between delta-chl and delta-phi is $R^2=0.40$, $p=6.4e-26$. This is less strong than the correlation between delta-phi and SST ($R^2=0.50$, $p=1.4e-34$). Therefore, whilst we expect both to correlate (i.e., less upwelling, enhanced Fe limitation, and lower chlorophyll are all expected to co-vary), the delta-phi appears much more sensitive to upwelling strength. This is likely because physiological nutrient limitation is directly influenced by nutrient supply rates, whereas chlorophyll-a biomass will also be under top-down grazing control. We have now included this in the revised manuscript. Note that delta-phi is now simply stated as F/Chl in the revised manuscript.

6. (lines 739-742) It would be good to show a figure of this correlation between OC3 chlorophyll and measured chlorophyll, as well as report the slope of the regression, not just the R2 and p value.

R1_21: This has now been included in the revised methods section.

7. (lines 757-769) I do not understand why for the current study it was chosen to revert nFLH to FLH rather than follow published approaches that correct nFLH for NPQ. See note above regarding how the inverse light function of NPQ corrections accounts for the flattening of the fluorescence yield response at saturating light. Following earlier approaches with NPQ corrections might yield a similar qualitative pattern, but it might also give quantitatively higher or lower yields. Another element that is not discussed but has been in earlier papers is that the absolute fluorescence yield is dependent on photoacclimation state, which can give spatial patterns in satellite fields that are not necessarily indicative of different levels of iron stress.

R1_22: We refer the reviewer to our earlier response, R1_16. We also note that throughout our manuscript we referred to our yields as relative, as the absolute values will depend upon their treatment (including for example normalization to absorbed irradiance or simply chlorophyll-a). In the revised manuscript delta-phi is now simply stated as F/Chl to avoid any confusion.

We agree that photoacclimation state might be playing an important role in regulating passive fluorescence yields in higher latitude regions with strong seasonality in available light. This is currently an unknown, with (to our knowledge) no direct seasonal field or laboratory data to support or refute it (i.e., passive fluorescence versus irradiance responses such as in Figure 3c need to be investigated under such conditions with corresponding ancillary data). However, there is minimal variability in upper water column growth irradiance for the equatorial region focussed on here, meaning that changes in photoacclimation state will in this case be minor (see e.g., Behrenfeld et al., 2005 Fig. 2f and Table 1).

Reference

Behrenfeld, M.J., Boss, E., Siegel, D.A. and Shea, D.M., 2005. Carbon-based ocean productivity and phytoplankton physiology from space. *Global biogeochemical cycles*, 19(1).

8. (lines 773-780) (a) why divide satellite FLH data by chlorophyll concentration rather than absorption? Pigment absorption is the more directly measured property from space, while chlorophyll is more derived.

R1_23: We agree that a robust satellite phytoplankton light absorption parameter would be a preferable option for normalizing FLH. However, current algorithms for directly measuring phytoplankton absorption from satellite remain much less well validated in comparison to chlorophyll-a algorithms. We note that the previous attempt to normalize satellite fluorescence to phytoplankton absorption used a general published relationship between absorption and chlorophyll; that is, they introduced an additional further

derivation/assumption regarding a globally-constant chlorophyll–phytoplankton absorption relationship (Behrenfeld et al., 2009). This potentially introduces bias/error as (i) the relationship might not be globally uniform, (ii) the relationship at low chlorophyll-a (which dominates our study region as well as the global ocean) is indistinguishable from linear (see, for example, Figure 4 in Bricaud et al. (1995) and figure reproduced with Equatorial Pacific data below), but a relationship generated across a full chlorophyll-a range (including many very high chlorophyll values in the dataset) leads to an extension of the curvature to the lower chlorophyll-a range (see e.g. Bricaud et al., 1995 Fig. 4 and Behrenfeld et al., 2009 Fig. 2b). For regions with relatively low chlorophyll-a concentrations, representing our study region, chlorophyll-a concentrations themselves therefore currently remain the better parameter for satellite-based normalization.

Plot of phytoplankton absorption at 440 nm versus total chlorophyll-a (chlorophyll-a + divinyl chlorophyll-a) for the Olipac cruise discussed in Bricaud et al. (2004), which investigated the equatorial and subequatorial Pacific region (i.e., similar to this study). The red line is the global relationship (Chl ~0 to >20 mg m⁻³) of Bricaud et al. (1995) $a_{ph}(440)=0.0403Chl^{0.668}$. Note how the curvature in the global fit is not apparent in the Equatorial Pacific data, which are indistinguishable from linear.

References

Behrenfeld, M.J., Westberry, T.K., Boss, E.S., O'Malley, R.T., Siegel, D.A., Wiggert, J.D., Franz, B.A., McLain, C.R., Feldman, G.C., Doney, S.C. and Moore, J.K., 2009. Satellite-detected fluorescence reveals global physiology of ocean phytoplankton. *Biogeosciences* **6**, 779–794.

Bricaud, A., Babin, M., Morel, A. and Claustre, H., 1995. Variability in the chlorophyll-specific absorption coefficients of natural phytoplankton: Analysis and parameterization. *Journal of Geophysical Research: Oceans*, 100(C7), pp.13321-13332.

Bricaud, A., Claustre, H., Ras, J. and Oubelkheir, K., 2004. Natural variability of phytoplanktonic absorption in oceanic waters: Influence of the size structure of algal populations. *Journal of Geophysical Research: Oceans*, 109(C11).

(b) I am concerned about the small range of chlorophyll concentrations over which satellite fluorescence quantum yields were assessed. The lower bound of <0.1 mg Chl m^{-3} is largely based on the original assessment of Abbott and Letelier, prior to actual MODIS measurements. The upper bound of >0.4 mg Chl m^{-3} is not necessary if effects of pigment packaging are accounted for. In the Behrenfeld et al. (2009) study, they show that by correcting for pigment packaging and NPQ, a first order linear relationship exists between chlorophyll and fluorescence yields across chlorophyll concentrations ranging from ~ 0.03 to > 2 mg Chl m^{-3} (see their figure 2). Moreover, if the detection limit truly is 0.1 mg Chl m^{-3} , then calculated fluorescence yields below this threshold would be random (it would look like speckling in a global map), but in the fore-stated 2009 publication it is clear that this is not the case and instead there are very coherent spatial patterns in observed yields across the central ocean gyres.

R1_24: Our lower bound was not based on the pre-launch detection limit of Letelier and Abbott (1996), who estimated this at 0.5 mg Chl m^{-3} , but on Huot et al. (2013). They calculated the detection limit in two ways:

- Firstly, by using the same approach as Letelier and Abbott (1996) but with actual MODIS-Aqua observations. This produced a detection limit between 0.035 and 0.14 mg chl m^{-3} .
- Secondly, by taking the mean and standard deviation of 273,383 groups of 25 adjacent pixels in satellite images. Setting a detection limit requirement of a signal: noise ratio greater than 2 for at least 50% of the points, they found a detection limit of 0.1 mg chl m^{-3} . A similar value for the detection limit was also estimated by Hu et al. (2012).

In addition to fluorescence signals, chlorophyll-a concentration estimates themselves also become less accurate at very low values. For example, in an extensive field-satellite matchup through the North and South Atlantic gyres, Brewin et al. (2016) found that at very low chlorophyll concentrations (<0.05 mg m^{-3}), satellite chlorophyll-a could be 50% of the observed value. Therefore, together, normalization of below-detection-limit fluorescence values to chlorophyll-a concentrations potentially inaccurate by 50% could lead to important inaccuracies in the resultant data fields not related to phytoplankton physiology.

Finally, we note that the field data do not exist to validate either, (i) the irradiance-passive fluorescence response (i.e., Fig. 3c) in the cores of the oligotrophic subtropical gyres where chlorophyll-a reaches such low values, or (ii) the nutrient dependence of potentially (but still an unknown) light-saturated values. We note that the exclusion of these values has no impact on our analysis, which focusses on the tropical Pacific region hosting chlorophyll-a concentrations $> 0.1 \text{ mg m}^{-3}$.

References

- Brewin, R.J., Dall'Olmo, G., Pardo, S., van Dongen-Vogels, V. and Boss, E.S., 2016. Underway spectrophotometry along the Atlantic Meridional Transect reveals high performance in satellite chlorophyll retrievals. *Remote sensing of environment*, 183, pp.82-97.
- Hu, C., Feng, L., Lee, Z., Davis, C.O., Mannino, A., McClain, C.R. and Franz, B.A., 2012. Dynamic range and sensitivity requirements of satellite ocean color sensors: learning from the past. *Applied Optics*, 51(25), pp.6045-6062.
- Huot, Y., Franz, B.A. and Fradette, M., 2013. Estimating variability in the quantum yield of Sun-induced chlorophyll fluorescence: A global analysis of oceanic waters. *Remote Sens. Environ.* 132, 238-253.
- Letelier, R., & Abbott, M. R. (1996). An analysis of chlorophyll fluorescence algorithms for the moderate resolution imaging spectrometer (MODIS). *Remote Sensing of Environment*, 58, 215–223.

9. (lines 830-855) (a) I do not understand the justification for stating that the rate constant for fluorescence loss would not be expected to change at night under iron limiting conditions. If the PQ pool is highly reduced under these conditions (and there is strong published evidence that this is the case), then a back transfer of electrons to Qa will result in an increase in F_o (i.e., an enhanced k_f during the initial FRR flashes). In the absence of disconnected antennae complexes, this would yield a decrease in F_v (as observed), while F_m would not be impacted by iron stress (which seems to be the case in iron/macronutrient co-limited cells).

R1_25: In response to both these questions and those from Reviewer 3 (see below), we now include a fuller formal description of the derivation and interpretation of the active fluorescence parameters. Before providing this, we note a number of key points: Firstly, the previous interpretation of changes in F_o , F_m and F_v by Behrenfeld and colleagues as well as others did not account for the expected (and demonstrable) effect that the marked changes in σ_{PSII} over the diel cycles must be having on all of F_o , F_m and F_v . A detailed mathematical treatment is included below, but the key point is that the changes in σ_{PSII} are, by definition, indicating that the amount of excitation energy being delivered to PSII is varying. As such the corresponding absolute fluorescence yields must, to first order, vary proportionally

(Oxborough et al. 2012). Secondly, the interpretation of 'back transfer' from a reduced PQ pool to Qa is, as far as we are aware (e.g., Behrenfeld and Milligan 2013), solely based on analysis of the very types of active chlorophyll fluorescence we (and Behrenfeld's group) are presenting. Instead, our interpretation would suggest that this is at most a minor component of the 'Fv' signal at night (noting notation issues outlined below). If there is independent evidence that such back transfer occurs, we would be very grateful if the reviewer would direct us to this. In the meantime, we believe the interpretation we provide is both more fully supported by theory and more parsimonious. Finally, we note that the main conclusions of our manuscript are not dependent on detailed interpretation of these diel signals. However, in providing a thorough treatment we hope to correct some previous issues with interpretation of such data in the region which have only recently become apparent as the theoretical treatment has been advanced (Oxborough et al. 2012).

To provide a formal treatment, starting from first principles (see, e.g., Kolber et al. 1998; Kramer et al. 2004; Oxborough et al. 2012) we can write:

$$F_o \propto \frac{k_f}{k_p + k_f + k_d} \sigma_{LHII} [RCII]$$

and:

$$F_m \propto \frac{k_f}{k_f + k_d} \sigma_{LHII} [RCII]$$

Where F_o and F_m are the minimum and maximum fluorescence values measured, k_f , k_p and k_d are the intrinsic rate constants for fluorescence, photochemistry and non-radiative decay respectively, $[RCII]$ is the concentration of RCII within the measured volume and σ_{LHII} is the average absorption cross section of the light harvesting system of all the RCII.

Further taking (Kolber et al. 1998; Oxborough et al. 2021):

$$\sigma_{PSII} = \frac{k_p}{k_p + k_f + k_d} \sigma_{LHII}$$

where σ_{PSII} is absorption cross of PSII photochemistry (as measured using a single turnover active chlorophyll fluorescence technique such as Fast Repetition Rate fluorometry) and for

simplicity neglecting the coefficient of proportionality, which will simply scale the actual measured value(s) of fluorescence for a given instrument, we have:

$$F_o = \frac{k_f}{k_p} \sigma_{PSII} [RCII]$$

and

$$F_m = \frac{k_f(k_p + k_f + k_d)}{(k_f + k_d)(k_p + k_f + k_d)} \sigma_{PSII} [RCII]$$

Thus:

$$F_v = F_m - F_o = \frac{1}{1 + k_d/k_f} \sigma_{PSII} [RCII]$$

And therefore:

$$\frac{F_v}{\sigma_{PSII}} = \frac{1}{1 + k_d/k_f} [RCII]$$

This demonstrates how a proportionality between F_v and σ_{PSII} is expected under conditions where neither k_d or k_f vary. We further note that these are the intrinsic rate constants for non-radiative and fluorescence decay respectively, i.e., they would not be expected to vary if, for example, a proportion of the RCII are shut.

Note, the terminology above is used under conditions where it is explicitly assumed that all of the reaction centres are open at F_o and closed at F_m .

Assuming now that a given measurement actually corresponded to a condition where a proportion of the reaction centres (C) were closed at the point of initiation of the measurement, i.e., where the measured minimal fluorescence $F^{(t)} > F_o$ and thus the variable

fluorescence should for clarity be denoted by a different symbol (usually F_q) to indicate the important difference in variable value and meaning (Genty et al., 1989; Kramer et al., 2004; Oxborough et al., 2012). Note further the prime notation ($'$) is usually added to indicate the measurements are made under the influence of background irradiance, which are normally the cause of closure of a proportion of the reaction centres, but this is clearly not the case if measuring at night, so it is neglected here.

Thus, if it were the case that a proportion (C) of the RCII were closed at night due to backreactions between a reduced PQ pool and the primary acceptor (Qa) of RCII we would have the measured variable fluorescence given by (Genty et al., 1989; Kramer et al., 2004; Oxborough et al., 2012):

$$F_q = F_m - F = F_m - (CF_m + (1 - C)F_o) = (1 - C)F_m - (1 - C)F_o = (1 - C)(F_m - F_o)$$

Due to the minimal fluorescence F resulting from the combined fluorescence from closed centres (C) fluorescing at F_m and the open centres (1-C) fluorescing at F_o , i.e., as $F = CF_m + (1 - C)F_o$

Thus, substituting in above we have:

$$F_q = (1 - C) \frac{1}{1 + k_d/k_f} \sigma_{PSII} [RCII]$$

and hence:

$$\frac{F_q}{\sigma_{PSII}} = (1 - C) \frac{1}{1 + k_d/k_f} [RCII] = (1 - C) \frac{F_v}{\sigma_{PSII}}$$

i.e., the variable fluorescence divided by the absorption cross of PSII photochemistry should drop in proportion to the fraction of closed centres (as an aside, note, rearrangement of above can provide the very well-known result $F_q/F_v = (1-C)$; e.g., Genty et al., 1989; Kramer et al., 2004). As no change in variable fluorescence normalised to the absorption cross of PSII photochemistry is actually observed at night (see Fig. 2f), the data actually indicate that there is unlikely to be significant closure of PSII at night due to back reactions between a reduced PQ pool and Qa.

References

- Genty, B., Briantais, J.M. and Baker, N.R., 1989. The relationship between the quantum yield of photosynthetic electron transport and quenching of chlorophyll fluorescence. *Biochim. Biophys. Acta - Gen. Subj.* **990**, 87-92.
- Kolber, Z.S., Prášil, O. and Falkowski, P.G., 1998. Measurements of variable chlorophyll fluorescence using fast repetition rate techniques: defining methodology and experimental protocols. *Biochim. Biophys. Acta - Bioenerg.* **1367**, 88-106.
- Kramer, D.M., Johnson, G., Kiirats, O. and Edwards, G.E., 2004. New fluorescence parameters for the determination of QA redox state and excitation energy fluxes. *Photosynth. Res.* **79**, 209-218.
- Oxborough, K., Moore, C.M., Suggett, D.J., Lawson, T., Chan, H.G. and Geider, R.J., 2012. Direct estimation of functional PSII reaction center concentration and PSII electron flux on a volume basis: a new approach to the analysis of Fast Repetition Rate fluorometry (FRRf) data. *Limnol. Oceanogr. Meth.* **10**, 142-154.

(b) A direct link between night time increases in fluorescence and disconnected antennae was earlier postulated by Behrenfeld and colleagues, with the specific idea that reduction of the PQ pool at night would signal for a state transition, but these transitions would not be complete (i.e., connection to PSI) in many cases if PSI levels are very low (as indicated in lab studies and during the current study).

R1_26: We agree that this remains a reasonable hypothesis, although, as we argued above, some decoupling from PSI and PSII is likely required as Fm/Chl increases at night (see also Macey et al. 2014).

Reference

- Macey, A.I., Ryan-Keogh, T., Richier, S., Moore, C.M. and Bibby, T.S., 2014. Photosynthetic protein stoichiometry and photophysiology in the high latitude North Atlantic. *Limnol. Oceanogr.* **59**, 1853-1864.

(c) if the Fv change is largely or in part due to back transfers to Qa with a highly reduced PQ pool and the state transitions are triggered by the same reduced pool, then both Fv and sigma-PSII changes will be correlated but not necessarily causatively. Division of Fv by sigma-PSII will thus result in a product that exhibits a dampened correlation with Fv/Fm. It is interesting that covariations in Fv/Fm and sigma-PSII were also often reported by the

Behrenfeld group, but they noted that this was not always the case – as shown in the currently cited Behrenfeld and Kolber (1999) paper in *Science*.

R1_27: See above (R1_25). The absolute values of F_v and σ_{PSII} are expected to be related from first principles and there is, as far as we are aware, no other evidence for back transfer of electrons from the PQ pool acting to close (i.e., reduce Q_a). Although there is evidence for changes in downstream electron transport rates (Behrenfeld and Milligan 2013), this still does not provide any direct evidence for closure of PSII due to a reduced PQ pool. Rather, as indicated above, such an effect has been inferred on basis of previous interpretations of F_v (and F_o) changes which did not fully consider the implications of the changes in σ_{PSII} potentially generated by the state-transitions (and in general energetic decoupling) referenced by the reviewer.

Reference

Behrenfeld, M.J., Worthington, K., Sherrell, R.M., Chavez, F.P., Strutton, P., McPhaden, M. and Shea, D.M., 2006. Controls on tropical Pacific Ocean productivity revealed through nutrient stress diagnostics. *Nature* **442**, 1025-1028.

Referee #2 (Remarks to the Author):

This paper reports ground-truthing measurements of increases in chlorophyll fluorescence of marine phytoplankton communities induced by iron limitation in a transect from low iron, high nitrate (HNLC) waters of the equatorial Pacific to low nitrate, low iron waters farther north. The authors were able to confirm iron limitation in the equatorial Pacific through a combination of iron and nitrate concentration measurements, increases chlorophyll concentrations with iron addition in bottle incubation experiments, molecular biomarkers for iron and nitrogen limitation, and measurement of chlorophyll (Chl) fluorescence and ambient Chl. Most importantly, they showed that the HNLC waters near the equator had high ratios of fluorescence to Chl, something that had been previously well documented in laboratory and field studies, including those in the equatorial Pacific. They further showed that fluorescence to Chl ratios could be measured remotely in response to daytime solar radiation, initially using a ship mounted fluorometer and discrete Chl measurements in samples, but also from archived satellite-based measurements of both chlorophyll and Chl fluorescence collected over an 18-year period in the equatorial Pacific. The data showed that iron limitation in the equatorial Pacific decreased with upwelling strength, and attendant decreases in seawater temperature as a result of the upwelling of colder, iron- and nitrogen-rich deeper waters. Fluorescence-based values for iron limitation then increased with increasing temperature away from upwelling centers as the upwelled water warmed and iron and nitrogen concentrations were drawn down by the growth of the emerging phytoplankton community. The resulting relationship between increasing sea-surface temperature and fluorescence:Chl ratios determined from the satellite data were significant and invariant over many El Niño–Southern Oscillation (ENSO) cycles, but differed by twofold from values predicted from a biogeochemical model for chlorophyll and marine productivity currently used to predict future effects of climate change. These are important results because they suggest that existing and future satellite data can be used to map iron limitation world-wide in the ocean and that models for marine chlorophyll and productivity currently used to predict future effects of Climate Change likely need modification.

We thank the Reviewer for their evaluation and useful comments. Specific responses are provided below.

Specific comments

Lines 118-122 – Yes, these diel changes in phytoplankton pigment fluorescent patterns are well documented for the transition from the iron limited equatorial Pacific to the N-limited North Pacific central gyre.

R2_1: We agree with the reviewer. Although we believe some important new insights were gained from our analysis, in response to feedback from Reviewer 1 and 4, we have now reduced discussion of diel variability in active fluorescence signals in the main text to focus on the main novel aspects of our study.

Lines 131-160 – The lack of detection of PSI and cyt b6/f Prochlorococcus proteins at the iron limited stations 1-3 is problematic as the cyanobacteria would need these proteins for linear electron flow between PSII and flavodoxin, needed for the production of reductant (NADPH) used for the fixation of carbon and the reduction of nitrate to ammonium. Note that PSII proteins and flavodoxin were detected at these stations, suggesting that PSI and the b6/f complexes must have also been present. A possible explanation for this discrepancy is that PSI and b6/f complexes were present in these Prochlorococcus communities, but that their proteins were different enough in these low-iron adapted populations from the protein “standards” used for identification, that they were not identified. The use of appropriate “standards” for protein identification is a common problem with proteomics.

R2_2: In order to assess for lack of detection of strain-specific PSI and cytochrome b6f (but not PSII) we performed an analysis that is outlined in the Supplementary Information (lines in the original manuscript: 678–685), which is repeated below here:

To further investigate if the low abundance/absence of photosystem I and cytochrome b₆-f proteins found in the high nitrate, Fe limited region (Sites 1–3) were due to strain-level differences in sequences that could have restricted their detection relative to low nitrate strains, we performed a further peptide analysis using METATRYP V2.0 (<https://metatryp.whoi.edu>; Ref. 62) to assess for Prochlorococcus strain matches for peptide sequences (Table S3). This analysis showed that all peptides matched to Prochlorococcus strains isolated in the Equatorial Pacific, in addition to strains isolated in other regions.

In general, and in line with the reviewer, we expect the lack of detection implies that they are at sufficiently low abundance not to be observed by this technique and not that there is none at all at the Fe limited sites.

Lines 167-169 – How rapid was this decrease? The data in figure 3a are for the samples taken at the end of the incubation experiments. However, time course data in other experiments conducted in the Equatorial Pacific iron-limited region indicate that such decreases in fluorescence/Chl occur within one day’s time (see Behrenfeld and Milligan 2013). This paper and those cited therein should be mentioned here.

R2_3: This decrease was in the iron addition bioassay experiments that were conducted over ~44-45 hours duration (now stated in the manuscript), but we agree that changes were likely operating on even faster timescales. We note that previous studies have focussed observations on changes in Fv/Fm in response to iron addition, whereas we focus on fluorescence per unit chlorophyll (as this is ultimately what the satellite observes). We nevertheless have now also referenced the suggested review paper at this point.

Lines 180-181 – There is substantial evidence that the highly fluorescent light harvesting

complexes under conditions of iron limitation and high available nitrogen (i.e., HNLC conditions) result not just from LNCs that are "weakly connected to reaction centers" but rather are disconnected to photosynthetic reaction centers (again see Behrenfeld and Milligan 2013).

R2_4: Now rephrased to state 'weak or no energetic connectivity'.

Lines 189 and 190 – This has also been shown previously (again see Behrenfeld and Milligan 2013 and references cited therein), which should be cited here.

R2_5: At this point we are specifically refereeing to chlorophyll-a normalized active fluorescence, which has to our knowledge not been demonstrated (note the Behrenfeld studies show Fv/Fm and other active fluorescence signals).

William Sunda

Referee #3 (Remarks to the Author):

The manuscript reports changes in the chlorophyll-fluorescence characteristics of phytoplankton populations along an oceanic transect. This transect transitions between distinct zones of nutrient limitation. The manuscript correlates measured physiology to changes in the satellite fluorescence record to derive modifications to climate models.

I have been asked to comment on the in situ photo-physiology aspects of this manuscript and I hope the authors find these comments helpful in the revision of the manuscript.

As mentioned in the manuscript, this is not the first report of changes associated with in situ photo-physiology as one moves north from the equatorial Pacific. I think the manuscript does a good job acknowledging the previous observations and interpretations. The manuscript also does a very good job established the nature of nutrient limitation in the different sampling locations.

The novelty associated with this work is the addition of proteomics to mechanistically explain what was previously hypothesized.

We thank the Reviewer for their evaluation and useful comments. Specific responses are provided below.

I have two issues with the manuscript that could be addressed by the authors.

First of all, the manuscript focusses on a derived fluorescence parameter, F_v , which is variable fluorescence. F_v is not directly measured by a fluorometer, but instead it is calculated by subtracting the minimal level of fluorescence (F_o) from the fluorescence maximum (F_m) measured by a fast repetition rate fluorometer. The manuscript goes through some effort to explain why it uses this parameter, but I am confused as to why this decision was made. Ultimately, disconnecting of light harvesting antenna from the photosystems causes an increase in F_o and this is what drives changes in the variable fluorescence parameter (F_v). This is well established in the literature. So, the authors should consider changing the approach to use F_o throughout this work. Supplementary discussion 1 includes a calculation derived from Oxborough et al. (2012). I note they use some different symbols, which is confusing. Please provide more detail about how this equation was derived. I was unable to convert the approach of Oxborough to what is reported here so there appear to be some missing steps or assumptions in this manuscript.

R3_1: Unfortunately, there are some misconceptions in these assertions and we refer the reviewer to Response R1_25 above for a fuller formal treatment. We apologise for the lack of a formal derivation in the previous manuscript and have now provided this, plus have changed our notation to be consistent with Oxborough et al. (use of k_d rather than k_h for the rate constant for non-radiative decay). We note that we actually use interpretations of both F_o , F_m and F_v in our analysis. We do so on the basis of first principle derivations (Genty, 1989; Kramer et al., 2004; Oxborough et al., 2012). In relation to the specific point raised by the reviewer here, it is clear that both F_o and F_m are increasing as a result of Fe stress,

which can, as we argue and the reviewer points out, only be interpreted in terms of energetically decoupled pigment-protein complexes (Macey et al. 2014). We further note that we present the light-saturated active fluorescence (i.e., Fm) as it is most comparable to that of light-saturated passive fluorescence, which is a major focus of this study. We note that chlorophyll-normalized Fm shows marked, ~3-fold changes decrease between both Fe limited and N limited sites and in response to Fe addition at Fe limited sites, strongly suggesting that it is very much impacted by nutrient limitation (in addition to Fo and Fv).

References

- Genty, B., Briantais, J.M. and Baker, N.R., 1989. The relationship between the quantum yield of photosynthetic electron transport and quenching of chlorophyll fluorescence. *Biochim. Biophys. Acta - Gen. Subj.* **990**, 87-92.
- Kramer, D.M., Johnson, G., Kiirats, O. and Edwards, G.E., 2004. New fluorescence parameters for the determination of QA redox state and excitation energy fluxes. *Photosynth. Res.* **79**, 209-218.
- Oxborough, K., Moore, C.M., Suggett, D.J., Lawson, T., Chan, H.G. and Geider, R.J., 2012. Direct estimation of functional PSII reaction center concentration and PSII electron flux on a volume basis: a new approach to the analysis of Fast Repetition Rate fluorometry (FRRf) data. *Limnol. Oceanogr. Meth.* **10**, 142-154.

A key argument associated with the in-situ physiology is the detachment of light harvesting antenna from the photosystems. The manuscript convincingly shows a change in the ratio of photosystem II to photosystem I core proteins via proteomics. However, were the light harvesting antenna not observed in this data set? Knowing the relative abundance changes in these proteins (or if they were even detected) would go a long way to supporting the hypotheses associated with the fluorescence data. Either way, the data shows changes in the proteome of *Prochlorococcus* but the interpretation of this data is based around a wide variety of organisms that have different light harvesting systems from *Prochlorococcus* including cyanobacteria with phycobilisomes (ref 32,37) or even eukaryotes (ref 31,36). This is not appropriate. Finally, the manuscript should not use the acronym LHC as it refers to a specific family of chlorophyll a binding proteins from eukaryotes.

R3_2: We agree with the reviewer that it would have been very useful to show changes in the detached pigment proteins from the proteomics analysis. We indeed searched but could not find these in the proteomics dataset, which could either be due to the protein being at relatively low abundance or due to an annotation issue. We also agree, that whilst it would be desirable to use metaproteomics to do a similar analysis for all of the most abundant phytoplankton types present, the methodology for field samples is not there yet; hence, we focus on one dominant phytoplankton type (i.e., *Prochlorococcus*). We are cautious in the manuscript to note the difference between community level signals in fluorescence and *Prochlorococcus*-specific protein changes. Finally, 'LHC' has now been replaced by the more general term 'pigment protein complexes'.

Referee #4 (Remarks to the Author):

Browning et al present multiple data types from a cruise in the equatorial Pacific aimed at understanding the physiological impacts of nutrient limitation on phytoplankton physiology and optical properties and using constructs from that data to more tightly constrain NPP projections in the context of climate change. The set of experimental measurements presented are uniquely comprehensive and well-integrated, making this study particularly valuable as a synoptic perspective. For example, strikingly clear evidence of nutrient limitation was observed using meta-proteomics approaches and the conclusions from that analysis are also supported by shipboard incubation experiments in which nutrients were added and the chlorophyll response was measured. The comprehensiveness and virtuosity of this work are important because they leave little to doubt about the physiological state of the phytoplankton, so that the association with corresponding optical measurements is sound. This is beautiful work.

We thank the Reviewer for their positive comments. We respond to each specific comment in turn below.

A central self-reported unique contribution from this work is utilizing radiometers on the vessel to derive a relative measurement of sunlight-stimulated chlorophyll fluorescence, and showing that a relative measure of SSCF can be obtained from satellite data, using the assumption of irradiance independence that is supported by data shown in Fig. 3C. The authors then go on to use these constructs to produce global maps of satellite-derived Fe limitation. A number of studies, which are cited, previously have applied remotely sensed optical signatures to predict nutrient limitation and primary productivity in this region of the ocean. Recommendations follow. I like this paper, but I think the message might need to be distilled to its essence.

R4_1: We thank the Reviewer for their recommendations – in addition to comments from other reviewers, we recognize that our main novel message required distilling and we have worked to do this in the revised manuscript.

1. Perhaps a bit simplistic, but isn't it to be expected that iron limitation will be somewhat binary, in that the phytoplankton community either doesn't have enough, or they do. Could the global model then still be right (i.e. not be overestimating NPP sensitivity to Fe input by 2 fold), in the sense that NPP is changing with changing Fe input but not the Fe limitation status at the cellular level? Towards the end of the paper this view is represented but towards the beginning the emphasis seems to be on the model underestimates.

R4_2: This is a good point made by the Reviewer; however, we think that the model should still be able to reflect relative levels of Fe limitation through the model Fe limitation term (in the manuscript methods referred to as L_F). This term is variable between 0-1 depending on Fe availability (intracellular availability, relative to minimum and maximum phytoplankton quotas). This term is then multiplied by the temperature/light regulated maximum

phytoplankton growth rate*, which therefore directly (and dynamically) regulates modelled NPP.

**Provided it remains lower than any other nutrient limitation term for other model nutrients, which is the case in the Equatorial Pacific.*

2. While I was impressed with the quality of this manuscript I was left uncertain about observations that confirm previous reports and observations that are new with respect to satellite detection. I recommend the authors clarify. Specifically, the authors should briefly contrast their approach and findings with previous studies that predicted Fe limitation from satellite data. My impression is this manuscript uses satellite data differently, and is different in emphasizing ENSO variation and its relatively low impact on Fe limitation, and suggesting that models overstate ENSO impacts on Fe limitation. But, is this insight a direct consequence of the new relative measure of SSCF?

R4_3: We have now included a statement about how our field validation of the satellite fluorescence observations, with respect to irradiance and nutrient limitation state in the context of phytoplankton community structure and light absorption properties, now makes interpretation of these signals tractable for the tropical Pacific region. Our finding is that ENSO variability drives major variations in Fe limitation, but these are not so large as the model predicts (see response R4_4 below for more discussion on this). This finding is a direct consequence of validating the satellite data, as field observations are far too scarce to generate reliable time series at any scale (intra-, interannual, or decadal). We also moderately expand our finding that the satellite data showed no evidence for transitions into Equatorial Pacific nitrogen limitation during the El Niño events in the MODIS record, but the capacity now exists to observe this into the future (including responses to stronger El Niño events and ongoing climate change impacts).

3. Line 255, and elsewhere. Central to the findings is the observation that nutrient limitation is resilient to fluctuations in nutrient input caused by ENSO. At various points in the manuscript there is a suggestion that phytoplankton community restructuring could explain this resilience, but the reasoning behind this is unclear. The data in the manuscript don't resolve phytoplankton diversity in great detail, except for the metaproteomics, which are very specific. Are the authors suggesting that taxa adapted to a continuous state of iron limitation inhabit iron-limited ocean regions? My interpretation of the manuscript is that the change in expression of proteins is a reflection of the physiological state of the cells, but elsewhere it sounds like the authors are invoking community turnover. To avoid confusion perhaps be more clear about how community turnover (i.e. changes in communities) could decouple nutrient input from NPP.

R4_4: We have now clarified this in the manuscript by stating that changes in community structure could buffer against the strength of Fe limitation because of variable (i) Fe requirements, and (ii) Fe uptake abilities. Both of these are likely highly simplistic in the

model for the two model phytoplankton types, nanophytoplankton and diatoms, in comparison to the real ocean. We therefore predict that NPP will be maintained to a greater extent in the real ocean in response to lower Fe availability than in the model, due to the possibility for selection from a far more diverse range of phytoplankton with different Fe quotas and uptake mechanisms. A simple, and non-exclusive, example could be a shift to progressively smaller picophytoplankton in the ocean with higher surface area to volume ratios for Fe uptake, whereas the model can only switch from diatoms to nanophytoplankton.

4. Line 796, L Lowercase F and (Line 792), L lowercase Fe. Is that a typo? If not clarify difference. I suggest defining Delta Fe lowercase lim, y axis label Fig. 4f, in legend since its easy for a reader to become confused.

R4_5: We agree this was easy for the reader to become confused. L_{Fe} is the model Fe limitation term, ranging from 0-1, with lower values indicating stronger Fe limitation (i.e., phytoplankton maximum growth rate is multiplied by a smaller value). In Figures 4e,f and S8 we convert this to another Fe limitation term, Fe_{Lim} , where $Fe_{Lim} = 1 - L_{Fe}$, so that higher values indicate stronger Fe limitation. This has now been clarified in the model methods section.

5. If more space is needed to strengthen the narrative then I suggest reducing the section about diel cycles. Its interesting but not essential to the central arguments.

R4_6: We agree with the reviewer, and have indeed shortened the manuscript including significantly (by ca. 500 words) by reducing the discussion of diel active fluorescence cycles and details of phytoplankton responses in the bioassay experiments. We agree that this has strengthened the narrative of the paper.